# Deep learning of left atrial structure and function provides link to atrial fibrillation risk

James P. Pirruccello [1,2,3,4] ✉, Paolo Di Achille [5,6], Seung Hoan Choi [7], Joel T. Rämö [5,8], Shaan Khurshid [5,9,10,11,12], Mahan Nekoui [5,12], Sean J. Jurgens [5,13,14], Victor Nauffal [5,15], Shinwan Kany [5,16], FinnGen*, Kenney Ng [17], Samuel F. Friedman [5,6], Puneet Batra [6], Kathryn L. Lunetta [18], Aarno Palotie [8,19,20], Anthony A. Philippakis [6], Jennifer E. Ho [6,12,21], Steven A. Lubitz [5,9,10,12] & Patrick T. Ellinor [5,9,10,12]

Increased left atrial volume and decreased left atrial function have long been associated with atrial fibrillation. The availability of large-scale cardiac magnetic resonance imaging data paired with genetic data provides a unique opportunity to assess the genetic contributions to left atrial structure and function, and understand their relationship with risk for atrial fibrillation. Here, we use deep learning and surface reconstruction models to measure left atrial minimum volume, maximum volume, stroke volume, and emptying fraction in 40,558 UK Biobank participants. In a genome-wide association study of 35,049 participants without pre-existing cardiovascular disease, we identify 20 common genetic loci associated with left atrial structure and function. We find that polygenic contributions to increased left atrial volume are associated with atrial fibrillation and its downstream consequences, including stroke. Through Mendelian randomization, we find evidence supporting a causal role for left atrial enlargement and dysfunction on atrial fibrillation risk.

Atrial fibrillation (AF) is a common arrhythmia that is projected to affect up to 12 million Americans by 2050[1]. As a leading cause of stroke[2,3], the risk factors for AF have been the subject of extensive investigation[4–6]. Enlargement of left atrial (LA) volumes is commonly observed with hypertension[7], heart failure[8], or after a diagnosis of AF[9,10]—and AF plays a causal role in this process[11]. Enlargement of the LA and decreased LA function have also been identified as independent risk factors for AF[10,12–17] and stroke[18–20]. Together, these atrial structural, contractile, or electrophysiological changes that have clinical consequences have been termed atrial cardiomyopathies[21,22].

The link between LA function and AF risk has prompted interest in determining the heritability and common genetic basis for variation in LA measurements. A large-scale genome-wide association study (GWAS) in 30,201 individuals with LA measurements ascertained by echocardiography did not identify any loci with $P < 5E-08$[23]. Recently, a GWAS of deep learning-derived diastolic measurements in 34,245 UK Biobank participants identified one variant associated with LA volume near *NPR3*[24,25], and a GWAS of a biplanar estimate of LA volume and function identified 14 unique loci in 35,658 participants[26].

Taking advantage of the precision of cardiovascular magnetic resonance imaging (MRI), we developed deep learning models to produce two-dimensional measurements of the LA in 40,558 participants in the UK Biobank[27,28], and applied a surface reconstruction technique to integrate these data into three-dimensional LA volume estimates. We reproduced prior observational associations between LA measurements and AF, heart failure, hypertension, and stroke. We then undertook analyses to identify common genetic variants associated with LA volumes in over 35,000 UK Biobank participants.

A full list of affiliations appears at the end of the paper. *A list of authors and their affiliations appears at the end of the paper. ✉e-mail: james.pirruccello@ucsf.edu

Finally, using common genetic variants as instruments for Mendelian randomization, we performed bidirectional causal analyses between LA volume and AF.

## Results

### Reconstruction of LA volumes from cardiovascular magnetic resonance images

We trained deep learning models to annotate the LA and left ventricular blood pools in four views (distinct models for the short axis view, and the two-, three-, and four-chamber long axis views). We then applied these models to all available UK Biobank cardiovascular magnetic resonance imaging (MRI) data (Methods)[27–29]. The quality of the deep learning models for measuring the LA was higher for the long axis views and lower for the short-axis views, which were not designed to capture the LA (Supplementary Note). We integrated the data from these separate cross-sections to compute the surface of a 3-dimensional representation of the LA (Supplementary Note), yielding LA volume estimates at 50 timepoints throughout the cardiac cycle for 40,558 participants (Fig. 1). We conducted analyses on the maximum LA volume (LAmax), the minimum LA volume (LAmin), the difference between those two volumes (stroke volume; LASV), and the emptying fraction (LASV/LAmax; LAEF), as well as their body surface area (BSA)-indexed counterparts (Supplementary Fig. 1).

### LA traits are associated with AF, heart failure, hypertension, and stroke

We analyzed the pattern of cardiac chamber volumes throughout the cardiac cycle in order to identify individuals with abnormal atrial

contraction (Supplementary Note; Supplementary Fig. 2). Interestingly, a subset of 1013 participants with abnormal cardiac filling patterns had markedly elevated LA volumes, similar to those with pre-existing AF (Fig. 2), and were excluded from downstream analyses.

In the remaining 39,545 participants, we evaluated the association between LA measurements and prevalent or incident AF (Supplementary Note). The LA phenotype most strongly associated with AF was the LA minimal volume (LAmin). The 813 individuals with pre-existing AF had a greater LAmin (+8.8 mL, $P = 9.2E\text{-}117$). In the 2.2 years of follow-up time (mean) available on average after MRI acquisition, the risk of incident AF was increased among those with greater LAmin (293 cases; HR 1.73 per standard deviation [SD] increase; 95% CI 1.60–1.88; $P = 4.0E\text{-}39$). We also observed significant associations between LA measurements and hypertension, heart failure, and stroke (Fig. 3 and Supplementary Tables 1–3), as well as continuous traits such as blood pressure, creatinine, and pack years of tobacco use (Supplementary Data 1).

### Common genetic variant analysis of LA size and function identifies 20 loci

After establishing that the LA measurements replicated previously established clinical associations, we then examined the association between common genetic variants and seven LA traits: LAmax, LAmin, LAEF, and LASV, as well as for BSA-indexed LA volumes. We conducted these analyses in 35,049 participants with genetic data and without a history of AF, coronary artery disease, or heart failure (Table 1; Supplementary Fig. 3). First, we examined the SNP-heritability of the LA traits, which ranged from 0.14 (LAEF) to 0.37 (LAmax; Supplementary Table 4). Genetic correlation between the LA measurements ranged

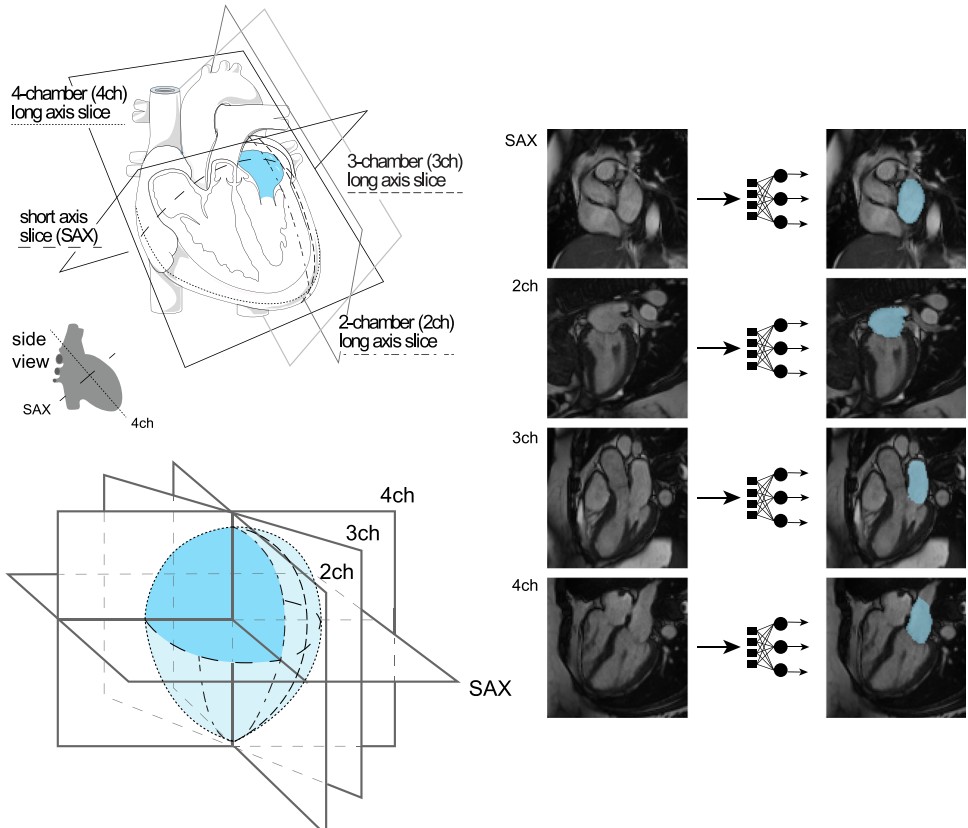

**Fig. 1 | Surface reconstruction for left atrial volume.** Study overview. Top left panel: orientation of the different planes in which images of the atrium were captured. The art in this panel is derived from Servier Medical Art (licensed under creativecommons by attribution, CC-BY-4.0 [https://creativecommons.org/licenses/by/4.0/]). Right panel: Example images from each of the four imaging planes; after interpretation with the deep learning model, the left atrium is colored blue. Reproduced by kind permission of UK Biobank ©. Bottom left panel: schematic overview representing reconstruction of the left atrium based on information obtained from the deep learning output from the four imaging planes.

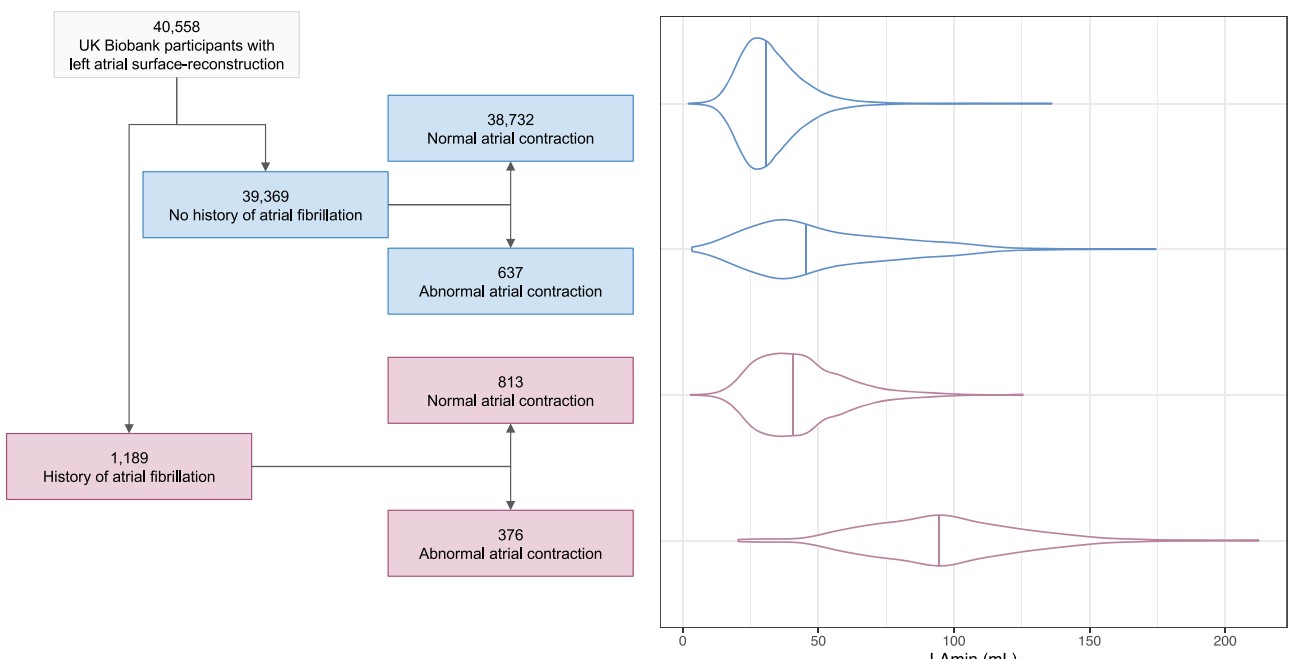

**Fig. 2 | Left atrial volume variation based on AF history and cardiac filling patterns.** In the left panel, a flow diagram breaks down the imaged population into groups with and without AF, and then further into groups that do and do not appear to have normal cardiac filling patterns. In the right panel, the LAmin volume is depicted for these groups with violin plots; the median for each group is demarcated with a vertical line. Source data are provided as a Source Data file.

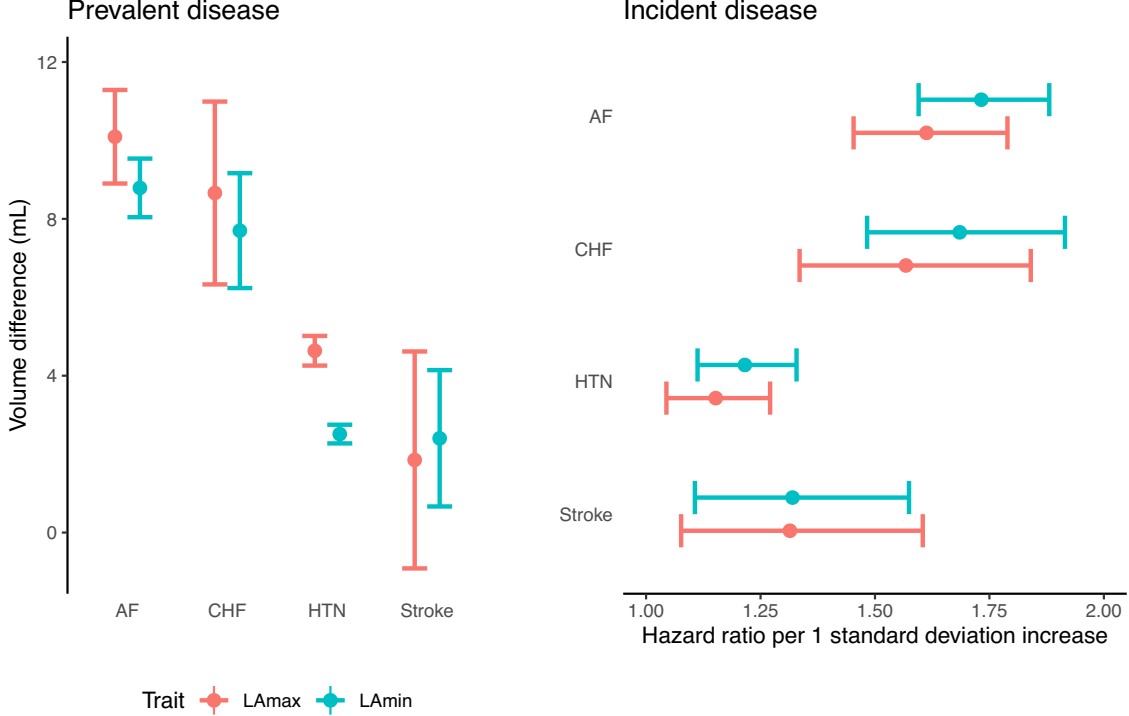

**Fig. 3 | Epidemiological relationships between left atrial volume and disease.** Left panel ("Prevalent disease"): the difference in LA volumes (*Y* axis) between UK Biobank participants with atrial fibrillation ("AF"), heart failure ("CHF"), hypertension ("HTN"), or stroke occurring prior to MRI compared to participants without disease (*X* axis). N = 39,545 participants; 813 with AF, 149 with stroke, 210 with CHF, and 11,852 with HTN. Right panel ("Incident disease"): hazard ratios for incidence of AF, CHF, HTN, and stroke (*Y* axis) occurring after MRI per 1 standard deviation increase in LA volumes (*X* axis). N = 36,900 (fewer due to prevalent disease for CHF and HTN; Supplementary Table 3); 293 with incident AF, 98 with stroke, 125 with CHF, 469 with HTN. Mean volume difference (left panel) or hazard ratio per standard deviation (right panel) estimates are represented by a circle; 95% confidence intervals for the estimate are represented by error bars. Source data are provided as a Source Data file.

**Table 1 | Participant characteristics**

| | Women | Men | Both |
|---|---|---|---|
| N | 18,916 | 16,133 | 35,049 |
| Age at time of MRI | 64 (8) | 65 (8) | 64 (8) |
| BMI (kg/m²) | 26 (5) | 27 (4) | 26 (4) |
| Height (cm) | 163 (6) | 176 (7) | 169 (9) |
| Weight (kg) | 69 (13) | 83 (13) | 75 (15) |
| Systolic blood pressure (mmHg) | 136 (19) | 142 (17) | 139 (19) |
| Diastolic blood pressure (mmHg) | 77 (10) | 81 (10) | 79 (10) |
| Left atrium maximum volume (cm³) | 64 (15) | 79 (19) | 71 (18) |
| Left atrium minimum volume (cm³) | 28 (9) | 37 (12) | 32 (11) |
| Left atrium stroke volume (cm³) | 36 (8) | 43 (11) | 39 (10) |
| Left atrium emptying fraction (%) | 57 (8) | 54 (7) | 56 (8) |
| Mitral regurgitation (%) | 10 (0) | 9 (0) | 19 (0) |
| Mitral stenosis (%) | 3 (0) | 0 (0) | 3 (0) |
| Heart failure (%) | 0 (0) | 0 (0) | 0 (0) |
| Hypertrophic cardiomyopathy (%) | 0 (0) | 0 (0) | 0 (0) |
| Congenital heart disease (%) | 3 (0) | 1 (0) | 4 (0) |
| Aortic valve disease (%) | 18 (0) | 21 (0) | 39 (0) |
| Atrial fibrillation or flutter (%) | 0 (0) | 0 (0) | 0 (0) |

Characteristics of the participants who contributed to the GWAS are listed as mean (standard deviation). Count data are listed as number (%).

from −0.72 (between LAmin and LAEF) to 0.95 (between LAmax and LAmin; Supplementary Table 4).

Next, we performed GWAS for all seven LA traits (Table 2), and as a sensitivity analysis, we also performed GWAS of LA volumes after indexing on left ventricular end-diastolic volume (Supplementary Materials and Supplementary Fig. 4). For all analyses, linkage disequilibrium score regression intercepts were near 1, indicating no significant evidence of inflation due to population stratification (Supplementary Table 5)[30]. No lead SNPs deviated from Hardy-Weinberg equilibrium (HWE) at a threshold of $P < 1E-06$ (Supplementary Data 2)[31].

In the GWAS of LA traits conducted without indexing to BSA, we identified five loci associated with LAmax, eight with LAmin, four with LAEF, and two with LASV (Fig. 4). Four loci were shared between LAmax and LAmin, with lead SNPs near *HLA-B*, *IRAK1BP1*, *BEND3*, and *FBXO32/RSPH6A*. LAmax was additionally associated with SNPs at the *HMGA2* locus, and LAmin was associated with SNPs near *ANKRD1*, *SSSCA1*, *IGF1R*, and *MYO18B*. The four LAEF loci were located near *FAF1*, *CASQ2*, *MYH6*, and *MYO18B*. The two LASV-associated loci included SNPs near *HLA-C* and *MYH6*.

Indexing on BSA yielded three additional loci shared by both LAmax and LAmin (*TTN*, *PITX2*, and *NPR3*), as well as *MYO18B* for LAmax, *UQCRB*, *HTR7*, and *GOSR2* for LAmin, and *OBP2B* for LASV. Additional loci were identified in a sensitivity analysis that accounted for left ventricular end diastolic volume (LVEDV; Supplementary Data 3). Because adjustment for heritable covariates can induce spurious association signals, interpretation of these loci requires caution[32]. Other sensitivity analyses (retaining participants with abnormal cardiac filling patterns; retaining only individuals with inlier genetic identities) are detailed in the Supplementary Note.

**Genetic relationship between AF risk and LA dysfunction**

To gain more insight into the genetic relationship between LA measurements and AF, we first evaluated their genetic correlations. Using

*ldsc*, the strongest genetic correlation was found between LAmin and AF (rg 0.37, $P = 2.0E-10$), a direction of effect that corresponds to a positive correlation between LA dysfunction (i.e., increased LAmin) and risk for AF (Supplementary Table 6)[33,34]. This relationship was minimally attenuated after indexing on BSA (rg 0.33, $P = 7.7E-09$). We also tested for association between LA measurements and stroke (all-cause or cardioembolic) from MEGASTROKE; the strongest association was between LAmin and all-cause stroke with nominal significance (rg 0.21, $P = 0.01$), which was directionally concordant with increased AF risk[35].

We then assessed the overlap between the 20 distinct LA loci identified in our study and 134 loci previously found to be associated with AF[34]. We found that 8 of the 20 LA loci overlapped with an AF locus, which was a significant enrichment based on permutation testing ($P = 1E-04$, which was the minimum possible $P$ value; see Methods)[36]. The 8 loci found in both the LA GWAS and the AF GWAS are nearest to *FAF1/C1orf85*, *CASQ2*, *TTN*, *PITX2*, *MYH6/MYH7*, *IGF1R*, *GOSR2*, and *MYO18B*. At all 8 loci, the effect of each SNP on AF risk was in opposition to its effect on LAEF, and in most cases the effect of each SNP on AF was concordant with its effect on LAmin (Fig. 5). None of the loci that were linked with both LA measurements and AF were associated at genome-wide significance with LAmax.

**Causal link between LA minimum volume and disease risk**

Because the genetic correlation analysis suggested that the strongest cross-trait association was between LAmin and AF, we performed bidirectional Mendelian randomization (MR) analyses to assess whether this relationship was causal. First, we assessed the causal effects of LAmin on the risk for AF. Variants that were associated with LAmin with $P < 1E-06$ were clumped and ambiguous alleles were excluded, leaving 19 SNPs. These variants were cross-referenced in summary statistics from a prior AF GWAS without UK Biobank participants to model the outcome[37]. The inverse variance weighted (IVW) model identified a significant association between LAmin and AF (OR 1.77 per SD increase in LAmin, 95% CI 1.3–2.3, $P = 4.7E-05$). Simple median, weighted median and MR-Egger showed the same direction of effects (Supplementary Fig. 5). There was significant effect heterogeneity ($P = 2.9E-05$ by Cochran Q), so the contamination mixture model approach and MR-PRESSO were applied, both of which showed a significant, positive relationship between LAmin and AF with the same direction of effects (Supplementary Data 4; Supplementary Fig. 5). MR-Egger results did not reach nominal significance, nor did they yield evidence for horizontal pleiotropy (intercept $P = 0.48$). Within the GWAS participants, three of the 19 SNPs had evidence for pleiotropic association with AF risk factors that were derived from the CHARGE-AF risk score (Supplementary Fig. 6)[4]; a sensitivity analysis excluding these three variants yielded similar results (IVW OR 1.89 per SD increase in LAmin, $P = 7.3E-06$; Supplementary Data 4; Supplementary Fig. 7).

Analyses treating each LA measurement as an exposure, using only instruments with $P < 5E-08$, revealed that the strongest statistical relationship was between LAEF and AF (OR 0.36 per SD increase in LAEF, $P = 1.6E-06$; Supplementary Data 5). Expanding the tested outcomes to heart failure[38] and stroke[35] revealed a nominal relationship between greater LAmin and increased risk for heart failure (OR 1.23 per SD increase in LAmin, $P = 0.03$), and between greater LAEF and reduced risk for cardioembolic stroke (OR 0.56 per SD increase in LAEF, $P = 5.3E-03$) but not all ischemic stroke ($P = 0.5$; Supplementary Data 5).

We then tested the causal effect of AF on LAmin. 38 instruments that were also present in the LAmin summary statistics were taken from the 2017 AF GWAS that was conducted without UK Biobank participants[37]. Increasing genetic risk of AF was significantly associated with LAmin (0.086 SD increase per unit increase of log of odds of AF liability, 95% CI 0.049–0.123 SD, $P = 6.2E-06$) using the IVW approach. The simple median, weighted median, MR-Egger bootstrap,

## Table 2 | GWAS lead SNPs

| Trait | CHR | BP | dbSNP | Effect Allele | Other Allele | EAF | BETA | SE | P | Nearest gene |
|---|---|---|---|---|---|---|---|---|---|---|
| LAmax | 6 | 31294375 | rs9265346 | G | A | 0.33 | 0.045 | 0.007 | 1.10E-09 | HLA-B |
| LAmax | 6 | 79554193 | rs6926537 | T | A | 0.512 | -0.039 | 0.007 | 9.10E-09 | IRAK1BP1 |
| LAmax | 6 | 107433645 | rs60237682 | T | C | 0.683 | -0.041 | 0.007 | 3.80E-08 | BEND3 |
| LAmax | 12 | 66343400 | rs1038196 | G | C | 0.486 | 0.039 | 0.007 | 1.10E-08 | HMGA2 |
| LAmax | 19 | 46213416 | rs62111731 | A | G | 0.479 | 0.042 | 0.007 | 7.10E-10 | FBXO46 |
| LAmax indexed | 2 | 179650954 | rs6715901 | G | A | 0.507 | -0.043 | 0.008 | 1.20E-08 | TTN |
| LAmax indexed | 4 | 111665783 | rs2634073 | T | C | 0.166 | 0.057 | 0.01 | 2.00E-08 | PITX2 |
| LAmax indexed | 5 | 32831670 | rs13154066 | T | C | 0.402 | -0.05 | 0.008 | 6.60E-11 | NPR3 |
| LAmax indexed | 6 | 31229203 | rs9264391 | G | A | 0.893 | -0.089 | 0.014 | 9.30E-11 | HLA-C |
| LAmax indexed | 6 | 107442277 | rs9480737 | A | G | 0.682 | -0.05 | 0.008 | 8.80E-10 | BEND3 |
| LAmax indexed | 19 | 46166806 | rs140153691 | T | TGC | 0.639 | -0.047 | 0.008 | 4.10E-09 | GIPR |
| LAmax indexed | 22 | 26156505 | rs133873 | T | A | 0.775 | -0.055 | 0.009 | 9.60E-10 | MYO18B |
| LAEF | 1 | 51322205 | rs79948214 | A | G | 0.984 | 0.161 | 0.029 | 4.70E-08 | FAF1 |
| LAEF | 1 | 116310967 | rs4074536 | T | C | 0.702 | -0.052 | 0.008 | 7.80E-11 | CASQ2 |
| LAEF | 14 | 23874117 | rs440466 | T | C | 0.622 | 0.056 | 0.008 | 7.30E-14 | MYH6 |
| LAEF | 22 | 31294375 | rs133885 | G | A | 0.548 | 0.046 | 0.007 | 3.60E-10 | MYO18B |
| LAEF | 6 | 31294375 | rs9265346 | G | A | 0.33 | 0.044 | 0.007 | 8.30E-09 | HLA-B |
| LAmin | 6 | 79518638 | rs35790661 | C | CCA | 0.63 | -0.043 | 0.007 | 4.40E-09 | IRAK1BP1 |
| LAmin | 6 | 107442277 | rs9480737 | A | G | 0.682 | -0.041 | 0.007 | 3.10E-08 | BEND3 |
| LAmin | 10 | 92681480 | rs780162510 | CATA | C | 0.49 | 0.04 | 0.007 | 2.20E-08 | ANKRD1 |
| LAmin | 11 | 65336819 | rs3782089 | C | T | 0.928 | 0.074 | 0.013 | 4.70E-08 | SSSCA1 |
| LAmin | 15 | 99248018 | rs4966014 | C | T | 0.3 | 0.045 | 0.008 | 4.20E-09 | IGF1R |
| LAmin | 19 | 46315357 | | AT | A | 0.476 | 0.047 | 0.007 | 1.20E-11 | RSPH6A |
| LAmin indexed | 22 | 26164079 | rs133902 | C | T | 0.56 | -0.041 | 0.007 | 4.30E-09 | MYO18B |
| LAmin indexed | 2 | 3912741 | rs56289263 | C | T | 0.938 | -0.093 | 0.017 | 4.40E-08 | DCDC2C |
| LAmin indexed | 2 | 179650954 | rs6715901 | G | A | 0.507 | -0.041 | 0.007 | 4.50E-08 | TTN |
| LAmin indexed | 4 | 24289655 | rs1533093 | C | T | 0.862 | -0.062 | 0.011 | 3.80E-08 | DHX15 |
| LAmin indexed | 4 | 111665783 | rs2634073 | T | C | 0.166 | 0.057 | 0.01 | 1.20E-08 | PITX2 |
| LAmin indexed | 5 | 32831670 | rs13154066 | T | C | 0.402 | -0.043 | 0.008 | 1.40E-08 | NPR3 |
| LAmin indexed | 6 | 31294375 | rs9265346 | G | A | 0.33 | 0.047 | 0.008 | 1.00E-08 | HLA-B |
| LAmin indexed | 6 | 107442277 | rs9480737 | A | G | 0.682 | -0.049 | 0.008 | 1.00E-09 | BEND3 |
| LAmin indexed | 8 | 97223162 | rs35216833 | C | T | 0.456 | 0.042 | 0.007 | 1.00E-08 | UQCRB |
| LAmin indexed | 10 | 92586289 | rs112343361 | A | ACT | 0.497 | -0.046 | 0.008 | 1.80E-09 | HTR7 |
| LAmin indexed | 15 | 99248018 | rs4966014 | C | T | 0.3 | 0.047 | 0.008 | 1.80E-08 | IGF1R |
| LAmin indexed | 17 | 45097337 | rs8078336 | G | T | 0.965 | -0.137 | 0.025 | 2.70E-08 | GOSR2 |
| LAmin indexed | 19 | 46292259 | rs7246377 | G | A | 0.534 | -0.049 | 0.007 | 2.80E-11 | DMWD |
| LAmin indexed | 22 | 26156512 | rs133874 | A | G | 0.773 | -0.061 | 0.009 | 6.80E-12 | MYO18B |
| LASV | 6 | 31225196 | rs199610865 | T | TA | 0.853 | -0.055 | 0.01 | 2.90E-08 | HLA-C |
| LASV | 14 | 23869029 | rs376439 | A | G | 0.606 | 0.045 | 0.007 | 2.70E-10 | MYH6 |
| LASV indexed | 6 | 31225196 | rs199610865 | T | TA | 0.853 | -0.067 | 0.011 | 4.70E-10 | HLA-C |
| LASV indexed | 9 | 136138765 | rs8176685 | GCGCCCACCACTA | G | 0.82 | -0.059 | 0.01 | 3.20E-09 | OBP2B |
| LASV indexed | 14 | 23869029 | rs376439 | A | G | 0.606 | 0.048 | 0.008 | 8.10E-10 | MYH6 |

BP GRCh37-base position, dbSNP dbSNP identifier, where available, EAF Effect allele frequency, BETA BOLT-LMM effect size of the effect allele, in units of the rank-based inverse normal transform which approximates a standard deviation change. SE standard error. P two-tailed BOLT-LMM P value. "Indexed" indicates that the trait has been divided by body surface area. Gene names are italicized.

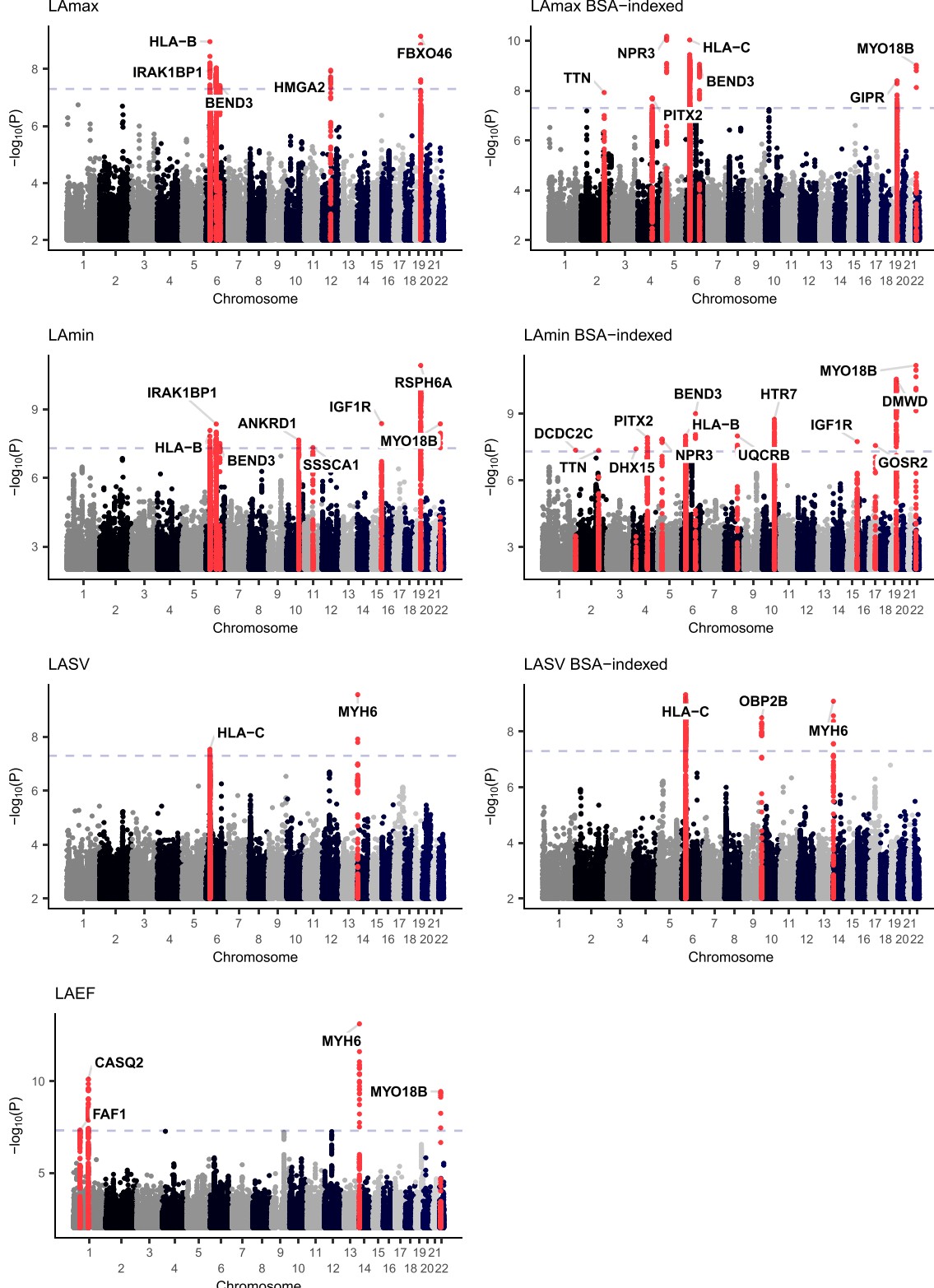

**Fig. 4 | Genome-wide association study Manhattan plots.** Manhattan plots showing the chromosomal position (*X* axis) and the strength of association (−log10 of the *P* value, *Y* axis) for all LA measurements and the BSA-indexed counterparts (except for LAEF, which is dimensionless). Loci that contain SNPs with two-tailed BOLT-LMM *P* < 5E-08 are colored red and labeled with the name of the nearest gene to the most strongly associated variant.

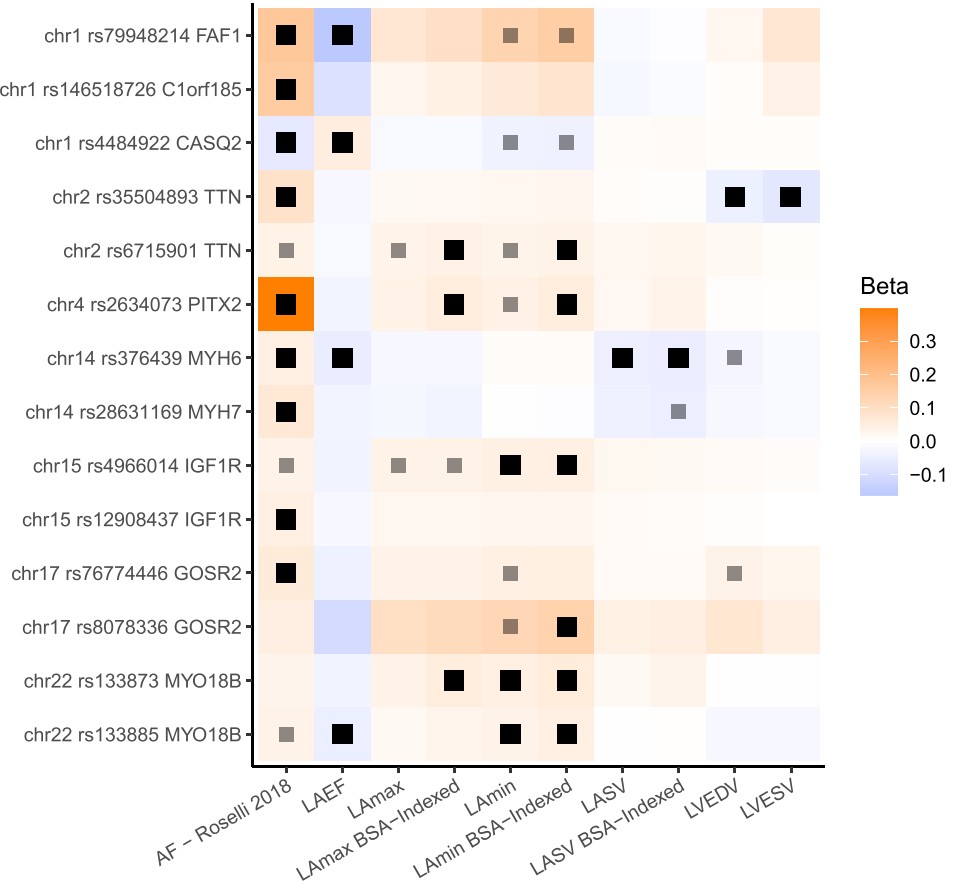

**Fig. 5 | Variants associated with left atrial structure and function and AF.** The 8 loci associated with LA measurements and AF are displayed. All loci (except those near *CASQ2* and *PITX2*) have multiple patterns of linkage disequilibrium and are therefore represented multiple times. Black boxes represent an association with two-tailed BOLT-LMM *P* < 5E-8; lighter gray boxes represent *P* < 5E-6. Effect sizes are oriented with respect to the minor allele. Effect size for AF loci represents the logarithm of the odds ratio. Source data are provided as a Source Data file.

MR-PRESSO, and contamination mixture models exhibited similar directional effects and nominal significance (Supplementary Data 4). The intercept of the MR-Egger and MR-Egger bootstrap were not significantly different from zero (MR-Egger intercept *P* = 0.83, MR-Egger bootstrap intercept *P* = 0.39; Supplementary Data 4, Supplementary Fig. 8).

**A polygenic risk score for AF is associated with LA phenotypes**
We constructed a 1.1-million SNP polygenic risk score (PRS) with PRScs using summary statistics from the Christophersen et al. AF GWAS, and applied this score in the 35,049 LA GWAS participants[37,39]. The AF PRS was statistically significantly associated with all measures of LA size and function, with a small effect size (Supplementary Table 7). The strongest association was with LAmin (0.052 SD increase in LAmin per SD increase in the PRS; 95% CI 0.042–0.061; *P* = 1.1E-25).

**Polygenic estimates of LA volume predict AF, stroke, and heart failure**
We created a 1.1-million SNP genome-wide polygenic score for each LA trait using PRScs[39] and tested each score in up to 423,821 UK Biobank participants who did not participate in the LA GWAS, of whom 417,881 did not have an AF diagnosis at enrollment and 21,147 developed AF afterwards. The strongest association was with the BSA-indexed LAmin polygenic score, which was linked to a modestly increased risk for incident AF or atrial flutter (HR = 1.09 per 1 SD increase in the score; *P* = 7.4E-32) (Fig. 6; Supplementary Table 8). This score was also associated with small increases in risks of incident all-cause stroke (7753

cases; HR = 1.04 per SD; *P* = 4.7E-04), ischemic stroke (5,444 cases; HR = 1.04 per SD; *P* = 4.7E-03), and heart failure (11,035 cases; HR = 1.05 per SD; *P* = 7.9E-08). Those in the top 5% of the score had a greater risk of AF (HR = 1.19, *P* = 7.9E-10), ischemic stroke (HR = 1.12, *P* = 0.06), and heart failure (HR = 1.14, *P* = 1.2E-03; Supplementary Data 6). In a sensitivity analysis that censored participants who developed AF prior to a diagnosis of heart failure, the magnitude of effect and strength of association between the LAmin score and heart failure was attenuated (7,888 cases; HR = 1.03 per SD; *P* = 0.01; Supplementary Data 6). Sensitivity analyses using lead SNP scores, different covariate adjustments, or different population subgroups yielded similar results (Supplementary Data 6).

**External validation of the LAmin polygenic score in FinnGen and *All of Us***
In FinnGen[40] study participants (Supplementary Data 7), comparable associations were observed for association between the BSA-indexed LAmin polygenic score and incident AF or atrial flutter (20,422 cases, HR = 1.08 per SD, *P* = 2.4E-30), ischemic stroke excluding subarachnoid hemorrhage (13,392 cases, HR = 1.03 per SD, *P* = 3.0E-03), ischemic stroke excluding all hemorrhage (11,822 cases, HR = 1.03 per SD, *P* = 5.6E-04), and heart failure (13,771 cases, HR = 1.04 per SD, *P* = 4.4E-06). Compared with the remaining 95% of FinnGen participants, those in the top 5% of genetically predicted LAmin indexed had an increased risk of AF (HR = 1.19 per SD, *P* = 8.4E-09). Those in the top 5% also had elevations in risk that were not statistically significant for ischemic stroke excluding subarachnoid hemorrhages (HR = 1.04 per SD, *P* = 0.36) and heart failure (HR = 1.07, *P* = 0.08).

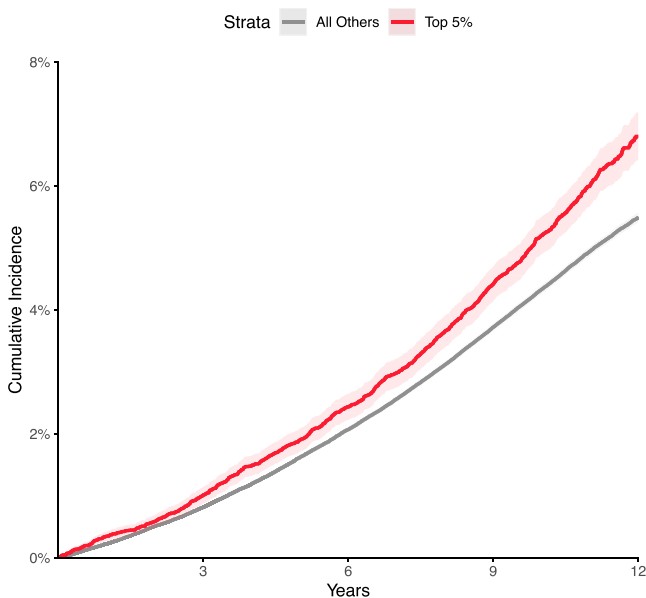

**Fig. 6 | Incident, atrial fibrillation risk, stratified by left atrial polygenic score.** Disease incidence curves for the 417,881 participants who were unrelated to within three degrees of the participants who underwent MRI in the UK Biobank. Those in the top 5% for the BSA-indexed LAmin PRS are depicted in red; the remaining 95% are in gray. The lighter-shaded bands around each line represent the 95% confidence interval. *X* axis: years since enrollment in the UK Biobank. *Y* axis: cumulative incidence of AF (19,875 cases in the bottom 95% and 1272 cases in the top 5%). Those in the top 5% of genetically predicted LAmin indexed had an increased risk of AF (Cox HR 1.19, *P* = 7.9E-10) compared with those in the remaining 95% in up to 12 years of follow-up time after UK Biobank enrollment.

In the US national biobank, *All of Us*[41], the BSA-indexed LAmin polygenic score remained significantly associated with AF (4859 incident cases, HR = 1.06 per SD, *P* = 1.7E-04) and heart failure (5712 incident cases, HR = 1.04 per SD, *P* = 2.0E-02), but not ischemic stroke (66 cases, *P* = 0.3; Supplementary Data 8). In logistic models that included all cases regardless of biobank enrollment date, more cases were identified and the statistical evidence was stronger (13,399 AF cases, OR = 1.10 per SD, *P* = 4.9E-19; 14,572 heart failure cases, OR = 1.04 per SD, *P* = 1.5E-04).

In addition, 680 participants in *All of Us* with genetic data had BSA-indexed LAmin volume measurements. The BSA-indexed LAmin polygenic score was associated with these measurements (0.10 SD per SD of the polygenic score, *P* = 8.5E-03). This relationship remained nominally significant when restricted to only the largest subset of participants by genetic identity (*N* = 619 participants with genetic identity similar to Europeans; 0.09 SD per SD, *P* = 1.5E-2).

## Discussion

We used a unique resource of more than 40,000 cardiac MRI studies available in the UK Biobank to enable a large, high-resolution assessment of LA structure and function. We trained deep learning models to segment LA cross-sections from cardiovascular MRI data and then derived estimates of LA volume from their 3-dimensional reconstructions. In turn, we performed an extensive series of epidemiological, genetic, polygenic, and Mendelian randomization analyses to link these LA traits to cardiovascular outcomes. Our findings permit at least five primary conclusions.

First, we were able to replicate previous observations demonstrating associations between greater LA volume and cardiovascular diseases[7–10,19,20]. Participants with a history of AF had larger LA volumes; and participants with larger LA volumes were more likely to be subsequently diagnosed with AF, stroke, or heart failure.

Second, these measurements enabled a large genetic analysis of LA measurements. In this work, 20 distinct genetic loci were associated with LAmax, LAmin, LAEF, LASV, or the BSA-indexed versions of these phenotypes. To our knowledge, one locus (near *NPR3*) has previously been associated at genome-wide significance with LA measurements in a study of diastolic function[25], while 14 were recently identified in association with LA structure and function[26]. Examining the genetic findings in the present study and in Ahlberg et al. six loci were shared across both studies (near *CASQ2*, *MYO18B*, *TTN*, *UQCRB*, *ANKRD1*, and *RSPH6A/FBXO46/SIX5*); eight were unique to Ahlberg et al. (near *CITED4*, *C9orf3*, *BEND7*, *MGAT1*, *DSP*, *CILP*, *COL8A1*, and *EIF2D*); and fourteen were unique to the present study (near *HLA-B*, *IRAK1BP1*, *BEND3*, *HMGA2*, *PITX2*, *NPR3*, *FAF1*, *MYH6*, *SSSCA1*, *IGF1R*, *DCDC2C*, *DHX15*, *GOSR2*, and *OBP2B*). We considered this overlap in loci to be substantial, particularly since the studies used completely different deep learning models to identify the LA, and different formulas to compute LA volume from the deep learning model output (biplane *vs* surface reconstruction). Forty percent of the loci in our study (eight of 20) were previously associated with AF[34], significantly more than expected by chance. At all eight loci, the allele associated with increased AF risk was directionally associated with a lower LAEF, and generally with greater LA volumes (Fig. 5). The opposed effect directions of these SNPs for AF risk and LAEF may be consistent with the concept of atrial cardiomyopathy[22].

As an example of the pattern of opposed SNP effects on LAEF and AF risk, we identified a missense variant within *CASQ2* (rs4074536; p.Thr66Ala) as a lead SNP for LAEF on chromosome 1. The T allele of this SNP (encoding Thr66) corresponds with a reduced LAEF in our GWAS, and with reduced expression of *CASQ2* in the right atrial appendage and left ventricle in GTEx[42]. This variant is also in LD (*r*² = 1.0) in non-African 1KG populations for the AF lead SNP rs4484922[34,43]. In the study by Roselli and colleagues, the rs4484922-G allele was associated with an increased risk for AF; notably, that risk-increasing allele corresponds to the LAEF-reducing T allele of rs4074536. The rs4074536-T allele has also previously been associated with a longer QRS complex duration[44,45]. *CASQ2* encodes calsequestrin 2, which resides in the sarcoplasmic reticulum in abundance and binds to calcium ions during the cardiac cycle. Missense variants in this gene have also been associated with catecholamine-induced polymorphic ventricular tachycardia, typically following a recessive inheritance pattern[46,47].

Even among LA-associated loci that were not previously associated with AF, several showed the same consistent pattern of inverse effect between AF risk and LAEF (e.g., near *NPR3*, *SSSCA1*, and *HMGA2*). However, this pattern did not uniformly hold. For example, at the gene-dense locus near *FBXO46/DMWD/RPSH6A*, the LA volume-increasing (and LAEF-decreasing) variants were weakly associated with decreased AF risk.

Also notable was the *PITX2* locus, which was the first locus associated with AF. In the present GWAS, SNPs at that locus were associated with BSA-indexed LAmax and LAmin. The lead SNP for AF (rs2129977 from Roselli et al. 2018) was in close LD with the lead SNP for LAmax and LAmin (rs2634073; *r*² = 0.85)[34,43]. Consistent with clinical expectations, the AF risk allele was associated with greater LA maximum and minimum volumes. These analyses excluded participants with a history of AF or abnormal cardiac filling patterns on MRI; therefore, these results support the hypothesis that the *PITX2* locus may be associated with an increase in LA volume that occurs prior to AF onset, which would be consistent with experimental data showing atrial enlargement during embryonic development in mice with knocked-down *PITX2*[48].

Fourth, we developed polygenic scores to gain additional insight into the relationship between LA volumes and cardiovascular diseases. A genome-wide 1.1-million variant AF PRS derived from Christophersen et al. 2017 was associated with all of the LA phenotypes—and most

strongly with LAmin—even after excluding participants known to have AF[37]. This genetic evidence is consistent with and extends prior observational evidence, and suggests that some of the genetic drivers of AF risk may manifest in ways that are detectable in LA size and function.

A 1.1-million variant polygenic predictor of BSA-indexed LAmin was modestly associated with incident AF (Fig. 6), and weakly with stroke, in the UK Biobank. The score was also associated with heart failure—an association that was almost completely attenuated after excluding participants who were diagnosed with AF prior to heart failure. This attenuation suggests that much of the heart failure association may be mediated through AF. The association between greater genetically predicted BSA-indexed LAmin volume, heart failure, and atrial fibrillation was validated externally in FinnGen and *All of Us*, and the weak but statistically significant increased risk of ischemic stroke was also confirmed in FinnGen.

Finally, we found evidence of substantial genetic correlation between LA phenotypes and AF. We pursued Mendelian randomization analyses to more formally assess the hypothesis of bidirectional causation between LA phenotypes and AF. These revealed strong evidence of a causal effect of AF on LAmin, as has been previously observed[11]. There was also evidence that LA volumes, particularly LAmin, may be causal for AF. The causal effect persisted even after excluding three variants associated with at least one risk factor from CHARGE-AF[4]. However, because AF can be paroxysmal and remain undiagnosed, we cannot exclude the possibility of cryptic reverse causation: namely, that some participants may have had larger atria because of undiagnosed paroxysmal AF, such that AF itself induced the genetic association with LA volumes.

In future work, it will be interesting to determine if targeting the genes and pathways associated with abnormalities in LA function will be helpful to reduce the risk of AF, heart failure, and stroke.

This study has several limitations. All LA measurements were derived from deep learning models of cardiovascular MRI. Because a complete trans-axial stack of atrial images was not part of the UK Biobank imaging protocol, the LA measurements are estimates that are interpolated from cross-sections of the LA. Because contrast protocols were not used during image acquisition, we were not able to ascertain atrial fibrosis. The deep learning models have not been tested outside of the specific devices and imaging protocols used by the UK Biobank and are unlikely to generalize to other data sets without fine tuning. Disease labels were determined by diagnostic and procedural codes; because AF can be paroxysmal and may go undetected, it is likely that a subset of the participants had undiagnosed AF prior to MRI, which would bias causal estimates of the impact of LA volume on disease risk away from the null. The study population was largely composed of people of European ancestries, limiting generalizability of the findings to global populations. The participants who underwent MRI in the UK Biobank tended to be healthier than the remainder of the UK Biobank population, which itself is likely to be healthier than the general population. At present, there is little follow-up time subsequent to the first MRI visit for most UK Biobank participants.

In conclusion, measures of LA structure and function are heritable traits that are associated with AF, stroke, and heart failure. Genetic predictors of LA volume are linked to an elevated risk of AF and, to a lesser extent, stroke and heart failure.

## Methods

### Study design
Access to UK Biobank was provided under application #7089 and approved by the Partners HealthCare institutional review board (protocol 2019P003144). All UK Biobank participants provided written informed consent[49]. Analysis of *All of Us* was considered exempt by the UCSF IRB (#22-37715). Each *All of Us* biobank participant provided written informed consent[41]. The FinnGen analysis and approvals are

detailed in the Supplementary Note. Study protocols complied with the tenets of the Declaration of Helsinki. Except where otherwise stated, all analyses were conducted in the UK Biobank, which is a richly phenotyped, prospective, population-based cohort that recruited 500,000 participants aged 40–69 years in the UK via mailer from 2006 to 2010[50]. We analyzed 487,283 participants with genetic data who had not withdrawn consent as of February 2020.

Statistical analyses were conducted with R version 3.6 (R Foundation for Statistical Computing, Vienna, Austria). All statistical tests were two-tailed unless otherwise specified.

### Definitions of diseases and medications
We defined disease status based on self-report, ICD codes, death records, and procedural codes from the UK Biobank's hospital episode statistics data (Supplementary Data 9). These data were obtained from the UK Biobank in June 2020, at which time the recommended phenotype censoring date was March 31, 2020. The UK Biobank defines that date as the last day of the month for which the number of records is greater than 90% of the mean of the number of records for the previous three months (https://biobank.ndph.ox.ac.uk/ukb/exinfo.cgi?src=Data_providers_and_dates).

We identified participants taking antihypertensive medications based on the Anatomical Therapeutic Classification (ATC)[51]. Medications taken by UK Biobank participants were previously mapped to ATC codes[52]. We considered medications with ATC codes beginning with C02, C09, C08CA, C03AA, C08CA01, or C03BA04 to be antihypertensives (medication names enumerated in Supplementary Data 10).

### Cardiovascular MRI protocols
At the time of this study, the UK Biobank had released images in over 45,000 participants of an imaging substudy that is ongoing[27,28]. Cardiovascular MRI was performed with 1.5 Tesla scanners (Syngo MR D13 with MAGNETOM Aera scanners; Siemens Healthcare, Erlangen, Germany), and electrocardiographic gating for synchronization[28]. Several cardiac views were obtained. For this study, four views (the long axis two-, three-, and four-chamber views, as well as the short axis view) were used. In these views, balanced steady-state free precession CINEs, consisting of a series of 50 images throughout the cardiac cycle for each view, were acquired for each participant[28]. For the three long-axis views, only one imaging plane was available for each participant, with an imaging plane thickness of 6 mm and an average pixel width and height of 1.83 mm. For the short-axis view, several imaging planes were acquired. Starting at the base of the heart, 8-mm-thick imaging planes were acquired with ~2 mm gaps between each plane, forming a stack perpendicular to the longitudinal axis of the left ventricle to capture the ventricular volume. For the short axis images, the average pixel width and height was 1.86 mm.

### Semantic segmentation
We labeled pixels using a process similar to that described in our prior work evaluating the thoracic aorta and which we describe here[53]. Cardiac structures were manually annotated in images from the short axis view and the two-, three-, and four-chamber long axis views from the UK Biobank by a cardiologist (JPP) using the *traceoverlay* software v0.1.0[54]. When present, the LA appendage was excluded, as were the pulmonary vein openings; the atrial and ventricular blood pools were distinguished by tracing a linear boundary at the base of the atrio-ventricular ring. To produce the models used in this manuscript, 714 short axis images were chosen, manually segmented, and used to train a deep learning model with PyTorch and fastai v1.0.61[29,55]. The same was done separately with 98 two-chamber images, 66 three-chamber images, and 445 four-chamber images. The models were based on a U-Net-derived architecture constructed with a ResNet34 encoder that was pre-trained on ImageNet[56–59]. The Adam optimizer

was used[60]. The models were trained with a cyclic learning rate training policy[61]. 80% of the samples were used to train the model, and 20% were used for validation. Held-out test sets with images that were not used for training or validation were used to assess the final quality of all models.

Four separate models were trained: one for each of the three long axis views, and one for the short axis view. During training, random perturbations of the input images (augmentations) were applied, including affine rotation, zooming, and modification of the brightness and contrast.

For the short axis images, all images were resized initially to 104 × 104 pixels during the first half of training, and then to 224 × 224 pixels during the second half of training. The model was trained with a mini-batch size of 16 (with small images) or 8 (with large images). Maximum weight decay was 1E-03. The maximum learning rate was 1E-03, chosen based on the learning rate finder[29,62]. A focal loss function was used (with alpha 0.7 and gamma 0.7), which can improve performance in the case of imbalanced labels[63]. When training with small images, 60% of iterations were permitted to have an increasing learning rate during each epoch, and training was performed over 30 epochs while keeping the weights for all but the final layer frozen. Then, all layers were unfrozen, the learning rate was decreased to 1E-07, and the model was trained for an additional 10 epochs. When training with large images, 30% of iterations were permitted to have an increasing learning rate, and training was done for 30 epochs while keeping all but the final layer frozen. Finally, all layers were unfrozen, the learning rate was decreased to 1E-07, and the model was trained for an additional 10 epochs.

For the two-chamber long axis images, all images were resized initially to 104 × 92 pixels during the first half of training, and then to 208 × 186 pixels during the second half of training. The model was trained with a mini-batch size of 8 (with small images) or 4 (with large images). Maximum weight decay was 1E-03. Per-pixel cross entropy loss was minimized[64]. 30% of iterations were permitted to have an increasing learning rate during each epoch. When training with small images, the maximum learning rate was initially 1E-03, and training was performed over 30 epochs while keeping all weights frozen except for the final layer. When training with large images, the maximum learning rate was set to 1E-03, and the model was trained for 12 epochs while keeping all but the final layer frozen. Finally, all layers were unfrozen, the learning rate was decreased to 1E-06, and the model was retrained for an additional 8 epochs.

For the three-chamber long axis images, all images were resized initially to 128 × 128 pixels during the first half of training, and then to 256 × 256 pixels during the second half of training. The model was trained with a mini-batch size of 4 (with small images) or 2 (with large images). Maximum weight decay was 1E-02. Per-pixel cross entropy loss was minimized[64]. 30% of iterations were permitted to have an increasing learning rate during each epoch. When training with small images, the maximum learning rate was initially 1E-03, and training was performed over 20 epochs while keeping all weights frozen except for the final layer. Then, all layers were unfrozen, the learning rate was decreased to 3E-05, and the model was trained for an additional 20 epochs, with 80% of iterations permitted to have an increasing learning rate during each epoch. When training with large images, the maximum learning rate was set to 3E-04, and the model was trained for 15 epochs while keeping all but the final layer frozen; 20% of iterations were permitted to have an increasing learning rate during each epoch. Finally, all layers were unfrozen, the learning rate was decreased to 1E-07, and the model was retrained for an additional 7 epochs.

For the four-chamber long axis images, all images were resized initially to 76 × 104 pixels during the first half of training, and then to 150 × 208 pixels during the second half of training. The model was trained with a mini-batch size of 4 (with small images) or 2 (with large images). Maximum weight decay was 1E-02. Per-pixel cross entropy loss was minimized[64]. 30% of iterations were permitted to have an increasing learning rate during each epoch. When training with small images, the maximum learning rate was initially 1E-03, and training was performed over 50 epochs while keeping all weights frozen except for the final layer. Then, all layers were unfrozen, the learning rate was decreased to 3E-05, and the model was trained for an additional 15 epochs. When training with large images, the maximum learning rate was set to 3E-04, and the model was trained for 50 epochs while keeping all but the final layer frozen. Finally, all layers were unfrozen, the learning rate was decreased to 1E-07, and the model was retrained for an additional 15 epochs.

Each model was applied to all available images from its respective view that were available in the UK Biobank as of November 2020.

## Semantic segmentation model quality assessment

The quality of the deep learning segmentation output was assessed against manually annotated segmentations in held-out test samples using the Sørensen-Dice coefficient, the Hausdorff distance, and the mean contour distance[65,66]. The Sørensen-Dice coefficient addresses the total segmentation area of the left atrium, and is a dimensionless value that ranges from 0 for an image where no pixels overlap between human and machine labels, to 1 for an image with perfect overlap between human and machine labels. The Sørensen-Dice was calculated by dividing twice the number of overlapping pixels between the two sets (the intersection) by the sum of the individual pixels considered to be left atrium in each set.

The Hausdorff distance and the mean contour distance address the perimeter of the manual and automated segmentations, and to obtain this perimeter the *binary_erosion* function from the python3 library *scikit-image* version 0.19.3 was used. The Hausdorff distance represents the maximum distance in millimeters (mm) for any point in the perimeter of the automated segmentation output to its nearest point in the perimeter of the manually annotated segmentation. The Hausdorff distance was calculated using the *directed_hausdorff* function from the *scipy.spatial.distance* python3 library, version 1.11.4. The mean contour distance represents the average distance in mm of each point on the automated segmentation output to its nearest point in the perimeter of the manually annotated segmentation. The mean contour distance was calculated for each point in the automated segmentation perimeter by testing the distance to every point in the perimeter of the manually annotated data; retaining the minimum distance for each point; and then taking the average for all points in the automated segmentation perimeter.

## Poisson surface reconstruction

To integrate the output from each of the four models into one LA volume estimate, Poisson surface reconstruction was performed[67,68]. Among the views included in the UK Biobank cardiac MRI data set, none fully captures the 3-D anatomical structure of the LA. The short axis stack only occasionally included the lower portion of the chamber, while the three long axis (i.e., two-, three-, and four-chamber) views provided only single-slice cross-sections of the LA at different orientations. To integrate information from the four incomplete MRI views into a consistent 3D representation of the LA anatomy, we followed a procedure similar to Pirruccello et al. (2021)[69]. Briefly, we first co-rotated the LA segmentation maps from the MRI views into the same reference system (shared 3D space) using standard DICOM metadata from the Image Position (Patient) [0020,0032] and Image Orientation (Patient) [0020,0037] tags. Then, the perimeters of each 2D atrial segmentation map were extracted, yielding a sparse 3D point cloud. In addition to the point coordinates, the reconstruction algorithm requires as input a vector representing the local normal directions for each point, which is used to constrain the curvature of the reconstructed surface. In our approach, we assumed that each perimeter point's normal vector lay on the MRI view plane and was radially oriented outwards from the center

of gravity of the LA segmentation from which the point was extracted. Using three inputs, consisting of the points, the normals, and the *depth* argument of 16 (representing the maximum depth of the tree that the library will use for reconstruction), we applied the Poisson surface reconstruction algorithm[67] with the *pypoisson* python binding for the *Screened Poisson Surface Reconstruction* C++ library v6.13[68]. This yielded interpolated 3-D surfaces from the sparse 3D point cloud. This approach is tolerant to missing segmentation data (e.g., from the frequently missing SAX data) as long as not all available points are coplanar. 3D surfaces of the LA were reconstructed for each of the 50 MRI frames acquired during the cardiac cycle. At each timepoint, the volume of the LA was computed from the reconstructed surface model using the *GetVolume* routine for triangulated meshes included in the VTK library (Kitware Inc.). From the reconstructed volume traces, we estimated the maximum and minimum LA volumes, as well as LA stroke volume and emptying fraction.

### Quality control after segmentation and reconstruction

Automated quality control was performed on the segmentation output to flag putatively invalid segmentations separately for each view. Studies were flagged based on the following heuristics: (a) if they had more than 1 connected component (i.e., if there were pixels in more than one connected surface that were being labeled as left atrium); (b) if the maximum single frame-to-frame change in pixels segmented as left atrium during the 50-frame CINE sequence was greater than five standard deviations beyond the population mean; (c) if no pixels were segmented as the left atrium; or (d) if the number of images in the CINE was not 50. The presence or absence of these flags was then tested for association with 3D surface reconstruction failure using logistic regression.

### Identification of abnormal cardiac filling patterns

In order to focus our analyses on normal variation, we sought to exclude participants from the GWAS if they had an abnormal atrial contraction at the time of acquisition of the MRI. Although MRI uses an electrocardiographic (ECG) signal for image acquisition, the underlying ECG signal from the time of MRI signal acquisition is not available for analysis. Therefore, we sought to identify participants who appeared to have abnormal cardiac filling patterns during the MRI as a proxy for this. We trained a deep-learning model to identify the presence or absence of typical patterns of cardiac filling throughout the cardiac cycle.

To create a training set for such a model, we first fetched CINE videos from the 2-, 3-, and 4-chamber long axis views of all participants with a history of atrial fibrillation. A cardiologist (JPP) evaluated whether the videos appeared to represent a typical cardiac cycle including an atrial contraction. A deep learning model was then trained to classify filling patterns as representing normal cardiac filling or not based on the segmentation output from the semantic segmentation deep learning models. Each input channel represented the pixel counts of a cardiac chamber from a different long axis view, normalized by the maximum number of pixels seen for each channel for that participant, over the entire cardiac cycle. The normalization step prevented the model from accessing information about the absolute size of the chambers, forcing it instead to identify patterns based on relative size changes throughout the cardiac cycle. In total, 8 channels were used as input: four from the 4-chamber long axis images (left atrium, right atrium, left ventricle, right ventricle), two from the 3-chamber long axis images (left atrium, left ventricle), and two from the 2-chamber long axis images (left atrium, left ventricle). Cases were excluded if all 8 channels were not available. Therefore, the shape of the input was 50×8 (8 channels for 50 time steps). Training was performed with FastAI version 2.2.5[29], using the TimeseriesAI library version 0.2.15 (github.com/timeseriesAI/tsai) to train an InceptionTime model[70]. The Ranger optimization function was used with cross entropy loss, and the number of filters in the InceptionTime model was 32, all of which are the software defaults in the TimeseriesAI library. Ranger incorporates RAdam and Lookahead to improve training stability early and later during training, respectively[71,72]. 20% of samples were randomly chosen as the validation set. The model was trained with a batch size of 32. Variable learning rates from 5E-06 to 5E-03 were permitted during training. Training was conducted using the One-Cycle policy for 20 epochs[61,62].

To evaluate the accuracy of the deep learning model, manual evaluation of the cardiac filling patterns was conducted by one cardiologist (JPP) for 100 participants flagged as having abnormal cardiac filling patterns and 100 flagged as having normal cardiac filling patterns, sampled at random from participants without a history of atrial fibrillation. Sensitivity and specificity and their confidence intervals were calculated with the *binom.test* function in R.

### Evaluation of the relationship between the LA, phenotypes, and cardiovascular diseases

For epidemiologic analyses of continuous traits, we performed linear regression, with the LA phenotypes as the dependent variable in a model with the phenotype of interest adjusted for sex, the first five principal components of ancestry, the genotyping array, the MRI scanner, and a third-degree spline of age at the time of imaging to account for possible nonlinear effects of age.

For the disease-based analyses, we focused on four disease definitions related to LA structure and function: AF or flutter, ischemic stroke, hypertension, and heart failure (defined below). For prevalent disease that was diagnosed prior to the time of imaging, linear models were used to test for an association between each disease (as a binary independent variable) and LA phenotypes (as the dependent variables), adjusting for the MRI serial number to account for inter-site differences, sex, age, and the interaction between sex and age.

For incident disease, participants with pre-existing diagnoses prior to the MRI were excluded from the analysis. A Cox proportional hazards model was used, with survival defined as the time between MRI and either the time of censoring, or disease diagnosis. The model was adjusted for the MRI serial number, sex, age, the interaction between sex and age, the cubic natural spline of height, the cubic natural spline of weight, and the cubic natural spline of BMI. As a sensitivity analysis, adjustment was additionally made for heart rate, P duration, QRS duration, P-Q interval, QTc interval, left ventricular end-systolic volume, left ventricular end diastolic volume, and left ventricular ejection fraction.

### Genotyping, imputation, and genetic quality control

UK Biobank samples were genotyped on either the UK BiLEVE or UK Biobank Axiom arrays and imputed into the Haplotype Reference Consortium panel and the UK10K + 1000 Genomes panel[73]. Variant positions were keyed to the GRCh37 human genome reference. Genotyped variants with genotyping call rate <0.95 and imputed variants with INFO score <0.3 or minor allele frequency ≤ 0.005 in the analyzed samples were excluded. After variant-level quality control, 11,253,549 imputed variants remained for analysis.

Participants without imputed genetic data, or with a genotyping call rate <0.98, mismatch between self-reported sex and sex chromosome count, sex chromosome aneuploidy, excessive third-degree relatives, or outliers for heterozygosity were excluded from genetic analysis[73]. Participants were also excluded from genetic analysis if they had a history of AF or flutter, hypertrophic cardiomyopathy, dilated cardiomyopathy, heart failure, myocardial infarction, or coronary artery disease documented prior to the time they underwent cardiovascular MRI at a UK Biobank assessment center. Our definitions of these diseases in the UK Biobank are provided in Supplementary Data 9.

## GWAS of the left atrium

We analyzed the four unadjusted LA phenotypes, as well as LAmax, LAmin, and LASV estimates that were adjusted for BSA or LVEDV (rationale detailed in the Supplementary Note), yielding 10 traits that underwent GWAS. Before conducting genetic analyses, a rank-based inverse normal transformation was applied[74]. All traits were adjusted for sex, age at enrollment, age and age[2] at the time of MRI, the first 10 principal components of ancestry, the genotyping array, and the MRI scanner's unique identifier.

BOLT-REML v2.3.4 was used to assess the SNP-heritability of the phenotypes, as well as their genetic correlation with one another using the directly genotyped variants in the UK Biobank[75]. GWAS for each phenotype were conducted using BOLT-LMM version 2.3.4 to account for cryptic population structure and sample relatedness[75,76]. We used the full autosomal panel of 714,577 directly genotyped SNPs that passed quality control (minor allele frequency ≥0.001; maximum genotype missingness ≤5% for each variant; maximum sample missingness ≤2%) to construct the genetic relationship matrix (GRM), with covariate adjustment as noted above. Associations on the X chromosome were also analyzed, using all autosomal SNPs and X chromosomal SNPs to construct the GRM ($N = 732,214$ SNPs), with the same covariate adjustments and significance threshold as in the autosomal analysis. In this analysis mode, BOLT treats individuals with one X chromosome as having an allelic dosage of 0/2 and those with two X chromosomes as having an allelic dosage of 0/1/2. Variants with association $P < 5 \times 10^{-8}$ were considered to be genome-wide significant[77].

We identified lead SNPs for each trait. Linkage disequilibrium (LD) clumping was performed with PLINK-1.9[31] using the same participants used for the GWAS. We outlined a 5-megabase window (--clump-kb 5000) and used a stringent LD threshold (--$r^2$ 0.001) in order to account for long LD blocks. With the independently significant clumped SNPs, distinct genomic loci were then defined by starting with the SNP with the strongest $P$ value, excluding other SNPs within 500 kb, and iterating until no SNPs remained. Independently significant SNPs that defined each genomic locus are termed the lead SNPs.

HWE for GWAS lead variants was tested using the statistical library available at https://github.com/chrchang/stats (commit @67c3f71), which was written as part of Plink[31].

Linkage disequilibrium (LD) score regression analysis was performed using *ldsc* version 1.0.0[30]. With *ldsc*, the genomic control factor (lambda GC) was partitioned into components reflecting polygenicity and inflation, using the software's defaults.

## Genetic correlation with atrial fibrillation

We used *ldsc* version 1.0.1 to perform cross-trait LD score regression to estimate genetic correlation between the LA measurements, atrial fibrillation (from Roselli et al. 2018), and all-cause or cardioembolic stroke (from Malik et al. 2018)[33–35]. Summary stats were pre-processed with the *munge_sumstats.py* script from *ldsc* 1.0.1 using the default settings, filtering out variants with imputation INFO scores less than 0.9 or minor allele frequencies below 0.01, as well as strand-ambiguous variants.

## Overlap of LA loci with atrial fibrillation loci

We identified the lead SNPs associated with AF from Supplementary Table 16 of Roselli et al.[34]. For this exercise, we used each of the 134 SNPs that achieved association $P < 5E-8$ in the primary GWAS (column 'I') or in the meta-analysis (column 'AD'). We counted the number of AF lead SNPs that fell within 500 kb of the LA lead SNP from our study. We used SNPsnap to generate 10,000 sets of SNPs that matched the LA lead SNPs based on parameters including minor allele frequency, SNPs in linkage disequilibrium, distance from the nearest gene, and gene density[36]. We then repeated the same counting procedure for each of the 10,000 synthetic SNPsnap lead SNP lists, to set a neutral expectation for the number of overlapping AF lead SNPs based on chance.

This allowed us to compute a one-tailed permutation P value (with the most extreme possible P value based on 10,000 randomly chosen sets of SNPs being 1E-04).

## Mendelian randomization

We sought to assess a potential causal relationship between LAmin and AF using Mendelian randomization (MR). We considered LAmin as the exposure and AF as the outcome. The genetic instruments for LAmin were generated using the genome-wide association results from this analysis. The variants from the exposure summary statistics were clumped with $P < 1E-06$, $r^2 < 0.001$, and a radius of 5 megabases using the *TwoSampleMR* package v0.5.7 in R[78]. These stringent clumping thresholds were intended to reduce the risk of including modestly correlated variants as if they were truly distinct instruments despite tagging the same underlying signal (e.g., having an $r^2$ 0.1 with one another). The variants with ambiguous alleles were removed. 19 variants were harmonized with a large AF GWAS that did not include UK Biobank participants[37]. The inverse variance weighted (IVW) method was performed as the primary MR analysis. We also performed simple median, weighted median, MR-Egger, and MR-PRESSO to account for violations of the instrumental variable assumptions[79,80]. Since MR-Egger provides robust estimates under the InSIDE (Instrument Strength Independent of Direct Effect) assumption, we additionally conducted the MR-Egger bootstrap method to confirm the results from MR-Egger. Heterogeneity was tested with Cochran Q[81]. Because of effect heterogeneity, the contamination mixture model approach— which performs robust Mendelian randomization in the presence of invalid instruments—was also employed[82].

To assess risk of pleiotropy of the LA genetic instruments through known pathways, each SNP was tested for association with risk factors from CHARGE-AF[4], an atrial fibrillation risk score, within the same participants in which the GWAS was conducted. Association between each of the 19 variants and seven risk factors (height, weight, systolic blood pressure, diastolic blood pressure, use of antihypertensive medications, diagnosis of diabetes, and current smoking) was tested in a linear regression model that accounted for age and age[2] at the time of MRI, sex, the MRI serial number, the genotyping array, and genetic principal components 1–10. Associations were considered significant if they exceeded Bonferroni significance ($P < 3.8E-04$).

To understand the bidirectional causal effects, we also performed an MR analysis using AF variants from the 2017 GWAS as the exposure and LA measurements as the outcome. After applying the same clumping threshold and filtering methods to AF summary statistics, 38 remaining variants were harmonized with the LAmin association results and used to construct the instrumental variable. The primary and sensitivity analyses were then conducted in the same manner as described above.

Additional Mendelian randomization analyses were conducted using each LA measurement as an exposure constructed from SNPs with $P < 5E-08$, tested against AF[37], heart failure from HERMES[38], and the trans-ancestry ischemic and cardioembolic stroke summary statistics from MEGASTROKE[35].

## Polygenic score for atrial fibrillation

We constructed a 1.1-million SNP PRS using PRScs based on summary statistics from Christophersen et al. 2017—a large AF GWAS that did not incorporate UK Biobank participants[37,39]. The score was constructed from 1,108,410 sites from the summary statistics that overlapped with the HapMap3 sites available in the UK Biobank as precomputed by the PRScs authors. The score was applied to the GWAS participants with LA measurements and tested for association using linear regression (Supplementary Table 7). For comparability, the score and the LA measurements were both standardized to a mean of zero and a standard deviation of 1.

## Derivation of LA measurement polygenic scores

A polygenic score for each LA GWAS was computed using PRScs with a UK Biobank European ancestry linkage disequilibrium panel[39]. This method applies a continuous shrinkage prior to the SNP weights. PRScs was run in 'auto' mode on a per-chromosome basis. This mode places a standard half-Cauchy prior on the global shrinkage parameter and learns the global scaling parameter from the data; as a consequence, PRScs-auto does not require a validation data set for tuning. Based on the software default settings, only the 1.1-million SNPs found at HapMap3 sites that were also present in the UK Biobank were permitted to contribute to the score. Other polygenic scores were produced as sensitivity analyses (Supplementary Note).

## Internal validation of LA polygenic scores in non-imaging participants

The LA polygenic scores were applied to the entire UK Biobank. Participants who had undergone MRI or related within 3 degrees of kinship to those who had undergone MRI, based on the precomputed relatedness matrix from the UK Biobank, were excluded from analysis[73]. We analyzed the relationship between this polygenic prediction of each LA measurement and incident disease (defined by self-report and diagnostic and procedural codes) in the UK Biobank using a Cox proportional hazards model as implemented by the R *survival* package[83]. The primary disease analyzed was atrial fibrillation. For each tested disease, we excluded participants with disease that was diagnosed prior to enrollment in the UK Biobank. We counted survival as the number of years between enrollment and disease diagnosis (for those with disease) or until death, loss to follow-up, or end of follow-up time (for those without disease).

We adjusted for covariates including sex, the cubic basis spline of age at enrollment, the interaction between the cubic basis spline of age at enrollment and sex, the genotyping array, the first five principal components of ancestry, and the cubic basis splines of height (cm), weight (kg), BMI (kg/m2), diastolic blood pressure (mmHg), and systolic blood pressure (mmHg). Sensitivity analyses included restriction participants to a genetic inlier population with European genetic identity (precomputed by the UK Biobank); adjusting for genetic principal components derived from the GWAS samples instead of the entire cohort; adjusting only for age and sex; applying score weights derived from the clumped lead variants with $P < 5E-08$ from each trait instead of PRScs; and thresholding the cohort into the top 5% for each polygenic score compared to the bottom 95% for the score.

## External validation of the BSA-indexed LAmin polygenic score in FinnGen

FinnGen is a collection of prospective Finnish epidemiological and disease-based cohorts and hospital biobank samples[40]. The FinnGen data used here comprise 377,277 individuals from FinnGen Data Freeze 9 (https://www.finngen.fi/en). The data were linked by unique national personal identification numbers to the registries of national hospital discharges (available from 1968), cause of death (1969-), medication reimbursement (1964-) and purchase (1995-), specialist outpatient visits (1998-) and primary care visits (2011-). Data comprised in FinnGen Data Freeze 9 are administered by regional biobanks (Auria Biobank, Biobank of Central Finland, Biobank of Eastern Finland, Borealis Biobank, Helsinki Biobank, Tampere Biobank), the Blood Service Biobank, the Terveystalo Biobank, and biobanks administered by the Finnish Institute for Health and Welfare (THL) for the following studies: Botnia, Corogene, FinHealth 2017, FinIPF, FINRISK 1992–2012, GeneRisk, Health 2000, Health 2011, Kuusamo, Migraine, Super, T1D, and Twins). Consortium members are listed in Supplementary Note.

Patients and control subjects in FinnGen provided informed consent for biobank research, based on the Finnish Biobank Act. Alternatively, separate research cohorts, collected prior the Finnish Biobank Act came into effect (in September 2013) and start of FinnGen

(August 2017), were collected based on study-specific consents and later transferred to the Finnish biobanks after approval by Fimea (Finnish Medicines Agency), the National Supervisory Authority for Welfare and Health. Recruitment protocols followed the biobank protocols and were approved by Fimea. The Coordinating Ethics Committee of the Hospital District of Helsinki and Uusimaa (HUS) statement number for the FinnGen study is Nr HUS/990/2017.

The FinnGen study is approved by Finnish Institute for Health and Welfare (permit numbers: THL/2031/6.02.00/2017, THL/1101/5.05.00/2017, THL/341/6.02.00/2018, THL/2222/6.02.00/2018, THL/283/6.02.00/2019, THL/1721/5.05.00/2019 and THL/1524/5.05.00/2020), Digital and population data service agency (permit numbers: VRK43431/2017-3, VRK/6909/2018-3, VRK/4415/2019-3), the Social Insurance Institution (permit numbers: KELA 58/522/2017, KELA 131/522/2018, KELA 70/522/2019, KELA 98/522/2019, KELA 134/522/2019, KELA 138/522/2019, KELA 2/522/2020, KELA 16/522/2020), Findata permit numbers THL/2364/14.02/2020, THL/4055/14.06.00/2020,,THL/3433/14.06.00/2020, THL/4432/14.06.00/2020, THL/5189/14.06.00/2020, THL/5894/14.06.00/2020, THL/6619/14.06.00/2020, THL/209/14.06.00/2021, THL/688/14.06.00/2021, THL/1284/14.06.00/2021, THL/1965/14.06.00/2021, THL/5546/14.02.00/2020, THL/2658/14.06.00/2021, THL/4235/14.06.00/202, Statistics Finland (permit numbers: TK-53-1041-17 and TK/143/07.03.00/2020 (earlier TK-53-90-20) TK/1735/07.03.00/2021, TK/3112/07.03.00/2021) and Finnish Registry for Kidney Diseases permission/extract from the meeting minutes on 4th July 2019.

The Biobank Access Decisions for FinnGen samples and data utilized in FinnGen Data Freeze 9 include: THL Biobank BB2017_55, BB2017_111, BB2018_19, BB_2018_34, BB_2018_67, BB2018_71, BB2019_7, BB2019_8, BB2019_26, BB2020_1, Finnish Red Cross Blood Service Biobank 7.12.2017, Helsinki Biobank HUS/359/2017, HUS/248/2020, Auria Biobank AB17-5154 and amendment #1 (August 17 2020), AB20-5926 and amendment #1 (April 23 2020) and it´s modification (Sep 22 2021), Biobank Borealis of Northern Finland_2017_1013, Biobank of Eastern Finland 1186/2018 and amendment 22 § /2020, Finnish Clinical Biobank Tampere MH0004 and amendments (21.02.2020 & 06.10.2020), Central Finland Biobank 1-2017, and Terveystalo Biobank STB 2018001 and amendment 25th Aug 2020.

FinnGen samples were genotyped using Illumina and Affymetrix arrays (Illumina Inc., San Diego, and Thermo Fisher Scientific, Santa Clara, CA, USA). Genotype imputation was performed using a population-specific SISu v3 imputation reference panel comprised high-coverage (25-30x) whole genome sequences from 3775 participants as described in a separate protocol (https://doi.org/10.17504/protocols.io.xbgfijw).

PRS weights were applied using PLINK v1.9[31,84]. Case and control statuses for atrial fibrillation or flutter, ischemic stroke excluding subarachnoid hemorrhage, ischemic stroke excluding all hemorrhages and heart failure were defined based on events in the hospital, cause of death, specialist outpatient, primary care, and medication reimbursement registries at any point during registry follow-up as detailed in Supplementary Data 7. The association of PRS with each outcome was assessed using Cox proportional hazards models with follow-up time scale using sex, baseline age, baseline age squared, 5 genomic principal components, and the genotyping array as fixed-effects covariates.

## External validation of the BSA-indexed LAmin polygenic score in All of Us

*All of Us* is an ongoing, diverse national biobank project in the United States[41]. Data include those from physical examination, biospecimen collection, the electronic health record (EHR), and surveys. All participants provided written, informed consent. At the time of analysis, the controlled-access data release version was 7. Within this release, we identified 245,149 participants with whole genome sequencing data.

At the time of analysis, whole genome sequencing (WGS) had been completed in 245,400 participants. Sequencing and sample quality

control in All of Us has been detailed previously[85,86]. In brief, sequencing was performed with Illumina NovaSeq 6000, aligned GRCh38 and variants called by DRAGEN v3.4.12. A joint call set was prepared centrally by *All of Us*. Sample-level quality control was performed centrally by *All of Us*: exclusion criteria included fingerprint concordance log likelihood ratio ≤−3; sex discordance between self-report and WGS-based chromosomal sex call (if sex reported at birth was either "Male" or "Female"); contamination rate ≥ 3%; or mean coverage <30×, or <90% of bases at 20× coverage, or <8E10 aligned Q30 bases, or <95% of bases in 59 hereditary disease risk genes with 20× coverage. Fingerprint concordance was checked at 114 sites using Picard v2.23.9. Variant-level filtration removed sites with no high-quality genotypes, with ExcessHet <54.69, or with QUAL < 60 for SNPs or <69 for Indels. Ancestry prediction was performed centrally by *All of Us*; briefly, Human Genome Diversity Project and 1000 Genomes samples were used to train a random forest to identify ancestry labels based on PCA from high-quality variant sites, and these loadings were then applied in *All of Us*.

PRScs-based polygenic score weights from the UK Biobank were lifted over from GRCh37 to GRCh38[87]. Polygenic scores were then applied to all participants with WGS as an allelic sum, with an average taken over all of the weights. The UK Biobank GWAS in-sample PCA loadings were applied to the *All of Us* participants in the same way. These were then tested for association with the presence or absence of disease at any point prior to enrollment or during follow-up in a logistic regression model after adjustment for age at enrollment, whether the individual's self-reported sex was male, and the first five principal components of ancestry. Similarly, the association with incident disease was tested with a Cox model with the same covariate adjustments after excluding individuals with disease prior to enrollment. All individuals with available data were analyzed. Sensitivity analyses examining only individuals with the "EUR" ancestry label were also conducted.

Atrial fibrillation was defined to be present starting on the first date any of the following diagnostic or procedural codes were reported:
- ICD10-CM: I48, I48.0, I48.1, I48.11, I48.19, I48.2, I48.20, I48.21, I48.3, I48.4, I48.9, I48.91, I48.92;
- ICD9-CM: 427.31;
- SNOMED: 49436004, 282825002, 426749004, 440059007, 440028005;
- CPT4: 92960.

Heart failure was defined by the following codes:
- SNOMED: 84114007, 42343007, 441530006, 441481004, 194779001, 15781000119107, 88805009, 5148006, 92506005, 10633002, 698296002, 426263006, 82523003, 96311000119109, 194781004, 698594003, 426611007, 15629541000119106, 23341000119109, 48447003, 10335000, 7411000175102, 424404003, 418304008, 443343001, 46113002, 417996009, 443254009, 120871000119108, 120861000119102, 56675007, 49584005, 359617009, 7421000175106, 722095005, 443344007, 153951000119103, 153931000119109, 85232009, 367363000, 83291003, 79955004, 16838951000119100, 44313006, 446221000, 703272007, 703273002

Ischemic stroke was defined by the following codes:
- SNOMED: 371041009, 9901000119100, 422504002

The only volumetric LA measurement available in *All of Us* was the BSA-indexed LAmin volume (labeled "Left atrial End-systolic volume/Body surface area [Volume/Area] by US.2D+Calculated by area-length method"). This was analyzed as a continuous trait and was tested for association with the BSA-indexed LAmin polygenic score with adjustment for age at the time of measurement acquisition, sex, and the first five principal components of ancestry.

## Reporting summary

Further information on research design is available in the Nature Portfolio Reporting Summary linked to this article.

## Data availability

GWAS summary statistics have been deposited in the GWAS Catalog under accession #GCP000842. Polygenic score weights have been deposited at doi:10.5281/zenodo.10814404[88]. LA measurements have been returned to the UK Biobank for use by any approved researcher. UK Biobank data are made available to researchers from research institutions with genuine research inquiries, following IRB and UK Biobank approval. *All of Us* data are available for analysis to qualified researchers on the *All of Us* research platform. FinnGen Freeze 9 GWAS summary statistics are available at https://www.finngen.fi/en/access_results. All other data are contained within the article and its supplementary information. Source data are provided with this paper.

## Code availability

Manual annotation for semantic segmentation was performed using *traceoverlay* v0.1.0[54]. The deep learning models have been returned to the UK Biobank for use by other researchers. The mri_la_poisson.py script used to perform Poisson surface reconstruction from segmentation output may be downloaded from Zenodo (doi:10.5281/zenodo.10811233) and is actively developed at https://github.com/broadinstitute/ml4h, available under an open-source BSD license[89].

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

## Acknowledgements

This work was supported by the Fondation Leducq (14CVD01), and by grants from the National Institutes of Health to Dr. Ellinor (1RO1HL092577, K24HL105780) and Dr. Ho (R01HL134893, R01HL140224, K24HL153669). This work was supported by a John S LaDue Memorial Fellowship, the Sarnoff Cardiovascular Research Foundation Scholar Award, and NIH K08HL159346 to Dr. Pirruccello. Dr. Kany was supported by the Walter Benjamin Fellowship from the Deutsche Forschungsgemeinschaft (521832260). Dr. Jurgens was supported by the Junior Clinical Scientist Fellowship from the Dutch Heart Foundation (grant no. 03-007-2022-0035). Dr. Nauffal is supported by NIH grant 5T32HL007604-35. Dr. Khurshid is supported by NIH grant K23HL169839 and American Heart Association 23CDA1050571. Dr. Lubitz was supported by NIH grants R01HL139731, R01HL157635, and American Heart Association 18SFRN34250007. This work was supported by a grant from the American Heart Association Strategically Focused Research Networks to Dr. Ellinor. This work was funded by a collaboration between the Broad Institute and IBM Research. We would like to thank Mary O'Reilly from the Broad Institute PATTERN Team for contributing to the graphical overview in Fig. 1. We want to acknowledge the participants and investigators of FinnGen study. The FinnGen project is funded by two grants from Business Finland (HUS 4685/31/2016 and UH 4386/31/2016) and the following industry partners: AbbVie Inc., AstraZeneca UK Ltd, Biogen MA Inc., Bristol Myers Squibb (and Celgene Corporation & Celgene International II Sàrl), Genentech Inc., Merck Sharp & Dohme LCC, Pfizer Inc., GlaxoSmithKline Intellectual Property Development Ltd., Sanofi US Services Inc., Maze Therapeutics Inc., Janssen Biotech Inc, Novartis AG, and Boehringer Ingelheim International GmbH. Following biobanks are acknowledged for delivering biobank samples to FinnGen: Arctic Biobank (https://www.oulu.fi/medicine/node/207208), Auria Biobank (www.auria.fi/biopankki), THL Biobank (www.thl.fi/biobank), Helsinki Biobank (www.helsinginbiopankki.fi), Biobank Borealis of Northern Finland (https://www.ppshp.fi/Tutkimus-ja-opetus/Biopankki/Pages/Biobank-Borealis-briefly-in-English.aspx), Finnish Clinical Biobank Tampere (www.tays.fi/en-US/Research_and_development/Finnish_Clinical_Biobank_Tampere), Biobank of Eastern Finland (www.ita-suomenbiopankki.fi/en), Central Finland Biobank (www.ksshp.fi/fi-FI/Potilaalle/Biopankki), Finnish Red Cross Blood Service Biobank (www.veripalvelu.fi/verenluovutus/biopankkitoiminta), Terveystalo Biobank (www.terveystalo.com/fi/Yritystietoa/Terveystalo-Biopankki/Biopankki/) and The Finnish Hematology Registry and Clinical Biobank (https://www.fhrb.fi/). All Finnish Biobanks are members of

BBMRI.fi infrastructure (www.bbmri.fi). Finnish Biobank Cooperative -FINBB (https://finbb.fi/) is the coordinator of BBMRI-ERIC operations in Finland. The Finnish biobank data can be accessed through the Fingenious® services (https://site.fingenious.fi/en/) managed by FINBB. The *All of Us* Research Program is supported by the National Institutes of Health, Office of the Director: Regional Medical Centers: 1 OT2 OD026549; 1 OT2 OD026554; 1 OT2 OD026557; 1 OT2 OD026556; 1 OT2 OD026550; 1 OT2 OD 026552; 1 OT2 OD026553; 1 OT2 OD026548; 1 OT2 OD026551; 1 OT2 OD026555; IAA #: AOD 16037; Federally Qualified Health Centers: HHSN 263201600085U; Data and Research Center: 5 U2C OD023196; Biobank: 1 U24 OD023121; The Participant Center: U24 OD023176; Participant Technology Systems Center: 1 U24 OD023163; Communications and Engagement: 3 OT2 OD023205; 3 OT2 OD023206; and Community Partners: 1 OT2 OD025277; 3 OT2 OD025315; 1 OT2 OD025337; 1 OT2 OD025276. The *All of Us* Research Program would not be possible without the partnership of its participants.

## Author contributions

P.T.E. and J.P.P. conceived of the study. S. Kurshid, K.L.L. and S.A.L. provided input into the analysis plan. J.P.P. annotated images. J.P.P. trained the deep learning models. P.D. performed surface reconstruction. J.P.P., P.D., S.J. and S.H.C. conducted bioinformatic analyses for UK Biobank data. J.P.P. conducted bioinformatic analyses for *All of Us* data. FinnGen and A.P. facilitated the FinnGen analyses, and J.T.R. conducted them. J.P.P., S.H.C., J.T.R. and P.T.E. wrote the paper. MN, S. Kany, V.N., K.N., S.F.F., P.B., A.A.P. and J.E.H. provided critical revisions.

## Competing interests

Dr. Pirruccello has served as a consultant for Maze Therapeutics. Dr. Lubitz is an employee of Novartis as of July 2022. Dr. Lubitz received sponsored research support from Bristol Myers Squibb, Pfizer, Boehringer Ingelheim, Fitbit, Medtronic, Premier, and IBM, and has consulted for Bristol Myers Squibb, Pfizer, Blackstone Life Sciences, and Invitae. Dr. Ng is employed by IBM Research. Dr. Ho is supported by a grant from Bayer AG focused on machine learning and cardiovascular disease and a research grant from Gilead Sciences. Dr. Ho has received research supplies from EcoNugenics. Dr. Philippakis is employed as a Venture Partner at GV; he is also supported by a grant from Bayer AG to the Broad Institute focused on

machine learning for clinical trial design. Dr. Ellinor is supported by a grant from Bayer AG to the Broad Institute focused on the genetics and therapeutics of cardiovascular diseases. Dr. Ellinor has also served on advisory boards or consulted for Bayer AG, Quest Diagnostics, MyoKardia and Novartis. The remaining authors report no disclosures.

## Additional information

[1]Division of Cardiology, University of California San Francisco, San Francisco, CA, USA. [2]Institute for Human Genetics, University of California San Francisco, San Francisco, CA, USA. [3]Bakar Computational Health Sciences Institute, University of California San Francisco, San Francisco, CA, USA. [4]Cardiovascular Genetics Center, University of California San Francisco, San Francisco, CA, USA. [5]Cardiovascular Disease Initiative, Broad Institute of MIT and Harvard, Cambridge, MA, USA. [6]Data Sciences Platform, Broad Institute of MIT and Harvard, Cambridge, MA, USA. [7]Cardiovascular Disease Initiative, Broad Institute, Cambridge, MA, USA. [8]Institute for Molecular Medicine Finland (FIMM), Helsinki Institute of Life Science (HiLIFE), University of Helsinki, Helsinki, Finland. [9]Cardiology Division, Massachusetts General Hospital, Boston, MA, USA. [10]Cardiovascular Research Center, Massachusetts General Hospital, Boston, MA, USA. [11]Demoulas Center for Cardiac Arrhythmias, Massachusetts General Hospital, Boston, MA, USA. [12]Harvard Medical School, Boston, MA, USA. [13]Department of Experimental Cardiology, Amsterdam UMC, University of Amsterdam, Amsterdam, NL, Netherlands. [14]Amsterdam Cardiovascular Sciences, Heart Failure & Arrhythmias, University of Amsterdam, Amsterdam, NL, Netherlands. [15]Division of Cardiovascular Medicine, Brigham and Women's Hospital, Boston, MA, USA. [16]Department of Cardiology, University Heart and Vascular Center Hamburg-Eppendorf, Hamburg, Germany. [17]IBM Research, Cambridge, MA, USA. [18]Department of Biostatistics, Boston University School of Public Health, Boston, MA, USA. [19]Analytic and Translational Genetics Unit, Massachusetts General Hospital and Harvard Medical School, Boston, MA, USA. [20]Stanley Center for Psychiatric Research, Broad Institute of MIT and Harvard, Boston, MA, USA. [21]CardioVascular Institute, Department of Medicine, Beth Israel Deaconess Medical Center, Boston, MA, USA. ✉e-mail: james.pirruccello@ucsf.edu

## FinnGen

Joel T. Rämö[5,8] & Aarno Palotie ⓘ [8,19,20]

A full list of members and their affiliations appears in the Supplementary Information.

