## [Peer Review File · Nature Communications]

Deep Learning of Left Atrial Structure and Function Provides Link to Atrial Fibrillation RiskREVIEWER COMMENTS

Reviewer #1 (Remarks to the Author):

Remarks to the Author

This is an interesting and important study by Pirruccello et al. Pirruccello et al performed a large MRI assessment of LA structure and function. They identified 20 common genetic variants associated with LA volumes or LAEF.

In line with these finding they found that a PRS of the minimal LA volume was associated with AF and interestingly also stroke. Pirruccello et el really deserves credit for all the work that have been put in to the paper in particular the development of the deep learning models to measure LA traits from cardiovascular magnetic resonance imaging. Do you plan to make the algorithm publicly available?

Introduction

Minor:

- The sample size of the GWAS w/ bipolar estimates of LA could also be declared.

Methods

- Could you please provide some more details on the training/test set used to create the model that identify abnormal contraction patterns? How many samples were in the training

set? Could you also provide some more metrics on the performance of this model, such as sensitivity and specificity?

- Could you evaluate the relationship to blood pressure?

- Is it necessary to adjust your cox models for height and weight when adjusting for BMI?

- It is not clear if the measurements of LA volume indexed? It might be more clinically relevant to use BSA indexed measurements.

- It is true that mixed models are able to account for population stratification and relatedness. However, the publications for commonly used software like BOLT-LMM (used in this study) and fastGWA have evaluated the confounding of population stratification on a somewhat homogeneous population with European ancestry, i.e., they have removed ethnical outliers from their UKB population. It is also true that there aren't that many samples that would be filtered from your analyses if you excluded ethnical outliers. Therefore, a sensitivity analyses would most likely show similar results. However, I would consider it problematic QC not to remove PC outliers from your GWAS analyses.

Furthermore, it is unclear how not removing outliers could affect some downstream post-analyses. I would prefer to see PC outliers removed and if included it should be through a meta-analysis. This is after all done in almost all GWAS for a good reason.

- Please provide principal component plots on the GWAS cohort.

- Would it be possible to add a histogram showing the distribution of the phenotypes for each trait? Were the traits directly or indirectly rank-transformed?

- Could you please motivate the reason for adjusting both for age at enrollment and age at the time of MRI?

- It is stated in the method section that you have conducted 10 GWAS's. I understand them to be LAEF, LASV, LAmin, LAmax, BSA indexed LASV, BSA indexed LAmin, BSA index LAmax, LVEDV indexed LASV, LVEDV indexed LAmin, LVEDV indexed LAmax. Why are the LVEDV indexed GWAS's reported as a sensitivity analyses and the BSA indexed GWAS's reported in the main analyses? In many of the post analyses only non-indexed phenotypes are reported. This becomes a bit confusing for the reader and it is unclear what traits you actually have studied. Could you please explain as to why you have made these decisions for your study design?

- In regards to overlap with atrial fibrillation loci. Would it not be more meaningful to see if the actual loci overlap instead of nearest gene?

- In your MR analysis, why did you choose a P-value threshold of 1×10^{-6} ? Could you provide a sensitivity analysis using only variants with $P < 1 \times 10^{-8}$?
- Did you test for heterogeneity in the TwoSampleMR package?
- In your MR analysis, could you test the other LA phenotypes as well? It would also be interesting to use additional exposures/outcomes such as blood pressure, heart failure and stroke.
- In the PRScs analyses, are the prior Beta values used on the ranked-based inverse transformation scale?
- Using a UK biobank European ancestry linkage disequilibrium panel, you use PRScs to calculate a PRS. And then, project the PRS onto the entire UK Biobank with the exception of individuals related to individuals with cardiac MRI. Do you exclude individuals with cardiac MRI in the COX regression, that is: your GWAS sample set, this is not clear to me? Since you are using a European ancestry linkage disequilibrium panel, would it not be more appropriate to perform this post-analysis on individuals with European ancestry only?
- Which PCs are you using? As I understand, when using PRScs-auto you do not need a validation set for tuning parameters but you still need an independent test set for evaluation. My concern here is that you have used the PCs given by UKBB, which might have been generated on the whole UK Biobank dataset. In that case, there is a risk of information leakage when you project your PRS onto the rest of UK Biobank. Because, you have previously adjusted the generated GWAS summary statistics, used as prior for PRScs, on those PCs and the "UK biobank European ancestry linkage disequilibrium panel" is also generated using the same samples. Could the PRS model be projected on to an independent dataset with AF patients to see if incident AF is associated with LAmin PRS or alternatively another approach that would avoid this problem?
- PRScs-auto require a large dataset for good performance. Is the sample size large enough? Could this be benchmarked by comparing performance to a P+T PRS approach by only selecting independent lead SNPs or similar?
- The methods section says that a PRS on LAmin was generated, whereas Figure 6 says PRS for BSA-indexed LAmin. Which one is correct? Why was only LAmin used for PRS? It would be interesting to see a COX analyses with PRS on other LA phenotypes, in particular LAEF and BSA indexed LA.

Minor:

- Hypertension is not mentioned as an outcome in Methods but is shown in Figure 3. Hypertension is also not defined in ST12.
- In the section Genome-wide association study of the left atrium, there is no mention LAEF as a phenotype.
- What are the QC steps for the 714,577 genotyped SNPs used as model SNPs to create the GRM?
- Please explain in more detail why you have chosen a 5 megabase window and a LD threshold of 0.001, since this is uncommon.
- I suggest that instead of writing “a commonly used threshold” to give a reference.
- In the PRS COX-model, is it necessary to adjust for height, weight and BMI? In particular if you have used BSA-indexed LAmin?

Results

Major:

- It is important to thoroughly evaluate the performance of the algorithm, since it has not been published previously. Could you include mean contour distance and Hausdorff distance? How do these metrics compare to previously published methods by Bai et al. (J. Cardiovasc. Magn. Reson., 2018)?
- In order to gauge how the measurements hold up against manual annotation and to see if there are unknown biases, I would like to see inter-observer variability between manual and estimated minimum/maximum volumes. Could this be performed by independent clinical experts on a minimum of 30 samples and be shown graphically?
- Would be nice to get some more details on the performance on the model used to classify abnormal atrial contraction.
- In which units are the Beta estimates in table 2, rank-based inverse transformation?

- In the heritability & genetic correlation analysis you don't provide estimates for BSA-indexed phenotypes, which you say (might be the case?) are in the main analysis? The same goes for Idsc intercepts.

- As a suggestion, if you are indeed indexing for LVEDV in the sensitivity analyses, you might want to reconsider and instead perform a conditional analysis with LVEDV.

- In the genetic correlation analysis with other traits, you are only reporting on non-indexed traits. Does this mean BSA-indexing is part of a sensitivity analysis?

- Is the correlation to AF and all cause stroke in part through other risk factors relating to body size, e.g. hypertension, rather than LA volume?

- The section on AF PRS association with LA phenotypes is not described in Methods.

- In the section on LA volume PRS to predict AF, the population used is better described. I think the method and result section on the PRS post-analysis probably needs to be synced. However, there is still a potential issue with information leakage by PC's when predicting AF. Still not clear whether this is LAmin or BSA-indexed LAmin, which it is says in Figure 6. Also, the figure is showing top 5% PRS, which is not really reported in the results.

- Eight of the loci associated with LA traits have also previously been associated with AF. Interestingly they find that "the AF risk alleles were associated with an increased LA minimum volume (LAmin) and a decreased LAEF. A Mendelian randomization analysis confirmed that AF causally affects LA volume (IVW $P = 6.2E-06$), and provided evidence that LAmin causally affects AF risk (IVW $P = 4.7E-05$)". Were these SNPs associated with hypertension or LVEF, could this be secondary to systolic dysfunction?

- A large part of the cases have hypertension have you tested the genetic correlation with Hypertension?

-

Discussion

- .

- The discussion mentions the use of a PRS on BSA-indexed LAmin as opposed to LAmin in the Method.

Conclusion

"In future work, it will be interesting to determine if targeting the genes and pathways associated with abnormalities in LA function will be helpful to reduce the risk of AF, heart failure, and stroke"

This a perspective that has not really been discussed in the discussion, I suggest that you discuss it or removed it from the conclusion.

Figures

- Figure 4, gene names in figures overlap.

- Figure 5, would you say that BSA-indexed LA volumes overlap more with AF loci?

- Supplementary figure 3 and 5, why is effect size of LAmin represented as LAESV?

Reviewer #2 (Remarks to the Author):

This paper investigates genetic factors that contribute to CMR-derived parameters of atrial structure and function using the UK Biobank data. Finding sensitive and accurate ways to assess atrial cardiomyopathy is a clinically important topic of current interest.

The first section describes application of deep learning methods to assess left atrial maximal and minimal volumes and the derived parameters, left atrial stroke volume and ejection fraction. This is a novel way to rapidly obtain these parameters in a standardised fashion. In a similar study design, Ahlberg et al (Eur Heart J 2021) used deep learning methods to investigate genetic correlates of atrial structure and function in same UK Biobank dataset.

This yielded overlapping but non-identical genetic results, highlighting the variability that different deep learning methods can introduce. Since all the subsequent analyses in Pirruccello et al's paper rely on these atrial parameters, it would be useful to undertake a validation analysis eg. by comparing CMR data with echocardiographic data in a subset of individuals.

In the Results (line 112), 1015 participants with increased left atrial volumes were removed from the analysis. Additional information is required to justify this, since exclusion of these individuals could change the subsequent analysis between LAmin and incident AF. It cannot be assumed that these individuals had undiagnosed AF. Were there any other identifiable causes for left atrial dilation?

The GWAS study and LAmin-PRS are important and expand the spectrum of genetic variants that are associated with AF and stroke risk. If possible, it would be useful to validate these associations in a replication cohort.

It can be problematic to extrapolate disease mechanisms from GWAS loci due to uncertainty of the target genes. In the Discussion (line 274), it is proposed that left atrial dilation may be the phenotype associated with the PITX2 locus. How does this fit in to the extensive pre-existing literature that has studied this locus?

Table 1. Clinical characteristics of the study cohort. This table could be expanded to include presence/absence of AF as well as co-morbidities and cardiac features, eg underlying cardiomyopathy/valve defects/congenital heart defects, that could influence left atrial size.

Figure 1. The middle panel is too small and it is difficult to see the detail in these images. What do the schematics between the two columns of CMR images represent? The font size of the text on all panels could be increased.

Reviewer #3 (Remarks to the Author):

* Summary

This paper performed a GWAS analysis using deep learning-derived left atrium (LA) imaging phenotypes of ~40K subjects from the UK Biobank cohort. After identifying 20 loci, the authors performed further statistical analyses, demonstrated the correlation between LA phenotypes and atrial fibrillation (AF) risks and investigated the bi-directional causality using the Mendelian randomisation analysis.

* Major comments

My main concern is about the clarity of the computational methods for determining the 3D LA volumes and detecting abnormal patterns for LA contraction, as well the validation of these methods.

1. All statistical analyses were based on the 3D LA volume measurements, which were derived from separate 2D imaging views using the Poisson surface reconstruction method. However, I could not find the detailed description of this method in the paper or in references [55,56]. Could the paper elaborate this method (Line 403-405)? Given 2D LA chamber segmentations on different short-axis or long-axis views, how exactly is the Poisson surface reconstruction applied?

2. How is the Poisson surface reconstruction method validated? How accurate are the 3D LA meshes? Are they evaluated qualitatively or quantitatively?

3. Since the short-axis view may not always capture the LA, how does the surface reconstruction method cope if LA is missing in the short-axis view?

4. How are the LA volumes quality controlled?

5. Line 422-423: "A deep learning model was then trained to classify filling patterns as representing a normal atrial contraction or not." What do these filling patterns look like? Could the paper provide some illustrative examples for normal patterns and abnormal patterns?

* Other comments

6. Figure 1, 4Ch view

Is the LA a bit over-segmented or maybe not? Does the annotation follow a consistent manner to cut the upper boundary of the LA?

7. Line 446-447: "... adjusting for the MRI serial number, sex, age, and the interaction between sex and age."

What is the MRI serial number here please? UK Biobank runs three imaging centres each with a MRI scanner of the same model. Is the MRI serial number used to adjust the inter-site difference?

8. Line 450-451: "A Cox proportional hazards model was used, with survival defined as the time between MRI and either the time of censoring, or disease diagnosis."

How is the disease diagnosis information obtained? The LA volumes are derived at the first imaging visit of the UK Biobank participants. Are the disease diagnosis codes obtained from their second imaging visit? If that is the case, what is the average duration between the two?

Reviewer Comments

Reviewer #1 (Remarks to the Author):

Remarks to the Author

This is an interesting and important study by Pirruccello et al. Pirruccello et al performed a large MRI assessment of LA structure and function. They identified 20 common genetic variants associated with LA volumes or LAEF.

In line with these finding they found that a PRS of the minimal LA volume was associated with AF and interestingly also stroke. Pirruccello et el really deserves credit for all the work that have been put in to the paper in particular the development of the deep learning models to measure LA traits from cardiovascular magnetic resonance imaging. Do you plan to make the algorithm publicly available?

Author response

We are grateful to the Reviewer for their generous comments. We do plan to make both the deep learning models and surface reconstruction code public: the deep learning model weights will be returned via the UK Biobank, and we have already made the code to integrate the deep learning output to perform surface reconstruction publicly available on Github at the following URL: https://github.com/broadinstitute/ml4h/blob/pd_atria/scripts/mri_la_poisson.py .

Manuscript change

The **Code availability** section now states:

The deep learning models will be returned to the UK Biobank for use by other researchers. The code used to perform Poisson surface reconstruction from segmentation output is located at https://github.com/broadinstitute/ml4h/blob/pd_atria/scripts/mri_la_poisson.py and is available under an open-source BSD license.

Introduction

Minor:

- The sample size of the GWAS w/ bipolar estimates of LA could also be declared.

Author response

Thanks for the recommendation, we have now added this information.

Manuscript change

The **Introduction** now states:

... Recently, a genome-wide association study of deep learning-derived diastolic measurements in 34,245 UK Biobank participants identified one variant associated with LA volume near NPR3(Bai et al., 2020; Thanaj et al., 2021), and a genome-wide association study of a biplanar estimate of LA volume and function identified 14 unique loci in 35,658 participants(Ahlberg et al., 2021).

Methods

- Could you please provide some more details on the training/test set used to create the model that identify abnormal contraction patterns? How many samples were in the training set? Could you also provide some more metrics on the performance of this model, such as sensitivity and specificity?

Author response

Thanks for these questions and points raised. We have now incorporated this information into the Supplement. Although we did not originally compute performance metrics for the motion detection model, we have now done so based on review of 100 participants flagged as abnormal and 100 flagged as normal. We find that the model is highly sensitive but has low specificity, which is compatible with our desired use-case for it. (Namely, the high sensitivity [90.3-100%] helps us reduce the risk of including people who cryptically had arrhythmia into the GWAS, which would contaminate the Mendelian randomization analyses, while the low specificity [53.1-68.5%] simply reduces power.)

Manuscript change

The **Results>Reconstruction of LA volumes from cardiovascular magnetic resonance images** section now points to these findings in the supplement:

The quality of the deep learning models for measuring the LA was higher for the long axis views and lower for the short axis views, which were not designed to capture the LA (Supplementary Note).

The **Supplementary Note>Methods>Identification of abnormal cardiac filling patterns** section now states:

To evaluate the accuracy of the deep learning model, manual evaluation of the cardiac filling patterns was conducted by one cardiologist (JPP) for 100 participants flagged as having abnormal cardiac filling patterns and 100 flagged as having normal cardiac filling patterns, sampled at random from participants without a history of atrial fibrillation. Sensitivity and specificity and their confidence intervals were calculated with the `binom.test` function in R.

The **Supplementary Note>Results>Quality control for the deep learning model for abnormal cardiac filling patterns** section now states:

Among 200 participants whose MRIs were manually reviewed (100 flagged as having abnormal cardiac filling patterns and 100 flagged as having normal cardiac filling patterns), manual review determined that 164 were normal and 36 were abnormal. The sensitivity of the model for identifying abnormal cardiac filling patterns was 100% (95% CI 90.3-100.0%) and the specificity was 61% (95% CI 53.1-68.5%). These findings suggested that the model may have over-detected abnormal cardiac filling—leading to the exclusion of more participants than necessary—but had little evidence for false negatives.

- Could you evaluate the relationship to blood pressure?

Author response

We now describe the relationship between left atrial volumes and hypertension, also visually depicted in **Figure 3** and detailed in the **Supplementary Tables 1-3**. We now also add a phenome-wide association analysis between continuous/real-valued measurements in UK Biobank and the LA measurements in the **Supplementary Note**. Higher blood pressure is associated with larger LA volumes.

Manuscript change

The **Results>LA traits are associated with AF, heart failure, hypertension, and stroke** section now states:

We also observed significant associations between LA measurements and hypertension, heart failure, and stroke (Figure 3 and Supplementary Tables 1-3), as well as continuous

traits such as blood pressure, creatinine, and pack years of tobacco use (Supplementary Table 4).

The **Methods>Evaluation of the relationship between the left atrium, phenotypes, and cardiovascular diseases** section now begins with:

For epidemiologic analyses of continuous traits, we performed linear regression, with the left atrial phenotypes as the dependent variable in a model with the phenotype of interest adjusted for sex, the first five principal components of ancestry, the genotyping array, the MRI scanner, and a third-degree spline of age at the time of imaging to account for possible nonlinear effects of age.

- Is it necessary to adjust your cox models for height and weight when adjusting for BMI?

Author response

This is an interesting question. Our rationale for adjusting for BMI in addition to weight and height is that BMI captures a higher-order function of the height term as well as an interaction between height and weight (i.e., is not fully captured by the height and the weight terms separately). However, empirically testing your question in the UK Biobank, a formula can be built that largely captures BMI within the population range as a simple linear function of height and weight with $r^2=0.99$. (Formula: $BMI = 55.32 - 0.3292 * height_cm + 0.3532 * weight_kg$). Meanwhile, the reverse is not true: a linear model predicting height from BMI in the UK Biobank explains < 1% of the variance, while a model predicting weight from BMI explains 70% of the variance. Practically, within the UK Biobank, our interpretation is that one could use height and weight without BMI and expect to capture 99% of the variance that BMI would offer, whereas one could probably not use BMI as a proxy for both height and weight.

Nevertheless, when we tested this empirically, we found little effect of removing BMI from the Cox model: a 1-SD change in LA_{min} had an HR of 1.73, $P = 4.0E-39$ in the model with BMI, *versus* an HR of 1.73 and $P = 5.2E-39$ in the model without BMI. Therefore, we have not changed the main analyses or text in this case.

- It is not clear if the measurements of LA volume indexed? It might be more clinically relevant to use BSA indexed measurements.

Author response

The LA volume measurements are treated both in their raw form and in their BSA-indexed form. We think that our most unclear description of this was in the GWAS results section, so we have now tried to improve our wording there to make this more clear. Thank you for bringing this to our attention.

Manuscript change

The **Results>Common genetic variant analysis of LA size and function identifies 20 loci** section now clarifies the seven traits in the section's introduction:

After establishing that the LA measurements replicated previously established clinical associations, we then examined the association between common genetic variants and seven LA traits: LA_{max}, LA_{min}, LA_{EF}, and LA_{SV}, as well as for body surface area (BSA)-indexed LA volumes.

- It is true that mixed models are able to account for population stratification and relatedness. However, the publications for commonly used software like BOLT-LMM (used in this study) and fastGWA have evaluated the confounding of population stratification on a somewhat homogeneous population with European ancestry, i.e., they have removed ethnical outliers from their UKB population. It is also true that there aren't that many samples that would be filtered

from your analyses if you excluded ethnical outliers. Therefore, a sensitivity analyses would most likely show similar results. However, I would consider it problematic QC not to remove PC outliers from your GWAS analyses. Furthermore, it is unclear how not removing outliers could affect some downstream post-analyses. I would prefer to see PC outliers removed and if included it should be through a meta-analysis. This is after all done in almost all GWAS for a good reason.

Author response

To try to address these important points, we have taken several steps. First, we have added information about Hardy Weinberg equilibrium P-values to the Supplementary Tables, confirming a lack of evidence for genotyping error or population stratification at the lead GWAS variants.

Second, we have conducted a series of sensitivity analyses: (a) a EUR-specific GWAS, (b) a GWAS of participants with the same sample size as the EUR-specific GWAS (by randomly dropping participants), and (c) a series of GWAS of four genetic inlier groups followed by meta-analysis. For (a) and (b) we were able to use BOLT because the sample size was large enough, but the non-EUR components of (c) were below the BOLT sample size recommendations and therefore were analyzed with REGENIE before meta-analysis using METAL. Our overall impression across all of the sensitivity analyses was that there was little evidence that the GWAS loci were driven by population stratification.

One notable consequence of analyzing the genetic inlier groups separately, even though they were ultimately meta-analyzed in the end, is that a reasonably large fraction of the total participants did not contribute to any of the components of the meta-analysis. In this case, ~7.5% of the original GWAS sample was not part of any genetic inlier group and therefore left out of the meta-analysis. Avoiding this substantial sample size loss may be one of the

explanations for why, where feasible, mixed-model based multi-ancestry joint analysis has been noted to maximize power for discovery (Wojcik, et al, “Genetic analyses of diverse populations improves discovery for complex traits”, Nature 2019). That said, we would certainly agree that ancestry-specific GWAS with meta-analysis is a successful model and remains the best option in many cases, such as multi-cohort meta-analyses.

Manuscript change

The **Results>Common genetic variant analysis of LA size and function identifies 20 loci** section now includes:

No lead SNPs deviated from Hardy-Weinberg equilibrium (HWE) at a threshold of $P < 1E-06$ (Supplementary Table 7) (Chang et al., 2015).

...

Other sensitivity analyses (retaining participants with abnormal cardiac filling patterns; retaining only individuals with inlier genetic identities) are detailed in the Supplementary Note.

The **Supplementary Note>Methods>GWAS sensitivity analysis: genetic diversity** section now describes these analyses:

The primary analyses permitted the inclusion of all participants with LA measurements, regardless of genetic identity (Supplementary Figure 9). As a sensitivity analysis, individuals were analyzed within genetic inlier groups instead of jointly. To accomplish this, first self-reported ethnicity—which is only informally correlated with genetic identity—was aggregated into European (British, Irish, and Other European), African (African, “Any_other_Black_background”, “White_and_Black_African”, and “White_and_Black_Caribbean”), South Asian (Bangladeshi, Indian, Pakistani), and East

Asian. Individuals with self-reported ancestry of “Any_other_mixed_background”, “Mixed”, “White_and_Asian”, “Any_other_Asian_background”, “Caribbean”, “Do_not_know”, “Other_ethnic_group”, or “Prefer_not_to_answer” were not analyzed further. Then, for each group of participants, the R package aberrant was run on the centrally computed genetic principal components of ancestry using a 40 standard deviation window similar to the approach of Bycroft, et al (Bellenguez et al., 2012; Bycroft et al., 2018). Inliers for each genetic identity group were retained. Individuals that were not part of an inlier genetic identity group were excluded. The genetic identity inlier groups were termed EUR, AFR, SAS, and EAS.

The sample sizes for the AFR, SAS, and EAS subsets were all well below the threshold recommended for the use of BOLT-LMM (“We recommend BOLT-LMM for analyses of human genetic data sets containing more than 5,000 samples”, BOLT-LMM v2.4.1 User Manual https://alkesgroup.broadinstitute.org/BOLT-LMM/BOLT-LMM_manual.html). Therefore, for each of the four genetic inlier groups, a GWAS was conducted with REGENIE v2.2.4 which does not have the same limitation (Mbatchou et al., 2021). All models were adjusted for sex, age and age² at the time of MRI, the first 10 principal components of ancestry, the genotyping array, and the MRI scanner’s unique identifier. Fixed-effect meta-analysis was then conducted with METAL (release version 2020-05-05) (Willer, Li and Abecasis, 2010).

Two additional GWAS were conducted in BOLT-LMM v2.3.4 using the same covariates as the primary GWAS: one for the inlier EUR population, and another where an equivalent number of individuals were dropped at random from the original GWAS cohort (without regard for genetic inlier grouping) to yield a sample size that was the same as the inlier EUR population.

GWAS loci from the primary analysis were fetched from the meta-analysis, the EUR-specific GWAS, and the GWAS in which individuals were dropped at random.

The **Supplementary Note>Results>GWAS sensitivity analysis: genetic diversity** section now provides a brief description of the results:

Data from all participants were used for the primary GWAS, incorporating a diversity of genetic identities (Supplementary Figure 9). In a sensitivity analysis, only individuals with inlier genetic identities for one of four inlier groups were retained and analyzed separately (EUR, AFR, SAS, or EAS; Supplementary Figure 11). In this analysis, the largest inlier group was that for EUR, with 31,878 participants (9.9% smaller than the primary analysis). The second largest group was comprised of the 2,655 participants (7.6%) who were not genetic inliers for any group and were therefore not included in these sensitivity analyses. This was followed by SAS (N=284), AFR (N=133), and EAS (N=99), together comprising about 1.5% of the primary GWAS sample size. GWAS were separately conducted for EUR, SAS, AFR, and EAS, and then meta-analyzed. Because of the loss of the participants who were included in the joint analysis but were not inliers for any genetic identity group, the multi-ancestry meta-analytic approach represented a loss of 7.6% of the total sample size compared to the primary analysis. These meta-analytic P-values were fetched for the lead variants from the primary analysis and are displayed in Supplementary Table 7 as the “P_META” column.

Two additional sensitivity analyses were performed using BOLT-LMM: a EUR-specific GWAS, and an analysis in which individuals were dropped at random to achieve the same sample size as the EUR-specific GWAS. The P-values for the primary analysis's

lead variants are also displayed in Supplementary Table 7 with the “P_EUR” and “P_RANDOMDROP” columns, respectively.

The weakest association signal occurred for the BSA-indexed LAmin phenotype in the multi-ancestry meta-analysis at the GOSR2 locus ($P=2.5E-06$), which was an order of magnitude weaker than the evidence for the EUR subgroup without meta-analysis ($P=2.0E-07$). Nevertheless, across these sensitivity analyses, we largely observed minor variation in association signal without clear evidence for population stratification.

- Please provide principal component plots on the GWAS cohort.

Author response

We have now added a PCA plot of the GWAS participants to the **Supplementary Note**. Given the updated multi-ancestry meta-analytic sensitivity analysis, we have also provided a version of the chart that is faceted by genetic inlier group.

Manuscript change

These figures have been added to the **Supplementary Note**:

Supplementary Figure 9:

Principal components of ancestry for the GWAS participants, as well as participants' self-described ethnicity mapped with color.

Supplementary Figure 11:

Principal components of ancestry for the GWAS participants, as well as participants' self-described ethnicity mapped with color. Each genetic inlier group is split into its own facet. The participants that were not part of any genetic inlier group are labeled "None".

- Would it be possible to add a histogram showing the distribution of the phenotypes for each trait? Were the traits directly or indirectly rank-transformed?

Author response

Thanks for the suggestion, we have now added these plots. The rank transformation in this case was direct; that is, the transformation was performed on the trait itself rather than on, e.g., residuals.

Manuscript change

This figure has been added to the **Supplementary Note**:

Supplementary Figure 1:

Trait distributions for the left atrial phenotypes without adjustment and after adjustment for body surface area (BSA). LAEF is dimensionless and is therefore not adjusted for BSA.

- Could you please motivate the reason for adjusting both for age at enrollment and age at the time of MRI?

Author response

Our rationale was that including both ages provides information about immortal time (which may be relevant because the participants underwent MRI after enrollment, and the selection for MRI was not a prospective decision at the time of enrollment). For a disease-based GWAS we think this type of information would be critical, but the effect of immortal time on a continuous-trait GWAS, we would expect, should be very small and most likely negligible.

- It is stated in the method section that you have conducted 10 GWAS's. I understand them to be LAEF, LASV, LAmin, LAmx, BSA indexed LASV, BSA indexed LAmin, BSA index LAmx, LVEDV indexed LASV, LVEDV indexed LAmin, LVEDV indexed LAmx. Why are the LVEDV indexed GWAS's reported as a sensitivity analyses and the BSA indexed GWAS's reported in the main analyses? In many of the post analyses only non-indexed phenotypes are reported. This becomes a bit confusing for the reader and it is unclear what traits you actually have studied. Could you please explain as to why you have made these decisions for your study design?

Author response

The distinction (between treating BSA-indexed traits as primary analyses, but LV-indexed traits as secondary analyses) comes from the common clinical use of body surface area adjustment. We hypothesized that adjustment for left ventricular volume could provide an even more precise way of accounting for what the left atrial size "should" be, but it comes with several caveats and we therefore consider it a secondary analysis. First and our primary reason for considering BSA-indexing a main analysis and LV-indexing as a sensitivity analysis: LV adjustment of LA phenotypes can certainly be done but is not routinely performed in clinical settings (as opposed to BSA adjustment which is the norm). Second, genetic variation that influences both LV and LA

phenotypes could be nullified by this approach, though we also thought it was one of the interesting aspects of the analysis, since it could emphasize LA-specific loci. However, third, introducing heritable covariates can also introduce spurious correlation signals so their interpretation requires caution. (For example, a variant that has no effect on the LA phenotype but which increases the LV phenotype would have an induced negative association; with close inspection, this can be discerned, but as Aschard, *et al*, Am J Hum Gene 2015 point out, there is not a clear automated way to do this.)

Manuscript change

In the **Methods>Genome-wide association study of the left atrium** section, we now link to the supplementary note with rationale:

We analyzed the four unadjusted LA phenotypes, as well as LAm_{ax}, LAm_{in}, and LASV estimates that were adjusted for BSA or LVEDV (rationale detailed in the Supplementary Note).

The **Supplementary Note>Methods>GWAS sensitivity analysis - LVEDV-indexing** section now states:

A sensitivity analysis was conducted to assess the consequence of accounting for body size based on each individual's LVEDV (rather than BSA). In addition to functioning as a sensitivity analysis for that purpose, accounting for left ventricular volume could, in principle, help to identify loci whose effects have the opposite effect direction between atrium and ventricle. However, adjusting for heritable covariates in GWAS can also induce associations via collider bias(Aschard et al., 2015). Like the primary analyses, the LVEDV-indexed sensitivity analyses were conducted with BOLT-LMM with the same covariates and settings (Online Methods). To attempt to identify LVEDV-indexed associations that were likely attributable to the adjustment for LVEDV, we also

conducted a GWAS of LVEDV in the same participants with the same settings, and then tested each of the LVEDV-indexed lead SNPs for independent association with LVEDV.

The **Supplementary Note>Results>GWAS sensitivity analysis - LVEDV-indexing** section now states:

We are not aware of a general solution to the interpretation of GWAS signals that incorporate adjustment for heritable covariates. However, we observed the LVEDV-indexed lead SNPs to fall into three patterns: first, some SNP associations appeared to be driven largely by the LVEDV indexing rather than LA volume. As an example of this pattern, the LVEDV-indexed LAm_{ax} association with BAG3 ($P=3.5E-10$) was comparable to that for the LVEDV association with BAG3 alone ($P=2.1E-10$), while the unadjusted LAm_{ax} measurement was not associated ($P > 1E-3$). At each of these loci, the effect direction in LVEDV was opposite to that in the respective LVEDV-indexed LA volume GWAS, which was expected. Practically, these signals appeared to be driven by the LVEDV values, with the LA measurements acting as noise. Second, some SNP associations appeared to be driven by the LAm_{ax} association alone, with only minimal contribution from the LVEDV adjustment. For example, the LVEDV-indexed LAm_{ax} association with IRAK1BP1 ($P=2.0E-8$) was similar to that for the LAm_{ax} association ($P=2.7E-11$), while the SNP was not associated with LVEDV ($P > 1E-3$). Third, some SNP associations appeared to be driven by the interplay between LA volumes and the LVEDV adjustment. For example, the NEDD4L locus was associated with LVEDV-indexed LAm_{ax} ($P=4.7E-8$) despite not being strongly associated with either LVEDV or LAm_{ax} alone ($P > 1E-3$ for both).

For the LVEDV-indexed LA volumes, 11 loci reached genome-wide significance for LAm_{ax}, 12 for LAm_{in}, and four for LASV. Of these, six of the LVEDV-indexed LAm_{ax}

loci had association $P < 1E-3$ with LVEDV, as did nine of the LAmin loci and two of the LASV loci. Novel loci that were not associated at genome-wide significance in the unadjusted GWAS, and which were not associated with LVEDV at $1E-3$ or stronger, included BLK, ANKRD1, MYH7, and NEDD4L for LAm_{ax}; CASQ2, DHX15, PROB1, UQCRB, ANKRD1, and MYH7 for LAmin, and TNKS and HNRNPM for LASV. Most of these loci were identified in the BSA-indexed GWAS as well.

- In regards to overlap with atrial fibrillation loci. Would it not be more meaningful to see if the actual loci overlap instead of nearest gene?

Author response

This is a good point. We have now updated our analysis to do just that (rather than testing via the nearest gene as an intermediary link). The same 8 loci are observed to be in proximity. Additionally, this reduced the number of randomly permuted control SNP sets with at least 8 overlapping loci from one to zero; the greatest number of overlapping loci by chance across 10,000 permutations was reduced to five. This trivially increased the significance of the result (now permutation $P=1E-04$ instead of $2E-04$). We have updated the manuscript to reflect this updated analysis.

Manuscript change

The **Results>Genetic relationship between AF risk and LA dysfunction** section has been updated (excerpt):

We then assessed the overlap between the 20 distinct LA loci identified in our study and 134 loci previously found to be associated with AF (Roselli et al., 2018). We found that 8 of the 20 LA loci overlapped with an AF locus, which was a significant enrichment based

on permutation testing ($P=1E-04$, which was the minimum possible P value; see Methods)(Pers, Timshel and Hirschhorn, 2015).

The **Methods>Overlap of left atrial loci with atrial fibrillation loci** section has been updated as follows:

We identified the lead SNPs associated with AF from Supplementary Table 16 of Roselli, et al(Roselli et al., 2018). For this exercise, we used each of the 134 SNPs that achieved association $P < 5E-8$ in the primary GWAS (column 'I') or in the meta-analysis (column 'AD'). We counted the number of AF lead SNPs that fell within 500kb of the LA lead SNP from our study. We used SNPsnap to generate 10,000 sets of SNPs that matched the LA lead SNPs based on parameters including minor allele frequency, SNPs in linkage disequilibrium, distance from the nearest gene, and gene density(Pers, Timshel and Hirschhorn, 2015). We then repeated the same counting procedure for each of the 10,000 synthetic SNPsnap lead SNP lists, to set a neutral expectation for the number of overlapping AF lead SNPs based on chance. This allowed us to compute a one-tailed permutation P value (with the most extreme possible P value based on 10,000 randomly chosen sets of SNPs being $1E-04$).

- In your MR analysis, why did you choose a P -value threshold of $1xE-6$? Could you provide a sensitivity analysis using only variants with $P < 1xE-8$?

Author response

We originally reasoned that estimates derived from independent SNPs at $P<1E-06$ would be more stable than $P<5E-08$ because of the small number of genome-wide significant loci—especially as we would be pruning away pleiotropic loci for our sensitivity analysis. To address this query, we have now also analyzed the data when restricted to just the $P<5E-08$ threshold.

We performed this for all LA measurements (as mentioned in another point from the Reviewer further down).

Manuscript change

The **Methods>Mendelian randomization** section now also states:

Additional Mendelian randomization analyses were conducted using each LA measurement as an exposure constructed from SNPs with $P < 5E-08$, tested against AF(Christophersen et al., 2017), heart failure from HERMES(Shah et al., 2020), and the trans-ancestry ischemic and cardioembolic stroke summary statistics from MEGASTROKE(Malik et al., 2018).

The **Results>Causal link between LA minimal volume and disease risk** section now states:

Analyses treating each LA measurement as an exposure, using only instruments with $P < 5E-08$, revealed that the strongest statistical relationship was between LAEF and AF (OR 0.36 per SD increase in LAEF, $P = 1.6E-06$; Supplementary Table 11).

- Did you test for heterogeneity in the TwoSampleMR package?

Author response

We have now run the Cochran Q heterogeneity tests and have included them in the results.

There was significant heterogeneity across many of the tests. As a consequence, we have now also incorporated a contamination mixture model analysis; this approach performs robust Mendelian randomization in the presence of invalid instruments. The findings remain similar.

Manuscript change

The **Methods>Mendelian randomization** section now also states:

Heterogeneity was tested with Cochran Q(Cochran et al., 2023). Because of effect heterogeneity, the contamination mixture model approach—which performs robust Mendelian randomization in the presence of invalid instruments—was also employed(Burgess et al., 2020).

The **Results>Causal link between LA minimal volume and disease risk** section now describes:

There was significant effect heterogeneity ($P=2.9E-05$ by Cochran Q), so the contamination mixture model approach and MR-PRESSO were applied, both of which showed a significant, positive relationship between LAmin and AF with the same direction of effects (Supplementary Table 10; Supplementary Figure 5).

- In your MR analysis, could you test the other LA phenotypes as well?

Author response

We have now calculated and incorporated MR findings using the other LA phenotypes as exposures. For some diseases, the LAEF instruments yielded stronger results as described below and in the manuscript.

Manuscript change

The **Methods>Mendelian randomization** section now states:

Additional Mendelian randomization analyses were conducted using each LA measurement as an exposure constructed from SNPs with $P<5E-08$ (...)

The **Results>Causal link between LA minimal volume and disease risk** section now describes the additional LA phenotypes.

Analyses treating each LA measurement as an exposure, using only instruments with $P < 5E-08$, revealed that the strongest statistical relationship was between LAEF and AF (OR 0.36 per SD increase in LAEF, $P = 1.6E-06$; Supplementary Table 11).

It would also be interesting to use additional exposures/outcomes such as blood pressure, heart failure and stroke.

Author response

We have now added MR findings that test all LA measurements against additional outcomes: heart failure and two stroke subtypes (all ischemic stroke and cardioembolic stroke). With varying degrees of statistical significance, the overall findings are concordant with the observational and polygenic analyses.

Manuscript change

The **Methods>Mendelian randomization** section now states:

Additional Mendelian randomization analyses were conducted using each LA measurement as an exposure constructed from SNPs with $P < 5E-08$, tested against AF(Christophersen et al., 2017), heart failure from HERMES(Shah et al., 2020), and the trans-ancestry ischemic and cardioembolic stroke summary statistics from MEGASTROKE(Malik et al., 2018).

The **Results>Causal link between LA minimal volume and disease risk** section now describes these additional outcomes.

Expanding the tested outcomes to heart failure(Shah et al., 2020) and stroke(Malik et al., 2018) revealed a nominal relationship between greater LAmin and increased risk for heart failure (OR 1.23 per SD increase in LAmin, $P = 0.03$), and between greater LAEF

and reduced risk for cardioembolic stroke (OR 0.56 per SD increase in LAEF, $P=5.3E-03$) but not all ischemic stroke ($P=0.5$; Supplementary Table 11).

- In the PRSs analyses, are the prior Beta values used on the ranked-based inverse transformation scale?

Author response

They are indeed similar to that but slightly different. The PRSs effect estimates are in terms of per standard deviation of the polygenic score. This is because PRSs does not use the effect estimate; rather, it computes a standardized effect from P values. Because their absolute values are therefore not particularly meaningful, we put them onto the standardized scale by subtracting the population mean and dividing by the population standard deviation to produce effect estimates in terms of a standard deviation change.

Manuscript change

We now describe the effects as being on a per-standard-deviation basis in the **Results>A polygenic estimate of LA volume predicts AF, stroke, and heart failure** section as follows:

The strongest association was with the BSA-indexed LAmin polygenic score, which was linked to a modestly increased risk for incident AF or atrial flutter (HR=1.09 per 1 SD increase in the score; $P=7.4E-32$) (Figure 6; Supplementary Table 13). This score was also associated with small increases in risks of incident all-cause stroke (7,753 cases; HR=1.04 per SD; $P=4.7E-04$), ischemic stroke (5,444 cases; HR=1.04 per SD; $P=4.7E-03$), and heart failure (11,035 cases; HR=1.05 per SD; $P=7.9E-08$).

- Using a UK biobank European ancestry linkage disequilibrium panel, you use PRSs to calculate a PRS. And then, project the PRS onto the entire UK Biobank with the exception of individuals related to individuals with cardiac MRI. Do you exclude individuals with cardiac MRI

in the COX regression, that is: your GWAS sample set, this is not clear to me? Since you are using a European ancestry linkage disequilibrium panel, would it not be more appropriate to perform this post-analysis on individuals with European ancestry only?

Author response

Thanks, we have now clarified that we exclude people with MRI or related within 3 degrees. (In other words, the GWAS sample set, and anyone related to them within 3 degrees, is excluded from the analysis.)

Regarding the choice of LD panel, the PRSs authors recommend that regardless of the target sample ancestry, the LD panel should align with the GWAS sample (see github.com/getian107/PRSs/issues/59 for some discussion). While the GWAS population was not 100% European as you noted above, it was 97% European and we felt that this LD panel was reasonable in this case.

Regarding genetic identity, we expect that the PRS will be less effective for participants with non-European genetic identities, since due to allele frequency differences it will be misspecified to some extent. Therefore, we felt that applying it to all participants was actually the most conservative approach which would avoid making the PRS seem more powerful than it is. We have now conducted a sensitivity analysis in participants with European genetic identities to answer this empirically. These are now reported in **Supplementary Table 14**. For example, when analyzing only European (non-MRI and non-MRI-related) participants, the HR for a one standard deviation greater polygenic score for BSA-indexed LAmin volume is 1.09 ($P=1.3E-28$), which is similar to the result without ancestry restriction (HR 1.09, $P=7.4E-32$).

Manuscript change

For the in-sample PCA adjustment, the **Supplementary Note>Methods>Internal validation of LA polygenic scores in non-imaging participants** section now clarifies:

The LA polygenic scores were applied to the entire UK Biobank. Participants who had undergone MRI or related within 3 degrees of kinship to those who had undergone MRI, based on the precomputed relatedness matrix from the UK Biobank, were excluded from analysis (Bycroft et al., 2018). We analyzed the relationship between this polygenic prediction of each LA measurement and incident disease (defined by self-report and diagnostic and procedural codes) in the UK Biobank using a Cox proportional hazards model as implemented by the R survival package (Therneau and Grambsch, 2000). The primary disease analyzed was atrial fibrillation. For each tested disease, we excluded participants with disease that was diagnosed prior to enrollment in the UK Biobank. We counted survival as the number of years between enrollment and disease diagnosis (for those with disease) or until death, loss to follow-up, or end of follow-up time (for those without disease).

We adjusted for covariates including sex, the cubic basis spline of age at enrollment, the interaction between the cubic basis spline of age at enrollment and sex, the genotyping array, the first five principal components of ancestry, and the cubic basis splines of height (cm), weight (kg), BMI (kg/m²), diastolic blood pressure (mmHg), and systolic blood pressure (mmHg). Sensitivity analyses included restriction participants to a genetic inlier population with European genetic identity (precomputed by the UK Biobank); adjusting for genetic principal components derived from the GWAS samples instead of the entire cohort; adjusting only for age and sex; applying score weights derived from the clumped lead variants with $P < 5E-08$ from each trait instead of PRSCs; and thresholding the cohort into the top 5% for each polygenic score compared to the bottom 95% for the score.

These sensitivity analyses are in the supplementary tables and are cited in the main text in the **Results>Polygenic estimates of LA volume predict AF, stroke, and heart failure** section:

Sensitivity analyses using lead SNP scores, different covariate adjustments, or different population subgroups yielded similar results (Supplementary Table 14).

- Which PCs are you using? As I understand, when using PRScs-auto you do not need a validation set for tuning parameters but you still need an independent test set for evaluation. My concern here is that you have used the PCs given by UKBB, which might have been generated on the whole UK Biobank dataset. In that case, there is a risk of information leakage when you project your PRS onto the rest of UK Biobank. Because, you have previously adjusted the generated GWAS summary statistics, used as prior for PRScs, on those PCs and the “UK biobank European ancestry linkage disequilibrium panel” is also generated using the same samples. Could the PRS model be projected on to an independent dataset with AF patients to see if incident AF is associated with LAmin PRS or alternatively another approach that would avoid this problem?

Author response

Thanks, you are correct and this is an important point. In our main PRS analysis, we are using the genetic PCs that were centrally computed by UK Biobank, which opens up the possibility of information leakage. We have one minor response and one more substantive response.

The minor response: We have added a sensitivity analysis that computes a new set of genetic PCs in only the GWAS derivation group and applies that to the rest of the UK Biobank.

However, because the UK Biobank PCs were used during the GWAS, and those PCs contain some information about the rest of the participants, this doesn't fully address the issue.

More substantively, to fully eliminate the possibility of false association of the polygenic score due to overlapping samples, we have now conducted replication in two external biobanks (*All of Us* and FinnGen). FinnGen, in particular, is sufficiently powered to replicate the positive relationships that we originally observed in UK Biobank between a greater value for the BSA-indexed LAmin polygenic score and higher risk of AF, heart failure, and stroke. Compared to FinnGen, *All of Us* has ~24% the case count for AF, 41% for heart failure, and only 0.5% as many ischemic stroke cases. Nevertheless, the AF and heart failure observations are also replicated in *All of Us*.

Because 680 *All of Us* participants also had BSA-indexed LAmin measurements, we were also able to show that the polygenic score from UK Biobank is correlated with its corresponding measurement in *All of Us*. (As a minor point, we note that currently this is the *only* volumetric LA measurement available in *All of Us*.)

Manuscript change

For the in-sample PCA adjustment, **Supplementary Table 14** now shows the full set of tests using PCA derived from the GWAS cohort instead of the centrally computed PCs.

The **Supplementary Note>Methods>Internal validation of LA polygenic scores in non-imaging participants** section now states:

Sensitivity analyses included restriction participants to a genetic inlier population with European genetic identity (precomputed by the UK Biobank); adjusting for genetic principal components derived from the GWAS samples instead of the entire cohort; adjusting only for age and sex; applying score weights derived from the clumped lead variants with $P < 5E-08$ from each trait instead of PRSCs; and thresholding the cohort into the top 5% for each polygenic score compared to the bottom 95% for the score.

For the external validation of the polygenic score in FinnGen, the **Supplementary Note>Methods>External validation of the BSA-indexed LAmin polygenic score in FinnGen** mentions:

...

PRS weights were applied using PLINK v1.9(Purcell et al., 2007; Chang et al., 2015). Case and control statuses for atrial fibrillation or flutter, ischemic stroke excluding subarachnoid hemorrhage, ischemic stroke excluding all hemorrhages and heart failure were defined based on events in the hospital, cause of death, specialist outpatient, primary care, and medication reimbursement registries at any point during registry follow-up as detailed in Supplementary Table 15. The association of PRS with each outcome was assessed using Cox proportional hazards models with follow-up time scale using sex, baseline age, baseline age squared, 5 genomic principal components, and the genotyping array as fixed-effects covariates.

For *All of Us*, the **Supplementary Note>Methods>External validation of the BSA-indexed LAmin polygenic score in All of Us** section mentions:

...

PRScs-based polygenic score weights from the UK Biobank were lifted over from GRCh37 to GRCh38(Hinrichs et al., 2006). Polygenic scores were then applied to all participants with WGS as an allelic sum, with an average taken over all of the weights. The UK Biobank GWAS in-sample PCA loadings were applied to the All of Us participants in the same way. These were then tested for association with the presence or absence of disease at any point prior to enrollment or during follow-up in a logistic regression model after adjustment for age at enrollment, whether the individual's self-reported sex was male, and the first five principal components of ancestry. Similarly, the

association with incident disease was tested with a Cox model with the same covariate adjustments after excluding individuals with disease prior to enrollment. All individuals with available data were analyzed. Sensitivity analyses examining only individuals with the “EUR” ancestry label were also conducted.

...

In the main text, the **Results>External validation of the LAmin polygenic score in FinnGen and All of Us** section is new:

In FinnGen(Kurki et al., 2022) study participants (Supplementary Table 15), comparable associations were observed for association between the BSA-indexed LAmin polygenic score and incident AF or atrial flutter (20,422 cases, HR=1.08 per SD, $P=2.4E-30$), ischemic stroke excluding subarachnoid hemorrhage (13,392 cases, HR=1.03 per SD, $P=3.0E-03$), ischemic stroke excluding all hemorrhage (11,822 cases, HR=1.03 per SD, $P=5.6E-04$), and heart failure (13,771 cases, HR=1.04 per SD, $P=4.4E-06$). Compared with the remaining 95% of FinnGen participants, those in the top 5% of genetically predicted LAmin-indexed had an increased risk of AF (HR=1.19 per SD, $P=8.4E-09$). Those in the top 5% also had elevations in risk that were not statistically significant for ischemic stroke excluding subarachnoid hemorrhages (HR=1.04 per SD, $P=0.36$) and heart failure (HR=1.07, $P=0.08$).

In the US national biobank, All of Us(Denny et al., 2019), the BSA-indexed LAmin polygenic score remained significantly associated with AF (4,859 incident cases, HR=1.06 per SD, $P=1.7E-04$) and heart failure (5,712 incident cases, HR=1.04 per SD, $P=2.0E-02$), but not ischemic stroke (66 cases, $P=0.3$; Supplementary Table 16). In logistic models that included all cases regardless of biobank enrollment date, more

cases were identified and the statistical evidence was stronger (13,399 AF cases, OR=1.10 per SD, P=4.9E-19; 14,572 heart failure cases, OR=1.04 per SD, P=1.5E-04).

In addition, 680 participants in All of Us with genetic data had BSA-indexed LAmin volume measurements. The BSA-indexed LAmin polygenic score was associated with these measurements (0.10 SD per SD of the polygenic score, P=8.5E-03). This relationship remained nominally significant when restricted to only the largest subset of participants by genetic identity (N=619 participants with genetic identity similar to Europeans; 0.09 SD per SD, P=1.5E-2).

- PRScs-auto require a large dataset for good performance. Is the sample size large enough? Could this be benchmarked by comparing performance to a P+T PRS approach by only selecting independent lead SNPs or similar?

Author response

This is a good point. The way that I typically think about sample size for PRScs is in terms of the effective sample size. For continuous traits this is just the sample size, but for binary traits this is twice the harmonic mean of cases and controls (Willer, Li, Abecasis, *Bioinformatics*, 2010).

Under this framework, many large binary-trait GWAS have an effective sample size that is not much larger than that of the present study. So on the one hand, the effective sample size here is well within the range of typical GWAS (even many seemingly large case-control GWAS, because of case-control imbalance). But on the other hand, the PRScs assumptions really hold for ~infinite sample sizes, which are certainly not satisfied here.

So, we have now applied the lead SNP PRS approach (namely, P+T at the GWAS threshold) and tested it within the non-MRI subset of the UK Biobank. For LA measurements with many GWAS loci, this approach works quite well. E.g., we observe an HR = 1.08 for the lead SNP PRS for BSA-indexed LAmin, which is very close to that of the PRScs estimate (HR = 1.09).

However, for traits with fewer loci, this approach does not work well. For example, for LAm_{max}, the PRScs estimate is HR=1.06, P=2.5E-16, whereas the lead SNP score estimate is HR=1.0, P=0.5.

We now include the lead SNP-based scores in the supplement. Our assessment is that for most tested outcomes, the PRScs scores were more strongly associated than the lead SNP scores.

Manuscript change

The **Supplementary Material>Methods>Polygenic score sensitivity analyses** section now states:

In addition to the primary LA polygenic scores produced with PRScs, an additional set of LA polygenic scores was created as a weighted allelic sum based on the lead variants for each trait. That is, for each tested participant, at each of the lead variant alleles, the number of effect alleles possessed by the participant was multiplied by the effect estimate; these were then summed for all alleles for each phenotype. They were tested for association with diseases in the same way as the PRScs scores.

The **Results>Polygenic estimates of LA volume predict AF, stroke, and heart failure** section now states:

Sensitivity analyses using lead SNP scores, different covariate adjustments, or different population subgroups yielded similar results (Supplementary Table 14).

- The methods section says that a PRS on LAm_{in} was generated, whereas Figure 6 says PRS for BSA-indexed LAm_{in}. Which one is correct? Why was only LAm_{in} used for PRS? It would be interesting to see a COX analyses with PRS on other LA phenotypes, in particular LAEF and BSA indexed LA.

Author response

Thanks, this is helpful feedback. We have now corrected our errant wording. Based on the other comments about sensitivity analyses and other LA measurements in this response document, we have tested six LA polygenic scores: LAm_{max}, LAm_{in}, LASV, LAEF, LAm_{max}-BSA-indexed, and LAm_{in}-BSA-indexed. While our primary focus remains the BSA-indexed LAm_{in} PRScs score, the wording in the Methods and Results now reflects the larger diversity of scores as a result of these revisions.

Manuscript change

The **Methods>Polygenic risk analysis of LA measurement genetic scores** section now states:

A polygenic score for each LA GWAS was computed using PRScs with a UK Biobank European ancestry linkage disequilibrium panel (Ge et al., 2019). This method applies a continuous shrinkage prior to the SNP weights.

The **Figure 6** caption now reads:

Disease incidence curves for the 417,881 participants who were unrelated to within 3 degrees of the participants who underwent MRI in the UK Biobank. Those in the top 5% for the BSA-indexed LAm_{in} PRS are depicted in red; the remaining 95% are in gray. X-axis: years since enrollment in the UK Biobank. Y-axis: cumulative incidence of AF (19,875 cases in the bottom 95% and 1,272 cases in the top 5%). Those in the top 5% of genetically predicted LAm_{in}-indexed had an increased risk of AF (Cox HR 1.19, P=7.9E-10) compared with those in the remaining 95% in up to 12 years of follow-up time after UK Biobank enrollment.

Minor:

- Hypertension is not mentioned as an outcome in Methods but is shown in Figure 3.

Hypertension is also not defined in ST12.

Author response

Thanks, we have now tried to clean up this wording and we have added the hypertension definition to the supplementary table.

Manuscript change

In **Methods>Evaluation of the relationship between the left atrium, phenotypes, and cardiovascular diseases** we now explain:

For the disease-based analyses, we focused on four disease definitions related to LA structure and function: AF or flutter, ischemic stroke, hypertension, and heart failure (defined below).

And in **Methods>Definitions of diseases and medications** we now explain:

We defined disease status based on self report, ICD codes, death records, and procedural codes from the UK Biobank's hospital episode statistics data (Supplementary Table 17).

Supplementary Table 17 now includes our definition of hypertension.

- In the section Genome-wide association study of the left atrium, there is no mention LAEF as a phenotype.

Author response

The four LAEF loci are briefly mentioned in the GWAS Results section and addressed more substantially in the “Genetic relationship between AF risk and LA dysfunction” section of the results and the Discussion.

Manuscript change

The **Results>Common genetic variant analysis of LA size and function identifies 20 loci** section mentions:

The four LAEF loci were located near FAF1, CASQ2, MYH6, and MYO18B.

The **Results>Genetic relationship between AF risk and LA dysfunction** section mentions:

The 8 loci found in both the LA GWAS and the AF GWAS are nearest to FAF1/C1orf85, CASQ2, TTN, PITX2, MYH6/MYH7, IGF1R, GOSR2, and MYO18B. At all 8 loci, the effect of each SNP on AF risk was in opposition to its effect on LAEF, and in most cases the effect of each SNP on AF was concordant with its effect on LAmin (Figure 5).

The **Discussion** mentions:

...

At all eight loci, the allele associated with increased AF risk was directionally associated with a lower LAEF, and generally with greater LA volumes (Figure 5). The opposed effect directions of these SNPs for AF risk and LAEF may be consistent with the concept of atrial cardiomyopathy (Goette et al., 2017).

As an example of the pattern of opposed SNP effects on LAEF and AF risk, we identified a missense variant within CASQ2 (rs4074536; p.Thr66Ala) as a lead SNP for LAEF on chromosome 1. The T allele of this SNP (encoding Thr66) corresponds with a reduced LAEF in our GWAS, and with reduced expression of CASQ2 in the right atrial

appendage and left ventricle in GTEx(Lonsdale et al., 2013). This variant is also in LD ($r^2=1.0$) in non-African 1KG populations for the AF lead SNP rs4484922(Machiela and Chanock, 2015; Roselli et al., 2018). In the study by Roselli and colleagues, the rs4484922-G allele is associated with an increased risk for AF; notably, that risk-increasing allele corresponds to the LAEF-reducing T allele of rs4074536. The rs4074536-T allele has also previously been associated with a longer QRS complex duration(Sotoodehnia et al., 2010; Prins et al., 2018). CASQ2 encodes calsequestrin 2, which resides in the sarcoplasmic reticulum in abundance and binds to calcium ions during the cardiac cycle. Missense variants in this gene have also been associated with catecholamine-induced polymorphic ventricular tachycardia, typically following a recessive inheritance pattern(Lahat et al., 2001, p. 2; Ng Kevin et al., 2020, p. 2).

Even among LA-associated loci that were not previously associated with AF, several showed the same consistent pattern of inverse effect between AF risk and LAEF (e.g., near NPR3, SSSCA1, and HMGA2). However, this pattern did not uniformly hold. For example, at the gene-dense locus near FBXO46/DMWD/RPSH6A, the LA volume-increasing (and LAEF-decreasing) variants were weakly associated with decreased AF risk.

- What are the QC steps for the 714,577 genotyped SNPs used as model SNPs to create the GRM?

Author response

Thanks, we've now added these details. In brief, we removed participants with sample genotyping missingness $\geq 2\%$, and we removed genetic variants with minor allele frequency ≤ 0.001 or with maximum genotype missingness greater than 5%.

Manuscript change

The **Methods>Genome-wide association study of the left atrium** section now states:

Genome-wide association studies for each phenotype were conducted using BOLT-LMM version 2.3.4 to account for cryptic population structure and sample relatedness(Loh et al., 2015, 2018). We used the full autosomal panel of 714,577 directly genotyped SNPs that passed quality control (minor allele frequency ≥ 0.001 ; maximum genotype missingness $\leq 5\%$ for each variant; maximum sample missingness $\leq 2\%$) to construct the genetic relationship matrix (GRM), with covariate adjustment as noted above.

- Please explain in more detail why you have chosen a 5 megabase window and a LD threshold of 0.001, since this is uncommon.

Author response

Thanks for noticing this. There is an important distinction between (a) clumping to find proxy SNPs, and (b) clumping to find independent signals. This distinction is important because what is conservative for one task is anti-conservative for the other. Our cutoffs here are more conservative *for the purpose of finding independent signals*, which was our goal. With large, dense imputation panels having tens of millions of SNPs, there are often numerous variants that are clearly in a shared LD block but which have a modest r^2 with the lead SNP. Further, LD structure spanning well beyond the common 1 megabase window is known to exist (Yang, ..., Visscher, Nat Genet 2012). Therefore, we attempted to aggressively clump these into the primary signals to avoid making secondary discoveries that are in fact merely variants in weaker LD with strong hits. Allowing SNPs with a higher r^2 to be treated as independent signals, or clumping in a narrower window, would yield an equal or greater number of correlated genetic instruments, so our more stringent cutoffs that (correctly, in our opinion) do not treat these as independent are *conservative*. Granted, for these phenotypes this is probably less critical because our strongest P values are still modest, but we think this is a good practice regardless.

Manuscript change

The **Methods>Mendelian randomization** section now explains:

The variants from the exposure summary statistics were clumped with $P < 1E-06$, $r^2 < 0.001$, and a radius of 5 megabases using the TwoSampleMR package in R(Hemani et al., 2018). These stringent clumping thresholds were intended to reduce the risk of including modestly correlated variants as if they were truly distinct instruments despite tagging the same underlying signal (e.g., having an r^2 0.1 with one another).

- I suggest that instead of writing “a commonly used threshold” to give a reference.

Author response

Thanks, we have now done so.

Manuscript change

The **Methods>Genome-wide association study of the left atrium** section now states:

Variants with association $P < 5 \cdot 10^{-8}$ were considered to be genome-wide significant(Risch and Merikangas, 1996).

- In the PRS COX-model, is it necessary to adjust for height, weight and BMI? In particular if you have used BSA-indexed LAmin?

Author response

Thanks, we have now tested this out. Our original goal was to avoid a scenario where the body mass or body surface component of the score was driving the association, rather than the left atrial component of the score. We expected that including each person's actual body size would weaken the apparent effect of the polygenic score, i.e., would be conservative. We have now

added a sensitivity analysis that adjusts only for age and sex. Compared to the main fully adjusted analysis, this led to trivial changes in the effect estimates (e.g., LA_{min}_indexed PRS risk for AF went from HR=1.09 and P=7.4E-32 with full adjustment to HR=1.09, P=1.2E-38 with adjustment for only age and sex).

Manuscript change

The **Supplementary Note>Methods>Internal validation of LA polygenic scores in non-imaging participants** section now states:

Sensitivity analyses included restriction participants to a genetic inlier population with European genetic identity (precomputed by the UK Biobank); adjusting for genetic principal components derived from the GWAS samples instead of the entire cohort; adjusting only for age and sex; applying score weights derived from the clumped lead variants with $P < 5E-08$ from each trait instead of PRScs; and thresholding the cohort into the top 5% for each polygenic score compared to the bottom 95% for the score.

The **Results>A polygenic estimate of LA volume predicts AF, stroke, and heart failure** section now states:

Sensitivity analyses using lead SNP scores, different covariate adjustments, or different population subgroups yielded similar results (Supplementary Table 14).

Results

Major:

- It is important to thoroughly evaluate the performance of the algorithm, since it has not been published previously. Could you include mean contour distance and Hausdorff distance? How does this metrics compare to previously published methods by Bai et al. (J. Cardiovasc. Magn.

Reson., 2018)?

Author response

Thanks, we have now included the Hausdorff distance and mean contour distance values. The performance metrics in the 2ch and 4ch views are slightly worse than those reported by Bai *et al*, but similar to their report, our mean contour distances are smaller than the in-plane distance between two pixels (1.83mm).

Manuscript change

The **Supplementary Note>Methods>Semantic segmentation model quality assessment** section now includes:

The quality of the deep learning segmentation output was assessed against manually annotated segmentations in held-out test samples using the Sørensen-Dice coefficient, the Hausdorff distance, and the mean contour distance (Dice, 1945; Huttenlocher, Klanderman and Rucklidge, 1993). The Sørensen-Dice coefficient addresses the total segmentation area of the left atrium, and is a dimensionless value that ranges from 0 for an image where no pixels overlap between human and machine labels, to 1 for an image with perfect overlap between human and machine labels. The Sørensen-Dice was calculated by dividing twice the number of overlapping pixels between the two sets (the intersection) by the sum of the individual pixels considered to be left atrium in each set.

The Hausdorff distance and the mean contour distance address the perimeter of the manual and automated segmentations, and to obtain this perimeter the `binary_erosion` function from the python3 library `scikit-image` version 0.19.3 was used. The Hausdorff distance represents the maximum distance in millimeters (mm) for any point in the perimeter of the automated segmentation output to its nearest point in the perimeter of

the manually annotated segmentation. The Hausdorff distance was calculated using the `directed_hausdorff` function from the `scipy.spatial.distance` python3 library, version 1.11.4. The mean contour distance represents the average distance in mm of each point on the automated segmentation output to its nearest point in the perimeter of the manually annotated segmentation. The mean contour distance was calculated for each point in the automated segmentation perimeter by testing the distance to every point in the perimeter of the manually annotated data; retaining the minimum distance for each point; and then taking the average for all points in the automated segmentation perimeter.

The **Supplementary Note>Results>Semantic segmentation quality assessment** section now states:

In a held-out test set of 20 manually annotated images from the two-chamber short axis view that were not used in training or validation, the average Dice coefficient was 0.89 (SD 0.06) for the left atrial blood pool. For 20 held-out images from the three-chamber view, the Dice score was 0.88 (SD 0.07). For 40 held-out images from the four-chamber view, the Dice score was 0.94 (SD 0.03).

The short axis imaging sequence was not designed to capture the atria: the atrial short axis sequence was eliminated from the acquisition protocol to save acquisition time (Petersen et al., 2016). The left atrium was nevertheless recognizable in the basal-most segments of images obtained in the short axis view. In the short axis view, the average Dice score for the left atrium was 0.78 (SD 0.35) when weighted by the total number of pixels assigned to the left atrium by the cardiologist or the model, or 0.90 (SD 0.28) when considering images correctly identified by the model as having no left atrial pixels to have a Dice score of 1.

In the two-chamber view, the average Hausdorff distance was 6.7mm (SD 4.0mm). In the three-chamber view, the average Hausdorff distance was 8.8mm (SD 8.5mm). In the four-chamber view, the average Hausdorff distance was 5.2mm (SD 4.1mm). In the short-axis view, the average Hausdorff distance was 5.8mm (SD 4.2mm).

In the two chamber view, the average mean contour distance was 1.8mm (SD 0.6mm). In the three-chamber view, the average mean contour distance was 2.3mm (SD 2.2mm). In the four-chamber view, the average mean contour distance was 1.3mm (SD 0.90mm). In the short-axis view, the average mean contour distance was 1.7mm (SD 1.7mm). The mean contour distance for the automated left atrial segmentation in each of these views was less than the in-plane pixel spacing of 1.83mm.

- In order to gage how the measurements holds up against manual annotation and to see if there are unknown biases, I would like to see inter-observer variability between manual and estimated minimum/maximum volumes. Could this be performed by independent clinical experts on minimum of 30 samples and be shown graphically?

Author response

The quality of the reconstructed volumes is a fundamental question, thanks for bringing this up. We now attempt to address it in two ways: first, by showing that our method behaves similarly to manually-measured biplane volumes when we force our method to use only 2ch and 4ch inputs; and second, by showing that our estimates are more strongly correlated with disease than the biplane estimates.

A key limit in the data underlying our analysis is that the UK Biobank imaging acquisition protocol did not perform volumetric capture of the left atrium (which was omitted to save substantial scanning time from the UK Biobank imaging protocol). Therefore, we do not have a three dimensional volume that can be manually measured by expert readers and used as a

ground truth model; instead, all left atrial volumes will be estimates of one form or another based on integrating data from different views. And, consequently, we don't have the raw material to perform manual segmentation in the volumetric stacks and provide visuals for them. Still, we can perform a comparison to a common clinical volume estimate: a biplane measurement (based on 2ch and 4ch) that was measured by an independent group of experts.

However, if the biplane estimate is treated as the ground truth, then incorporating additional information from the 3ch and short axis views will necessarily worsen the apparent correlation between our measurements and the truth. So our goal in this response is not to show that our values most closely approximate the biplane measurements, but instead that *the Poisson surface reconstruction approach more closely approximates the biplane results when we include only the 2ch and 4ch data and exclude the 3ch and SAX data from the reconstruction process.*

And even then, we do not expect our model to produce the same results as the standard biplane formula: the assumptions of the classical biplane formula are that the left atrium is an ellipsoid with similar shapes in the 2ch and 4ch view, whereas the Poisson surface reconstruction model assumes only that the surface is smooth and continuous, and can therefore model more complex shapes.

With those caveats, we turned to left atrial biplane volumes that have been manually computed by expert cardiologists in the group of Dr. Steffen Petersen in the UK and made available to UK Biobank researchers as a "Return" on the UK Biobank Showcase. As detailed below, we found that when we included only the 2ch and 4ch views ("Poisson biplane"), the LA volumes had a high correlation with the manual biplane measurements (LAmax $r=0.887$, from 0.814 for the full Poisson model; LAmin $r=0.860$, from 0.768 for the full Poisson model). And despite the stronger correlation between the Poisson biplane and the manual measurements, we found that the full model had an association that was equally strong or stronger for identifying prevalent atrial

fibrillation (LAmax OR 1.72 for the full Poisson model vs 1.65 for the Poisson biplane model; and, respectively, LAmin OR 1.86 vs 1.80).

Although we recognize that this does not address every element of your inquiry, we interpret these findings as supporting the notion that the Poisson technique correlates well with the biplane model when fed the same input planes, and where it differs from the biplane measurements it does so in a way that is better linked with relevant disease.

Manuscript change

We have created a **Supplementary Note>Methods>Comparison with left atrial biplane measurements** section:

The UK Biobank's cardiovascular MRI imaging protocol did not include a volumetric short-axis stack throughout the left atrium (Petersen et al., 2016), so left atrial measurements represent estimates of an unmeasured true left atrial volume. To assess quality, we compared the Poisson surface reconstruction approach with biplane measurements and tested each for association with prevalent atrial fibrillation. Using the R function `cor.test`, we correlated the Poisson surface reconstruction algorithm-based left atrial volume measurements with biplane-based volumes manually measured by experts (Petersen et al., 2017).

A new **Supplementary Note>Results>Comparison with left atrial biplane measurements** section now reads:

We correlated the Poisson surface reconstruction algorithm-based left atrial volume measurements with biplane-based volumes manually measured by experts in 3,401 participants (Petersen et al., 2017). When limiting the inputs into the Poisson surface reconstruction algorithm to only the two- and four-chamber long axis views ("Poisson

biplane”), which are the two views used to calculate the biplane volume, the correlation improved for both LAm_{max} (from $r=0.814$, 95% CI 0.802 to 0.825, $P=2.9E-804$ with the full reconstruction to $r=0.887$, 95% CI 0.880 to 0.894, $P=4.5E-1143$ with the Poisson biplane) and LAm_{min} (from $r=0.768$, 95% CI 0.754 to 0.781, $P=1.1E-659$ to $r=0.860$, 95% CI 0.851 to 0.868, $P=6.9E-994$). We interpreted these results as supporting the notion that, when presented with the same input information, the modeling approach yields estimates that are similar to the standard biplane estimation.

We then used logistic regression to recapitulate prior observations that individuals with pre-existing atrial fibrillation have larger atrial volumes (Sanfilippo et al., 1990; Sardana Mayank et al., no date). In a subset of 39,148 participants, of whom 808 had atrial fibrillation, both the full Poisson reconstruction and the Poisson biplane reconstruction could be performed. Although the Poisson biplane better correlated with the manual measurements in the previous analysis, the full Poisson reconstruction was more strongly associated with prevalent atrial fibrillation (LAm_{max} OR 1.72, $P=1.3E-78$ and LAm_{min} OR 1.86, $P=1.0E-132$) compared to the Poisson biplane model (LAm_{max} OR 1.65, $P=6.3E-66$ and LAm_{min} OR 1.80, $P=2.8E-130$).

We interpreted these findings as indicating that (1) the Poisson-based measurements were well correlated with manual measurements, and (2) while full volumetric imaging stacks through the atria were not available to adjudicate correctness, the Poisson-based measurements that incorporated all available views (2ch, 3ch, 4ch, and SAX) were more strongly correlated with atrial fibrillation than the Poisson biplane measurements.

- Would be nice to get some more details on the performance on the model used to classify abnormal atrial contraction.

Author response

To address this, we have conducted two analyses. First, we have selected 100 examples flagged as abnormal and 100 examples flagged as normal by the model and manually reviewed the images to assign a ground truth label to produce estimates for sensitivity and specificity. (Because of severe class imbalance, randomly selecting 200 images would likely only have chosen ~4 images flagged by the model, hence this analytic design.) The model was sensitive (95% CI 90.3-100.0%) but nonspecific (95% CI 53.1-68.5%), which we interpreted to mean that most abnormal cardiac motion was likely captured by the model, while some of the participants may have been unnecessarily excluded, which would reduce power. Because our purpose here was to avoid incorporating data from people who may have been in atrial fibrillation (which would impair the causal interpretation of the Mendelian randomization), and because few participants were excluded by the model, we felt that this was acceptable.

Second, we have now conducted sensitivity-analysis GWASes for LAmin and BSA-indexed LAmin that were like the primary GWAS in all ways except that they did not exclude samples based on the abnormal contraction model. This increased the GWAS sample size by 615 (because some of the 1,013 participants that were recovered by disabling this filter still failed other sample filters). Notably, the association strength for the *PITX2* locus for non-indexed LAmin increased from $P=4.6E-06$ to $P=3.10E-08$ in this sensitivity analysis. On balance, an additional two loci with $P<5E-08$ were gained in this analysis for LAmin, compared to the primary analysis.

Manuscript change

The **Supplementary Note>Methods>Identification of abnormal cardiac filling patterns** section now includes this quality assessment:

To evaluate the accuracy of the deep learning model, manual evaluation of the cardiac filling patterns was conducted by one cardiologist (JPP) for 100 participants flagged as

having abnormal cardiac filling patterns and 100 flagged as having normal cardiac filling patterns, sampled at random from participants without a history of atrial fibrillation.

Sensitivity and specificity and their confidence intervals were calculated with the `binom.test` function in R.

The **Supplementary Note>Methods>GWAS sensitivity analysis: no exclusion for abnormal cardiac filling patterns** section was created:

A sensitivity analysis was conducted to assess the consequence of retaining participants identified by the deep learning model as having apparently abnormal cardiac filling. For this sensitivity analysis, only LAmin and BSA-indexed LAmin were evaluated. Like the primary analyses, BOLT-LMM was used for this analysis with the same covariates and settings (Online Methods).

The **Supplementary Note>Results>Quality control for the deep learning model for abnormal cardiac filling patterns** section now states:

Among 200 participants whose MRIs were manually reviewed (100 flagged as having abnormal cardiac filling patterns and 100 flagged as having normal cardiac filling patterns), manual review determined that 164 were normal and 36 were abnormal. The sensitivity of the model for identifying abnormal cardiac filling patterns was 100% (95% CI 90.3-100.0%) and the specificity was 61% (95% CI 53.1-68.5%). These findings suggested that the model may have over-detected abnormal cardiac filling—leading to the exclusion of more participants than necessary—but had little evidence for false negatives.

The **Supplementary Note>Results>GWAS sensitivity analysis: no filtering for abnormal cardiac filling patterns** section now states:

Given the high sensitivity but low specificity of the model detecting abnormal cardiac filling patterns, sensitivity analysis retained the 615 participants who were not identified as having a normal cardiac filling pattern for GWAS of LAmin and BSA-indexed LAmin, yielding a total sample size of N=35,664 participants. (Because some participants are excluded by other criteria downstream of this filter in the primary GWAS, this number is smaller than the 1,013 noted in Supplementary Figure 3.) The lead SNPs are recorded in Supplementary Table 20. Compared with the main analysis of 35,049 participants, some loci with marginal P-values were lost while others were gained; net, an additional two loci (10 in total) were identified for LAmin and an unchanged number of loci (13) were significant for BSA-indexed LAmin. For example, the association signal for PITX2 variant rs2466455 for LAmin increased in significance from $P=4.6E-06$ to $P=3.10E-08$ in this sensitivity analysis. Similarly the strongest associated variant near PITX2 for BSA-indexed LAmin in this analysis (rs2723334, $P=1.70E-10$) had stronger evidence for association than in the primary analysis ($P=2.2E-08$).

- In which units are the Beta estimates in table 2, rank-based inverse transformation?

Author response

You are correct, this is a rank-based inverse normal transformation unit, which is approximately a standard deviation.

Manuscript change

The **Table 2** caption now explains:

BP: GRCh37-base position. dbSNP: dbSNP identifier, where available. EAF: Effect allele frequency. BETA: BOLT-LMM effect size of the effect allele, in units of the rank-based inverse normal transform which approximates a standard deviation change. SE:

Standard error. P: BOLT-LMM P-value. “Indexed” indicates that the trait has been divided by body surface area.

- In the heritability & genetic correlation analysis you don't provide estimates for BSA-indexed phenotypes, which you say (might be the case?) are in the main analysis? The same goes for ldsc intercepts.

Author response

Thanks for noting this. This omission was due to a quirk in how we were (sequentially) computing these. We now compute them in parallel, making their computation feasible, so we are able to report these features.

Manuscript change

The **Results>Common genetic variant analysis of LA size and function identifies 20 loci** section now states:

First, we examined the SNP-heritability of the LA traits which ranged from 0.14 (LAEF) to 0.37 (LAm_{ax}; Supplementary Table 5). Genetic correlation between the LA measurements ranged from -0.72 (between LAm_{in} and LAEF) to 0.95 (between LAm_{ax} and LAm_{in}; Supplementary Table 5).

- As a suggestion, if you are indeed indexing for LVEDV in the sensitivity analyses, you might want to reconsider and instead perform a conditional analysis with LVEDV.

Author response

Thanks, this is a good idea and makes a lot of sense for this sensitivity analysis. Given the extensive sensitivity analyses that we have conducted to address other points that have been raised, we have not implemented this at this time, but we'll keep it in mind for future analyses.

- In the genetic correlation analysis with other traits, you are only reporting on non-indexed traits. Does this mean BSA-indexing is part of a sensitivity analysis?

Author response

Thanks for noting this error. We have now added the BSA-indexed genetic correlation results to the Supplementary Table. These do not change our main results, which focus on the strongest associations, but we do now specifically call them out now in the Results.

Manuscript change

The **Results>Genetic relationship between AF risk and LA dysfunction** section now mentions this:

Using ldsc, the strongest genetic correlation was found between LAmin and AF (r_g 0.37, $P=2.0E-10$), a direction of effect that corresponds to a positive correlation between LA dysfunction (i.e., increased LAmin) and risk for AF (Supplementary Table 9)(B. Bulik-Sullivan et al., 2015; Roselli et al., 2018). This relationship was minimally attenuated after indexing on BSA (r_g 0.33, $P=7.7E-09$).

- Is the correlation to AF and all cause stroke in part through other risk factors relating to body size, e.g. hypertension, rather than LA volume?

Author response

This is a very good question. Because this is also a causal question, an approach to address this would likely incorporate both mediation analyses and a set of Mendelian randomization analyses for each of these traits, with sensitivity analyses to assess each assumption. We feel that such an analysis would require a detailed undertaking that would best be evaluated in its own dedicated manuscript(s) so that the details can be properly addressed.

- The section on AF PRS association with LA phenotypes is not described in Methods.

Author response

Thanks for pointing out our oversight. We have now added this description to the Methods. We also took this opportunity to slightly modify the analysis: we now apply the score only to the same group that participated in the GWAS (rather than also applying it to participants who were excluded from the GWAS). This had no material consequence on the findings but we felt it would be a bit more cohesive for the reader.

Manuscript change

The **Results>A polygenic risk score for AF is associated with LA phenotypes** section now contains the slightly updated findings:

We constructed a 1.1-million SNP polygenic risk score (PRS) with PRScs using summary statistics from the Christophersen, et al, AF GWAS, and applied this score in the 35,049 LA GWAS participants(Christophersen et al., 2017; Ge et al., 2019). The AF PRS was statistically significantly associated with all measures of LA size and function, with a small effect size (Supplementary Table 12). The strongest association was with LAmin (0.052 SD increase in LAmin per SD increase in the PRS; 95% CI 0.042-0.061; $P=1.1E-25$).

The **Methods>Polygenic score for atrial fibrillation** section now states:

We constructed a 1.1-million SNP PRS using PRScs based on summary statistics from Christophersen, et al, 2017—a large AF GWAS that did not incorporate UK Biobank participants (Christophersen et al., 2017; Ge et al., 2019). The score was constructed from 1,108,410 sites from the summary statistics that overlapped with the HapMap3 sites available in the UK Biobank as precomputed by the PRScs authors. The score was

applied to the GWAS participants with LA measurements and tested for association using linear regression (Supplementary Table 12). For comparability, the score and the LA measurements were both standardized to a mean of zero and a standard deviation of 1.

- In the section on LA volume PRS to predict AF, the population used is better described. I think the method and result section on the PRS post-analysis probably needs to be synced. However, there is still a potential issue with information leakage by PC's when predicting AF. Still not clear whether this is LAmin or BSA-indexed LAmin, which it is says in Figure 6. Also, the figure is showing top 5% PRS, which is not really reported in the results.

Author response

Thanks, these are all good points. We have now made sure to sync our definition of atrial fibrillation to align with the same definition we used for GWAS exclusions (previously we errantly used different versions, which differed in the use of cardioversion as an inclusion criterion). This has only a trivial effect on the statistical results, but it slightly changes the case counts.

We have now also computed PCA within the GWAS samples only and used those to perform a sensitivity analysis, which produced no meaningful change in the results.

The KM curves require groups for display, so we chose a high threshold to demonstrate the effect of a high PRS, and we now report them numerically in the supplement as well.

Manuscript change

The **Supplementary Note>Methods>Internal validation of LA polygenic scores in non-imaging participants** section now incorporates these additional analyses:

...

Sensitivity analyses included restriction participants to a genetic inlier population with European genetic identity (precomputed by the UK Biobank); adjusting for genetic principal components derived from the GWAS samples instead of the entire cohort; adjusting only for age and sex; applying score weights derived from the clumped lead variants with $P < 5E-08$ from each trait instead of PRSCs; and thresholding the cohort into the top 5% for each polygenic score compared to the bottom 95% for the score.

The **Results>Polygenic estimates of LA volume predict AF, stroke, and heart failure** section now states:

Those in the top 5% of the score had a greater risk of AF (HR=1.19, $P=7.9E-10$), ischemic stroke (HR=1.12, $P=0.06$), and heart failure (HR=1.14, $P=1.2E-03$; Supplementary Table 14).

...

Sensitivity analyses using lead SNP scores, different covariate adjustments, or different population subgroups yielded similar results (Supplementary Table 14).

- Eight of the loci associated with LA traits have also previously been associated with AF. Interestingly they find that “the AF risk alleles were associated with an increased LA minimum volume (LAmin) and a decreased LAEF. A Mendelian randomization analysis confirmed that AF causally affects LA volume (IVW $P = 6.2E-06$), and provided evidence that LAmin causally affects AF risk (IVW $P = 4.7E-05$)”. Were these SNPs associated with hypertension or LVEF, could this be secondary to systolic dysfunction?

Author response

We were quite curious about this and our attempts to understand this are described in the Supplementary Note. We specifically looked at seven AF risk factors from the CHARGE-AF

predictive model, including SBP, DBP, hypertension, height, weight, diabetes, and smoking status. Three SNPs were associated with one or more of those (rs10878349 near IRAK3, rs56129480 near SP3, and rs78033733 near MYL4). Therefore, we also performed a sensitivity analysis excluding these SNPs. Of note we also performed Mendelian randomization sensitivity analyses with MR-PRESSO and the contamination mixture model, which are robust to some degree of invalid instruments.

A deeper dive into the complex relationships between blood pressure, the left ventricle, and the left atrium is very interesting but we felt that this interplay would require its own manuscript to carefully dissect.

Manuscript change

The **Results>Causal link between LA minimal volume and disease risk** section now states:

There was significant effect heterogeneity ($P=2.9E-05$ by Cochran Q), so the contamination mixture model approach and MR-PRESSO were applied, both of which showed a significant, positive relationship between LAmin and AF with the same direction of effects (Supplementary Table 10; Supplementary Figure 5). MR-Egger results did not reach nominal significance, nor did they yield evidence for horizontal pleiotropy (intercept $P=0.48$). Within the GWAS participants, three of the 19 SNPs had evidence for pleiotropic association with AF risk factors that were derived from the CHARGE-AF risk score (Supplementary Figure 6) (Alonso et al., 2013); a sensitivity analysis excluding these three variants yielded similar results (IVW OR 1.89 per SD increase in LAmin, $P=7.3E-06$; Supplementary Table 10; Supplementary Figure 7).

The **Supplementary Note>Supplementary Figure 6 - Pleiotropic associations for variants used in Mendelian randomization** figure is as follows:

Each of the 19 SNPs from the Lamin Mendelian randomization analysis was tested for association with seven phenotypes previously identified as atrial fibrillation risk factors in CHARGE-AF. For each SNP, this figure displays the point estimate of the effect of 1 unit change in the dosage of the non-reference allele on each trait. Traits where the association with the SNP achieves Bonferroni significance are shown in red. Three of the 19 SNPs were identified to have a significant association with at least one putative confounding factor (rs10878349 near IRAK3, rs56129480 near SP3, and rs78033733 near MYL4).

- A large part of the cases have hypertension have you tested the genetic correlation with Hypertension?

Author response

This is a good question. As we observed in this manuscript, confirming prior observations, there is an epidemiological correlation between hypertension and greater left atrial volumes. Here, we focused on diseases that we thought might be downstream of atrial dysfunction (e.g., atrial fibrillation, stroke) but we agree that upstream causes of atrial dysfunction (e.g., hypertension) are interesting, too.

Our approach for in-sample genetic correlation (BOLT-REML) of the LA measurements elsewhere in the manuscript is not well suited to binary traits like hypertension. So, we investigated using *ldsc* for performing summary statistic-based genetic correlation between LA traits and hypertension. Hypertension summary statistics are somewhat limited; the International Consortium for Blood Pressure did not make its GWAS summary statistics publicly available, for example. We therefore conducted a GWAS of hypertension in the whole UK Biobank and then performed *ldsc* genetic correlation to try to address this. The genetic correlation was positive between hypertension and LAmax ($rg=0.19$, $P=2.7E-09$), LAmin ($rg=0.16$, $P=1.9E-06$), and LASV ($rg=0.17$, $P=7.3E-06$). However, the result was muted for LAEF ($rg=-0.04$, $P=0.4$). The null finding for LAEF could certainly be true, but it could also be due to power (and the effect is otherwise in the expected inverse direction). We feel that this is important enough to get right that, before reporting such results, it would require a more thorough set of troubleshooting analyses including, likely, a request to ICBP to obtain the best-powered hypertension summary statistics to clarify whether the LAEF findings are null due to low power or due to biology. Other approaches might provide greater clarity (e.g., the Latent Causal Variable model from O'Connor *Nat Genet* 2018, or Mendelian randomization). Given the above observations, and based on the scope of the present work, we feel that an in-depth assessment of the hypertension relationship likely warrants a standalone body of work, ideally one that also addresses left ventricular

hypertrophy in this process. Therefore, we have not incorporated these results into the manuscript at this time.

Discussion

- The discussion mentions the use of a PRS on BSA-indexed LAmin as opposed to LAmin in the Method.

Author response

Thanks for pointing out the error in our text. Given the above sensitivity analyses with additional polygenic scores, this wording has been modified in the Methods. The sensitivity analyses have been described above so are not repeated here.

Manuscript change

The **Methods>Polygenic risk analysis** section now begins:

A polygenic score for each LA GWAS was computed using PRSCs with a UK Biobank European ancestry linkage disequilibrium panel(Ge et al., 2019).

...

Other polygenic scores were produced as sensitivity analyses (Supplementary Note).

Conclusion

"In future work, it will be interesting to determine if targeting the genes and pathways associated with abnormalities in LA function will be helpful to reduce the risk of AF, heart failure, and stroke"

This a perspective that has not really been discussed in the discussion, I suggest that you discuss it or removed it from the conclusion.

Author response

Thanks, you are right that it's not really a concluding statement and we have moved it out of the Conclusion and into the Discussion accordingly.

Manuscript change

We have moved this sentence to the discussion.

Figures

- Figure 4, gene names in figures overlap.

Author response

Thanks for pointing this out. We have now modified the figure to eliminate gene-name overlap and to try to improve appearance and legibility.

Manuscript change

Figure 4 has been updated:

- Figure 5, would you say that BSA-indexed LA volumes overlap more with AF loci?

Author response

Our interpretation is that they are quite similar but that BSA indexing enhances the signal. At every locus that has $P < 5E-08$ for the BSA-indexed traits, at least one of the non-indexed traits has $P < 5E-06$. We don't see, e.g., signal that is exclusive to the BSA-indexed traits. To us, this is reassuring that BSA-indexing here is largely refining signal that was already present in the non-indexed traits.

- Supplementary figure 3 and 5, why is effect size of LAmin represented as LAESV?

Author response

Thanks for noticing this, it was an inadvertent consequence of internal variable names that were left in the figures that we have now corrected thanks to your observation. The figure labels have been corrected.

Reviewer #2 (Remarks to the Author):

This paper investigates genetic factors that contribute to CMR-derived parameters of atrial structure and function using the UK Biobank data. Finding sensitive and accurate ways to assess atrial cardiomyopathy is a clinically important topic of current interest.

The first section describes application of deep learning methods to assess left atrial maximal and minimal volumes and the derived parameters, left atrial stroke volume and ejection fraction. This is a novel way to rapidly obtain these parameters in a standardised fashion. In a similar study design, Ahlberg et al (Eur Heart J 2021) used deep learning methods to investigate genetic correlates of atrial structure and function in same UK Biobank dataset. This yielded overlapping but non-identical genetic results, highlighting the variability that different deep learning methods can introduce. Since all the subsequent analyses in Pirruccello et al's paper rely on these atrial parameters, it would be useful to undertake a validation analysis eg. by comparing CMR data with echocardiographic data in a subset of individuals.

Author response

This is a very interesting point and it would be a major boon if there were overlapping samples with MRI and echocardiographic data. However, such data do not exist in the UK Biobank. We have therefore tried to triangulate as best we could: first, by showing that our method behaves similarly to manually-measured biplane volumes when we force our method to use only 2ch and 4ch inputs; and second, by showing that our estimates are more strongly correlated with disease than the biplane estimates.

In addition to the absence of echo data, the UK Biobank imaging acquisition protocol did not perform volumetric capture of the left atrium; they saved substantial scanning time by removing the short axis volumetric atrial stack from the protocol. Therefore, we do not have a three

dimensional volume that can be manually measured by expert readers and used as a ground truth model; instead, all left atrial volumes will be estimates of one form or another based on integrating data from different views.

As detailed below, we found that when we included only the 2ch and 4ch views (“Poisson biplane”), the LA volumes had a high correlation with the Petersen group’s manual biplane measurements (LA_{max} $r=0.887$, from 0.814 for the full Poisson model; LA_{min} $r=0.860$, from 0.768 for the full Poisson model). And despite the fact that there was a stronger correlation between the Poisson biplane and the manual measurements, we found that the full (non-biplane) Poisson model had an association that was equally strong or stronger with prevalent atrial fibrillation (LA_{max} OR 1.72 for the full Poisson model vs 1.65 for the Poisson biplane model; and, respectively, LA_{min} OR 1.86 vs 1.80).

We think that these data, when taken together in aggregate, are supportive of our approach.

Manuscript change

The **Supplementary Note>Methods>Comparison with left atrial biplane measurements** section now states:

The UK Biobank’s cardiovascular MRI imaging protocol did not include a volumetric short-axis stack throughout the left atrium(Petersen et al., 2016), so left atrial measurements represent estimates of an unmeasured true left atrial volume. To assess quality, we compared the Poisson surface reconstruction approach with biplane measurements and tested each for association with prevalent atrial fibrillation. Using the R function `cor.test`, we correlated the Poisson surface reconstruction algorithm-based left atrial volume measurements with biplane-based volumes manually measured by experts(Petersen et al., 2017).

The **Supplementary Note>Results>Comparison with left atrial biplane measurements**

section now states:

We correlated the Poisson surface reconstruction algorithm-based left atrial volume measurements with biplane-based volumes manually measured by experts in 3,401 participants (Petersen et al., 2017). When limiting the inputs into the Poisson surface reconstruction algorithm to only the two- and four-chamber long axis views (“Poisson biplane”), which are the two views used to calculate the biplane volume, the correlation improved for both LAm_{ax} (from $r=0.814$, 95% CI 0.802 to 0.825, $P=2.9E-804$ with the full reconstruction to $r=0.887$, 95% CI 0.880 to 0.894, $P=4.5E-1143$ with the Poisson biplane) and LAm_{in} (from $r=0.768$, 95% CI 0.754 to 0.781, $P=1.1E-659$ to $r=0.860$, 95% CI 0.851 to 0.868, $P=6.9E-994$). We interpreted these results as supporting the notion that, when presented with the same input information, the modeling approach yields estimates that are similar to the standard biplane estimation.

We then used logistic regression to recapitulate prior observations that individuals with pre-existing atrial fibrillation have larger atrial volumes (Sanfilippo et al., 1990; Sardana Mayank et al., no date). In a subset of 39,148 participants, of whom 808 had atrial fibrillation, both the full Poisson reconstruction and the Poisson biplane reconstruction could be performed. Although the Poisson biplane better correlated with the manual measurements in the previous analysis, the full Poisson reconstruction was more strongly associated with prevalent atrial fibrillation (LAm_{ax} OR 1.72, $P=1.3E-78$ and LAm_{in} OR 1.86, $P=1.0E-132$) compared to the Poisson biplane model (LAm_{ax} OR 1.65, $P=6.3E-66$ and LAm_{in} OR 1.80, $P=2.8E-130$).

We interpreted these findings as indicating that (1) the Poisson-based measurements were well correlated with manual measurements, and (2) while full volumetric imaging

stacks through the atria were not available to adjudicate correctness, the Poisson-based measurements that incorporated all available views (2ch, 3ch, 4ch, and SAX) were more strongly correlated with atrial fibrillation than the Poisson biplane measurements.

In the Results (line 112), 1015 participants with increased left atrial volumes were removed from the analysis. Additional information is required to justify this, since exclusion of these individuals could change the subsequent analysis between LAmin and incident AF. It cannot be assumed that these individuals had undiagnosed AF. Were there any other identifiable causes for left atrial dilation?

Author response

This is a good point. Our original intent was to be aggressive with our exclusion of participants with known or potential atrial fibrillation to maximize the validity of our analyses looking at left atrial measurements and future risk of atrial fibrillation, and the genetic relationship between the two. We felt that being over-aggressive about exclusion would not be philosophically problematic (aside from squandering power), while inadequate exclusion could lead to spurious causal claims (e.g., if cryptic AF were actually driving early LA volume enlargement). However, your points are well taken. We have tried to address them in two ways.

First, we have updated Table 1 to include the count data for conditions such as mitral regurgitation and mitral stenosis among GWAS participants. These are vanishingly rare in this group (19 people with mitral regurgitation and 3 people with mitral stenosis), and conditions such as heart failure and atrial fibrillation were exclusion criteria and therefore have precisely 0 remaining participants with such diagnoses at the time of MRI in the GWAS cohort.

Second, given the limited data (e.g., no velocity-encoded imaging through the mitral valve plane to infer mitral stenosis or regurgitation), we have also tried to address this question by adding a sensitivity analysis to show what would change if we did not filter out these participants but

instead included them in the GWAS. We did this for LAmin and BSA-indexed LAmin—these sensitivity analyses were conducted like the primary GWAS in all ways except that they did not exclude samples based on the abnormal contraction model. This increased the GWAS sample size by 615 (because some of the 1,013 participants that were recovered by disabling this filter still failed other sample filters). Notably, the association strength for the *PITX2* locus for non-indexed LAmin increased from $P=4.6E-06$ to $P=3.10E-08$ in this sensitivity analysis. On balance, an additional two loci with $P<5E-08$ were gained in this analysis for LAmin, compared to the primary analysis.

Manuscript change

The **Supplementary Note>Methods>GWAS sensitivity analysis: no exclusion for abnormal cardiac filling patterns** section now states:

A sensitivity analysis was conducted to assess the consequence of retaining participants identified by the deep learning model as having apparently abnormal cardiac filling. For this sensitivity analysis, only LAmin and BSA-indexed LAmin were evaluated. Like the primary analyses, BOLT-LMM was used for this analysis with the same covariates and settings (Online Methods).

The **Supplementary Note>Results>GWAS sensitivity analysis: no exclusion for abnormal cardiac filling patterns** section now states:

Given the high sensitivity but low specificity of the model detecting abnormal cardiac filling patterns, sensitivity analysis retained the 615 participants who were not identified as having a normal cardiac filling pattern for GWAS of LAmin and BSA-indexed LAmin, yielding a total sample size of $N=35,664$ participants. (Because some participants are excluded by other criteria downstream of this filter in the primary GWAS, this number is smaller than the 1,013 noted in Supplementary Figure 3.) The lead SNPs are recorded

in Supplementary Table 20. Compared with the main analysis of 35,049 participants, some loci with marginal P-values were lost while others were gained; net, an additional two loci (10 in total) were identified for LAmin and an unchanged number of loci (13) were significant for BSA-indexed LAmin. For example, the association signal for PITX2 variant rs2466455 for LAmin increased in significance from $P=4.6E-06$ to $P=3.10E-08$ in this sensitivity analysis. Similarly the strongest associated variant near PITX2 for BSA-indexed LAmin in this analysis (rs2723334, $P=1.70E-10$) had stronger evidence for association than in the primary analysis ($P=2.2E-08$).

In addition, **Table 1** now displays the count data for several conditions:

	Women	Men	Both
N	18,916	16,133	35,049
Age at time of MRI	64 (8)	65 (8)	64 (8)
BMI (kg/m²)	26 (5)	27 (4)	26 (4)
Height (cm)	163 (6)	176 (7)	169 (9)
Weight (kg)	69 (13)	83 (13)	75 (15)
Systolic Blood Pressure (mmHg)	136 (19)	142 (17)	139 (19)
Diastolic Blood Pressure (mmHg)	77 (10)	81 (10)	79 (10)
Left atrium maximum volume (cm³)	64 (15)	79 (19)	71 (18)
Left atrium minimum volume (cm³)	28 (9)	37 (12)	32 (11)
Left atrium stroke volume (cm³)	36 (8)	43 (11)	39 (10)
Left atrium emptying fraction (%)	57 (8)	54 (7)	56 (8)
Mitral regurgitation (%)	10 (0)	9 (0)	19 (0)
Mitral stenosis (%)	3 (0)	0 (0)	3 (0)

Heart failure (%)	0 (0)	0 (0)	0 (0)
Hypertrophic cardiomyopathy (%)	0 (0)	0 (0)	0 (0)
Congenital heart disease (%)	3 (0)	1 (0)	4 (0)
Aortic valve disease (%)	18 (0)	21 (0)	39 (0)
Atrial fibrillation or flutter (%)	0 (0)	0 (0)	0 (0)

Characteristics of the participants who contributed to the GWAS are listed as mean (standard deviation). Count data are listed as number (%).

The GWAS study and Lamin-PRS are important and expand the spectrum of genetic variants that are associated with AF and stroke risk. **If possible, it would be useful to validate these associations in a replication cohort.**

Author response

Thanks. To try to be responsive to this point, we have now pursued validation of the PRS in two large external biobanks: *All of Us* and FinnGen.

First, we should mention that we also investigated trying to replicate the GWAS itself, but did not identify a satisfactory cohort in which to pursue this. While there are several cohorts with genetics and ventricular measurements, to our knowledge the largest non-UK Biobank cohort with *atrial* and genetic data available is Framingham. There, a GWAS was published by Magnani et al in JAHA in 2014 (*Genetic Loci Associated With Atrial Fibrillation: Relation to Left Atrial Structure in the Framingham Heart Study*). Both MRI (N=1,555) and echocardiographic measurements (N=6,861) were tested; however, there were no significant results. For example, we can look up their strongest SNP, which is near the classical atrial fibrillation-linked *PITX2* locus. The SNP rs2200733, had P=0.01 in their MRI cohort and P=0.8 in their larger echo

cohort. Yet we find the locus to be strongly associated (that exact SNP, which is not the lead SNP at the locus in our study, has $P=9.7E-07$). Given the severe lack of power in FHS, for external validation we instead relied on the aggregate genetic information from the polygenic score.

In our external validation analyses for the BSA-indexed LAmin polygenic score, FinnGen, in particular, was sufficiently powered to replicate the positive relationships that we originally observed in UK Biobank between a greater value for the BSA-indexed LAmin polygenic score and higher risk of AF, heart failure, and stroke. Compared to FinnGen, *All of Us* has ~24% the case count for AF, 41% for heart failure, and only 0.5% as many ischemic stroke cases.

Nevertheless, we were also able to replicate the AF and heart failure observations in *All of Us*.

Somewhat relevant to the point above about Framingham, because 680 *All of Us* participants also had BSA-indexed LAmin measurements, we were also able to show that the polygenic score from UK Biobank is correlated with its corresponding measurement in *All of Us*. (As a minor point, we note that currently this is the *only* volumetric LA measurement available in *All of Us*—more standard measures are not available. We have contacted *All of Us* about this, but their reply was that this is a function of what the data providers are reporting to them.)

Manuscript change

(The FinnGen and *All of Us* methods occupy 5 pages of the supplement and are not duplicated here for brevity.)

In the main text, the **Results>External validation of the LAmin polygenic score in FinnGen and All of Us** section is new:

In FinnGen(Kurki et al., 2022) study participants (Supplementary Table 15), comparable associations were observed for association between the BSA-indexed LAmin polygenic

score and incident AF or atrial flutter (20,422 cases, HR=1.08 per SD, P=2.4E-30), ischemic stroke excluding subarachnoid hemorrhage (13,392 cases, HR=1.03 per SD, P=3.0E-03), ischemic stroke excluding all hemorrhage (11,822 cases, HR=1.03 per SD, P=5.6E-04), and heart failure (13,771 cases, HR=1.04 per SD, P=4.4E-06). Compared with the remaining 95% of FinnGen participants, those in the top 5% of genetically predicted LAmin-indexed had an increased risk of AF (HR=1.19 per SD, P=8.4E-09). Those in the top 5% also had elevations in risk that were not statistically significant for ischemic stroke excluding subarachnoid hemorrhages (HR=1.04 per SD, P=0.36) and heart failure (HR=1.07, P=0.08).

In the US national biobank, All of Us (Denny et al., 2019), the BSA-indexed LAmin polygenic score remained significantly associated with AF (4,859 incident cases, HR=1.06 per SD, P=1.7E-04) and heart failure (5,712 incident cases, HR=1.04 per SD, P=2.0E-02), but not ischemic stroke (66 cases, P=0.3; Supplementary Table 16). In logistic models that included all cases regardless of biobank enrollment date, more cases were identified and the statistical evidence was stronger (13,399 AF cases, OR=1.10 per SD, P=4.9E-19; 14,572 heart failure cases, OR=1.04 per SD, P=1.5E-04).

In addition, 680 participants in All of Us with genetic data had BSA-indexed LAmin volume measurements. The BSA-indexed LAmin polygenic score was associated with these measurements (0.10 SD per SD of the polygenic score, P=8.5E-03). This relationship remained nominally significant when restricted to only the largest subset of participants by genetic identity (N=619 participants with genetic identity similar to Europeans; 0.09 SD per SD, P=1.5E-2).

It can be problematic to extrapolate disease mechanisms from GWAS loci due to uncertainty of the target genes. In the Discussion (line 274), it is proposed that left atrial dilation may be the

phenotype associated with the PITX2 locus. How does this fit in to the extensive pre-existing literature that has studied this locus?

Author response

Our focus had been on drawing a strong link between our findings and the AF literature, and in that context we agree that we didn't address the left atrial structural literature on *PITX2*. To try to keep this concise, we have added one citation pointing to mouse-based analyses that we think exemplifies some of the *PITX2*-atrial structure literature, although we acknowledge that a large number of citations could be made given the intense interest and study of this locus.

Manuscript change

The **Discussion** now refers to the Chinchilla, *et al*, mouse work:

Also notable was the PITX2 locus, which was the first locus associated with AF. In the present GWAS, SNPs at that locus were associated with BSA-indexed LA_{max} and LA_{min}. The lead SNP for AF (rs2129977 from Roselli, et al, 2018) was in close LD with the lead SNP for LA_{max} and LA_{min} (rs2634073; $r^2=0.85$)(Machiela and Chanock, 2015; Roselli et al., 2018). Consistent with clinical expectations, the AF risk allele was associated with greater left atrial maximum and minimum volumes. These analyses excluded participants with a history of AF or abnormal cardiac filling patterns on MRI; therefore, these results support the hypothesis that the PITX2 locus may be associated with an increase in LA volume that occurs prior to AF onset, which would be consistent with experimental data showing atrial enlargement during embryonic development in mice with knocked-down PITX2(Chinchilla et al., 2011).

Table 1. Clinical characteristics of the study cohort. This table could be expanded to include presence/absence of AF as well as co-morbidities and cardiac features, eg underlying

cardiomyopathy/valve defects/congenital heart defects, that could influence left atrial size.

Author response

Thank you, this is a very helpful recommendation. Table 1 has now been updated to include the count data for conditions such as atrial fibrillation (N=0), heart failure (N=0), hypertrophic cardiomyopathy (N=0), valve defects (N=19 with MR, N=3 with MS, N=39 with aortic valve disease), and congenital heart disease (N=4). These numbers are small in part because of our exclusions on heart failure and atrial fibrillation diagnosed before MRI, and also because the UK Biobank is a healthy volunteer cohort.

Manuscript change

In addition, **Table 1** now displays the count data for several conditions:

	Women	Men	Both
N	18,916	16,133	35,049
Age at time of MRI	64 (8)	65 (8)	64 (8)
BMI (kg/m²)	26 (5)	27 (4)	26 (4)
Height (cm)	163 (6)	176 (7)	169 (9)
Weight (kg)	69 (13)	83 (13)	75 (15)
Systolic Blood Pressure (mmHg)	136 (19)	142 (17)	139 (19)
Diastolic Blood Pressure (mmHg)	77 (10)	81 (10)	79 (10)
Left atrium maximum volume (cm³)	64 (15)	79 (19)	71 (18)
Left atrium minimum volume (cm³)	28 (9)	37 (12)	32 (11)
Left atrium stroke volume (cm³)	36 (8)	43 (11)	39 (10)
Left atrium emptying fraction (%)	57 (8)	54 (7)	56 (8)

Mitral regurgitation (%)	10 (0)	9 (0)	19 (0)
Mitral stenosis (%)	3 (0)	0 (0)	3 (0)
Heart failure (%)	0 (0)	0 (0)	0 (0)
Hypertrophic cardiomyopathy (%)	0 (0)	0 (0)	0 (0)
Congenital heart disease (%)	3 (0)	1 (0)	4 (0)
Aortic valve disease (%)	18 (0)	21 (0)	39 (0)
Atrial fibrillation or flutter (%)	0 (0)	0 (0)	0 (0)

Characteristics of the participants who contributed to the GWAS are listed as mean (standard deviation). Count data are listed as number (%).

Figure 1. The middle panel is too small and it is difficult to see the detail in these images. What do the schematics between the two columns of CMR images represent? The font size of the text on all panels could be increased.

Author response

Thanks—we have updated the figure to occupy more vertical space and have tried to increase the font sizes to be more visible. The two columns of CMR images represent the raw image followed by the image with the left atrial segmentation output colorized.

Manuscript change

The Figure now looks as follows:

Reviewer #3 (Remarks to the Author):

* Summary

This paper performed a GWAS analysis using deep learning-derived left atrium (LA) imaging phenotypes of ~40K subjects from the UK Biobank cohort. After identifying 20 loci, the authors performed further statistical analyses, demonstrated the correlation between LA phenotypes and atrial fibrillation (AF) risks and investigated the bi-directional causality using the Mendelian randomisation analysis.

* Major comments

My main concern is about the clarity of the computational methods for determining the 3D LA volumes and detecting abnormal patterns for LA contraction, as well the validation of these methods.

1. All statistical analyses were based on the 3D LA volume measurements, which were derived from separate 2D imaging views using the Poisson surface reconstruction method. However, I could not find the detailed description of this method in the paper or in references [55,56]. Could the paper elaborate this method (Line 403-405)? Given 2D LA chamber segmentations on different short-axis or long-axis views, how exactly is the Poisson surface reconstruction applied?

Author response

Thanks for pointing out the citation issue. In addition to the Kazhdan 2013 paper, we have now added a citation to the Kazhdan 2006 article within the Eurographics archive (a bit trickier to cite since it's not a typical journal article, but we think it does a better job of explaining the core

concept of Poisson surface reconstruction before Kazhdan 2013). Further, we have tried to more clearly explain the three inputs into this algorithm (the points, the normals, and the depth argument). We also link to the python code that executed these functions.

Manuscript change

The **Methods>Poisson surface reconstruction** section states:

To integrate the output from each of the four models into one LA volume estimate, Poisson surface reconstruction was performed (Kazhdan, Bolitho and Hoppe, 2006; Kazhdan and Hoppe, 2013a). Among the views included in the UK Biobank cardiac MRI dataset, none fully captures the 3-D anatomical structure of the LA. The short axis stack only occasionally included the lower portion of the chamber, while the three long-axis (i.e., two-, three-, and four-chamber) views provided only single-slice cross-sections of the LA at different orientations. To integrate information from the four incomplete MRI views into a consistent 3D representation of the LA anatomy, we followed a procedure similar to Pirruccello et al. (2021) (Pirruccello et al., 2021). Briefly, we first co-rotated the left atrial segmentation maps from the MRI views into the same reference system (shared 3D space) using standard DICOM metadata from the Image Position (Patient) [0020,0032] and Image Orientation (Patient) [0020,0037] tags. Then, the perimeters of each 2D atrial segmentation map were extracted, yielding a sparse 3D point cloud. In addition to the point coordinates, the reconstruction algorithm requires as input a vector representing the local normal directions for each point, which is used to constrain the curvature of the reconstructed surface. In our approach, we assumed that each perimeter point's normal vector lay on the MRI view plane and was radially oriented outwards from the center of gravity of the LA segmentation from which the point was extracted. Using three inputs, consisting of the points, the normals, and the depth

argument of 16 (representing the maximum depth of the tree that the library will use for reconstruction), we applied the Poisson surface reconstruction algorithm(Kazhdan, Bolitho and Hoppe, 2006) with the pypoison python binding for the Screened Poisson Surface Reconstruction C++ library v6.13(Kazhdan and Hoppe, 2013b). This yielded interpolated 3-D surfaces from the sparse 3D point cloud. This approach is tolerant to missing segmentation data (e.g., from the frequently missing SAX data) as long as not all available points are coplanar. 3D surfaces of the LA were reconstructed for each of the 50 MRI frames acquired during the cardiac cycle. At each timepoint, the volume of the LA was computed from the reconstructed surface model using the GetVolume routine for triangulated meshes included in the VTK library (Kitware Inc.). From the reconstructed volume traces, we estimated the maximum and minimum LA volumes, as well as LA stroke volume and emptying fraction. Quality control and comparison to other methods for estimating LA volumes are detailed in the Supplementary Note.

The **Code availability** section now reads:

The deep learning models will be returned to the UK Biobank for use by other researchers. The code used to perform Poisson surface reconstruction from segmentation output is located at https://github.com/broadinstitute/ml4h/blob/pd_atria/scripts/mri_la_poisson.py and is available under an open-source BSD license.

2. How is the Poisson surface reconstruction method validated? How accurate are the 3D LA meshes? Are they evaluated qualitatively or quantitatively?

Author response

We now provide a description of quantitative quality assessment and comparison to previously performed manual biplane measurements in the **Supplementary Note**. There is a fundamental

limit to our ability to infer truth because the volumetric short axis stack of the left atrium was not included in the UK Biobank imaging protocol. The lack of volumetric atrial stacks also places a limit on how much we can glean from qualitative model assessment, e.g., by visualizing projections of the 3D models against the various planes, since it's not projecting onto a ground truth volume but instead back onto the same planar data that was used in its construction. Therefore, we make inferences about quality based on the behaviors of our modeling approach with respect to previously reported biplane measurements (which make simplifying assumptions about atrial shape) and with respect to disease correlation.

A small subset of participants (~3,400) had manual left atrial biplane measurements produced by the group of Dr. Steffen Petersen and returned to the UK Biobank for research use. We found the correlation between the biplane L_Amax and our Poisson L_Amax to be 0.814. When we then discarded the 3Ch and SAX data, keeping only the 2Ch and 4Ch data that are typically used for biplane measurements, the correlation between our measurements and the biplane measurements increased from 0.814 to 0.887, consistent with the notion that the Poisson approach can approximate an LA biplane measurement when we only use the same two planes as input data. (Note that we don't expect our Poisson reconstructions to yield the *exact* same values as the biplane measurements because they make different mathematical assumptions. The Poisson reconstructions aim to approximate a possibly complex surface, while the biplane assumption is that we have two views of the same ellipsoid.)

We then found that when we compared the full Poisson model with this 2Ch and 4Ch-only biplane-like Poisson model, the full Poisson model was more strongly associated with prevalent atrial fibrillation diagnosed prior to imaging.

While there are no gapless ground truth left atrial volumes, we considered these results to be supportive of our reconstruction approach in general, and also supportive of the notion that

incorporating additional views to better constrain the left atrial shape based on all available evidence, rather than just the 2Ch and 4Ch views of the biplane approach, made the measurements more disease-relevant.

Manuscript change

We have added the **Supplementary Note>Methods>Comparison with left atrial biplane measurements** section:

The UK Biobank's cardiovascular MRI imaging protocol did not include a volumetric short-axis stack throughout the left atrium(Petersen et al., 2016), so left atrial measurements represent estimates of an unmeasured true left atrial volume. To assess quality, we compared the Poisson surface reconstruction approach with biplane measurements and tested each for association with prevalent atrial fibrillation. Using the R function cor.test, we correlated the Poisson surface reconstruction algorithm-based left atrial volume measurements with biplane-based volumes manually measured by experts(Petersen et al., 2017).

We have added the **Supplementary Note>Results>Comparison with left atrial biplane measurements** section:

We correlated the Poisson surface reconstruction algorithm-based left atrial volume measurements with biplane-based volumes manually measured by experts in 3,401 participants(Petersen et al., 2017). When limiting the inputs into the Poisson surface reconstruction algorithm to only the two- and four-chamber long axis views ("Poisson biplane"), which are the two views used to calculate the biplane volume, the correlation improved for both LAm_{ax} (from $r=0.814$, 95% CI 0.802 to 0.825, $P=2.9E-804$ with the full reconstruction to $r=0.887$, 95% CI 0.880 to 0.894, $P=4.5E-1143$ with the Poisson biplane) and LAm_{in} (from $r=0.768$, 95% CI 0.754 to 0.781, $P=1.1E-659$ to $r=0.860$, 95%

CI 0.851 to 0.868, $P=6.9E-994$). We interpreted these results as supporting the notion that, when presented with the same input information, the modeling approach yields estimates that are similar to the standard biplane estimation.

We then used logistic regression to recapitulate prior observations that individuals with pre-existing atrial fibrillation have larger atrial volumes (Sanfilippo et al., 1990; Sardana Mayank et al., no date). In a subset of 39,148 participants, of whom 808 had atrial fibrillation, both the full Poisson reconstruction and the Poisson biplane reconstruction could be performed. Although the Poisson biplane better correlated with the manual measurements in the previous analysis, the full Poisson reconstruction was more strongly associated with prevalent atrial fibrillation (L_{max} OR 1.72, $P=1.3E-78$ and L_{min} OR 1.86, $P=1.0E-132$) compared to the Poisson biplane model (L_{max} OR 1.65, $P=6.3E-66$ and L_{min} OR 1.80, $P=2.8E-130$).

We interpreted these findings as indicating that (1) the Poisson-based measurements were well correlated with manual measurements, and (2) while full volumetric imaging stacks through the atria were not available to adjudicate correctness, the Poisson-based measurements that incorporated all available views (2ch, 3ch, 4ch, and SAX) were more strongly correlated with atrial fibrillation than the Poisson biplane measurements.

3. Since the short-axis view may not always capture the LA, how does the surface reconstruction method cope if LA is missing in the short-axis view?

Author response

This is an important question and we now explicitly address it in the methods. As long as the Poisson algorithm has access to points that are not simply coplanar, it can usually generate a surface. Since most participants have three long-axis views, losing the orthogonal SAX view is therefore generally not problematic. This is why, in our response to your prior point, we were

able to simply drop both the 3Ch and SAX data while keeping the rest of the algorithm intact to perform the biplane sensitivity analysis.

Manuscript change

We have updated the **Methods>Poisson surface reconstruction** section to mention that the approach is tolerant to missingness:

This approach is tolerant to missing segmentation data (e.g., from the frequently missing SAX data) as long as not all available points are coplanar.

4. How are the LA volumes quality controlled?

Author response

We have now added this to the Supplementary Note. Briefly, we flagged each sample separately for each segmented view. After surface reconstruction, we observed that participants in whom reconstruction failed were significantly more likely to have had at least one quality control flag. (Surface reconstruction failure yields an otherwise uninformative segmentation fault in the underlying reconstruction library, so we do not have additional information about the reason for failure; however, this finding suggests that it is likely correlated with segmentation failures.)

Manuscript change

The **Methods>Poisson surface reconstruction section** now points to the **Supplementary**

Note:

Quality control and comparison to other methods for estimating LA volumes are detailed in the Supplementary Note.

The **Supplementary Note>Methods>Segmentation and reconstruction quality control**

section now describes the analysis:

Automated quality control was performed on the segmentation output to flag putatively invalid segmentations separately for each view. Studies were flagged based on the following heuristics: (a) if they had more than 1 connected component (i.e., if there were pixels in more than one connected surface that were being labeled as left atrium); (b) if the maximum single frame-to-frame change in pixels segmented as left atrium during the 50-frame CINE sequence was greater than 5 standard deviations beyond the population mean; (c) if no pixels were segmented as the left atrium; or (d) if the number of images in the CINE was not 50. The presence or absence of these flags was then tested for association with 3D surface reconstruction failure using logistic regression.

The **Supplementary Note>Results>Segmentation and reconstruction quality control**

section now describes the analysis:

QC-flagged samples (due to more than 1 connected component, frame-to-frame pixel changes greater than 5 standard deviations above the mean, the absence of left atrial pixels, or an abnormal number of CINE images as detailed in the Supplementary Methods) were significantly more likely to fail to achieve a successful Poisson reconstruction (OR 1.4, $P=1.3E-19$). Among the left atria that were successfully reconstructed, we tested whether the presence or absence of any of the QC flags was associated with volumetric measurements. However, the distribution was similar regardless of QC status (Supplementary Figure 10); the presence of QC flags was statistically non-significant for LAmin (0.020 SD greater with a flag, $P=0.06$) and had a similarly small effect estimate for LAmax (0.036 SD greater with a flag, $P=5E-04$). Therefore, all samples that were successfully reconstructed were retained for analysis.

We have added **Supplementary Figure 10** to the **Supplementary Note** that shows the stacked histogram of those with and without QC failures:

Histogram of distribution of Lamin volumes among participants with successful left atrial surface reconstruction. Values for those with at least one QC-flagged MRI segmentation series are colored in red, while those for participants with no flagged series are colored in turquoise. The segmentations are stacked.

5. Line 422-423: "A deep learning model was then trained to classify filling patterns as representing a normal atrial contraction or not." What do these filling patterns look like? Could the paper provide some illustrative examples for normal patterns and abnormal patterns?

Author response

Great suggestions, thanks. We have now plotted exemplar curves using actual data. The chief visual distinction (to our eyes) is the lack of the plateau phase and atrial kick with the abnormal contraction patterns (best seen in frames 30-50 below).

Manuscript change

Supplementary Figure 2 has been added to the **Supplementary Note** depicting normal and abnormal examples:

Normal pattern

Abnormal pattern

Curves depicting the data used in the abnormal filling pattern detector are displayed for one individual with a normal pattern (top panel) and one with an abnormal pattern (bottom panel). For visual simplicity, only the left atrial and left ventricular curves from the four-chamber view are displayed. Each datum represents the cross-sectional area at each time point for each chamber. Values are scaled between 0 and 1 (y-axis) on a per-chamber basis so that the maximum is always 1 and the minimum is always 0 for each

chamber independently, which is consistent with how the data are transformed prior to being input into the deep learning model. Values are visualized at the 50 timepoints during image acquisition (x-axis). Both panels begin at ventricular end-diastole. The example in the top panel reveals a triphasic pattern: ventricular systole continues until timepoint 20, passive ventricular filling until timepoint 45, and then an active ventricular filling phase due to atrial systole from 45-50. The example in the bottom panel reveals a biphasic pattern: there is only ventricular systole until timepoint ~25 and ventricular diastole for the remainder of the cycle, with the atrium passively filling and emptying in parallel.

* Other comments

6. Figure 1, 4Ch view

Is the LA a bit over-segmented or maybe not? Does the annotation follow a consistent manner to cut the upper boundary of the LA?

Author response

Thanks, we agree that it looks a bit over-segmented. The sample itself was chosen for display because its randomized sample ID was the first in numerical order, but the 4ch view for this particular sample has a left atrium that transitions into a large pulmonary vein. We feel that this degree of over-segmentation is similar to what we can see in the cardiovascular MRI literature even for manual segmentation (e.g., <https://jcmr-online.biomedcentral.com/articles/10.1186/1532-429X-12-65> Figure 2, middle panel), but nevertheless we would keep the segmentation tighter if we were manually annotating the image. In the training data, we do exclude the pulmonary vein openings as well as the left atrial appendage. We now clarify this aspect of the annotation procedure in the manuscript.

Manuscript change

The **Methods>Semantic segmentation and quality control** section now states:

When present, the left atrial appendage was excluded, as were the pulmonary vein openings; the atrial and ventricular blood pools were distinguished by tracing a linear boundary at the base of the atrioventricular ring.

7. Line 446-447: "... adjusting for the MRI serial number, sex, age, and the interaction between sex and age."

What is the MRI serial number here please? UK Biobank runs three imaging centres each with a MRI scanner of the same model. Is the MRI serial number used to adjust the inter-site difference?

Author response

Yes this is correct - our idea was to account for differences in UK Biobank imaging centers in a way that would be future-proof in case the assessment centers started using a new machine that might have different calibration (for example). It is true that right now there are four machines that we are able to detect at four centers: Bristol, Cheadle, Newcastle, and Reading—one machine per center. (And a vanishingly small number of participants have imaging from Bristol, which is a pilot center as far as we understand, so there are three active imaging centers as you point out.) Therefore, it is equally true at present that we could adjust for the center instead of the serial number and achieve the exact same adjustments. We now clarify our rationale in the text.

Manuscript change

The **Results>Evaluation of the relationship between the left atrium, phenotypes, and cardiovascular diseases** section now explains the rationale the first time we mention the MRI serial number:

For prevalent disease that was diagnosed prior to the time of imaging, linear models were used to test for an association between each disease (as a binary independent variable) and LA phenotypes (as the dependent variables), adjusting for the MRI serial number to account for inter-site differences, sex, age, and the interaction between sex and age.

8. Line 450-451: "A Cox proportional hazards model was used, with survival defined as the time between MRI and either the time of censoring, or disease diagnosis."

How is the disease diagnosis information obtained? The LA volumes are derived at the first imaging visit of the UK Biobank participants. Are the disease diagnosis codes obtained from their second imaging visit? If that is the case, what is the average duration between the two?

Author response

Thanks for this question, we have now taken the opportunity to provide a more clear explanation in the text. To add more detail, the data come from the UK Biobank's hospital episode statistics, which are longitudinal data that are refreshed frequently. They are not directly linked to the imaging visits - the data are automatically pulled into the UK Biobank's database and they are noted at the time of a procedure, a hospitalization, or death. Therefore, the amount of follow-up time after MRI depends on when the MRI was performed. On average, this was 2.2 years' worth of follow-up time after MRI until the recommended censoring date at the time the analysis was conducted (available from UK Biobank and periodically updated at https://biobank.ndph.ox.ac.uk/ukb/exinfo.cgi?src=Data_providers_and_dates). The UK Biobank defines that date as the last day of the month for which the number of records is greater than

90% of the mean of the number of records for the previous three months. For clarity, these diagnostic codes are not tied to a second imaging visit and do not depend on the participant having any additional contact with the UK Biobank.

Manuscript change

The **Results>LA traits are associated with AF, heart failure, hypertension, and stroke** section states:

In the 2.2 years of follow-up time (mean) available on average after MRI acquisition, the risk of incident AF was increased among those with greater LAmin (293 cases; HR 1.73 per standard deviation [SD] increase; 95% CI 1.60-1.88; P=4.0E-39).

The **Methods>Definitions of diseases and medications** section now explains:

We defined disease status based on self report, ICD codes, death records, and procedural codes from the UK Biobank's hospital episode statistics data (Supplementary Table 17). These data were obtained from the UK Biobank in June 2020, at which time the recommended phenotype censoring date was March 31, 2020. The UK Biobank defines that date as the last day of the month for which the number of records is greater than 90% of the mean of the number of records for the previous three months (https://biobank.ndph.ox.ac.uk/ukb/exinfo.cgi?src=Data_providers_and_dates).

REVIEWERS' COMMENTS

Reviewer #1 (Remarks to the Author):

It is a well-worked out second version of the manuscript.

Pirruccello et al provided very exciting and novel way to calculate cross-sections of a 3-dimensional representation of the LA.

The genetic analyzes are clearly explained and the PRS is well described.

I only have tree minor comments

Overlap of left atrial loci with atrial fibrillation loci

"We identified the lead SNPs associated with AF from Supplementary Table 16 of Roselli, et al."

This meta-analysis included controls from UKB if I understand it correctly, this seems to lead to a sample overlap with the sample in the cMR study in UK Biobank.

Could that be a problem when you " counted the number of AF SNPs that fell within 500kb of the LA lead SNP from our study " ?

MR part

".Variants that were associated with LAmin with $P < 1E-06$ were clumped and ambiguous alleles were excluded, leaving 19 SNPs"

MR studies normally included only SNP with a P lover than 5×10^{-8} . Why did you use this threshold?

Other phenotypes

Is it possible to calculate "LA passiv EF" from your algorithm?

-

Reviewer #2 (Remarks to the Author):

Concerns raised have been adequately addressed.

Reviewer #3 (Remarks to the Author):

Thanks the authors for the revised the submission, which has satisfactorily addressed all my comments. I do not have further comments.

Reviewer #1

It is a well-worked out second version of the manuscript.

Pirruccello et al provided very exciting and novel way to calculate cross-sections of a 3-dimensional representation of the LA.

The genetic analyzes are clearly explained and the PRS is well described.

I only have tree minor comments

Overlap of left atrial loci with atrial fibrillation loci

"We identified the lead SNPs associated with AF from Supplementary Table 16 of Roselli, et al."

This meta-analysis included controls from UKB if I understand it correctly, this seems to lead to a sample overlap with the sample in the cMR study in UK Biobank.

Could that be a problem when you " counted the number of AF SNPs that fell within 500kb of the LA lead SNP from our study " ?

This is a very good point, and we will offer a few perspectives to try to address it. First, in response to this point, we did look into the results from conducting the analysis using the Christophersen et al 2017 Nature Genetics AF loci (which did not use UK Biobank in the primary analysis) instead of the Roselli AF loci. The locus overlap between LA GWAS loci and AF GWAS loci remained greater than expected by chance ($P=0.007$). The major limitation with the Christophersen AF GWAS is that the smaller sample size in that study yielded about 1/6th the number of loci as the Roselli AF GWAS (23 loci vs 124 loci). As might be expected, we therefore found proportionally fewer overlapping loci (2 overlapping loci for Christophersen vs 8 overlapping loci for Roselli) and we think this paucity of data would be less informative for the readers than the Roselli AF locus overlap analysis. The second perspective is that the sample overlap is likely to be substantial from the perspective of the LA GWAS, but very modest from the standpoint of the AF GWAS. There were 585k participants in the Roselli et al GWAS, including 350k from UK Biobank. So we can deduce that about $(350k/500k)$ 70% of participants in the LA GWAS were probably also in the Roselli AF GWAS. But 70% of ~35k LA GWAS participants also implies that only $(24.5k/585k)$ 4% of the Roselli AF GWAS was drawn from these LA GWAS participants. Since the overlap analysis used only genome-wide significant loci, we think that this makes it unlikely that the 4% of participants substantially drove the AF loci to significance. The third perspective is that, because this was an overlap analysis rather than a two-sample Mendelian randomization, we would consider the results to be of interest even with 100% sample overlap. For example, imagine that we were studying HDL and coronary artery disease. Even if the GWAS of both were to be in the same samples, the overlap of loci that achieved genome-wide significance for HDL alone, coronary artery disease alone, or both would still be of interest for interested readers in this admittedly non-causal framework. Finally, the

directional/causal tests were formalized in two-sample Mendelian randomization analyses, and for those analyses in this manuscript we did use the Christophersen AF loci (despite their lower power) in order to avoid confounding from even the modest degree of sample overlap with our study and the Roselli AF GWAS.

MR part

".Variants that were associated with LAmin with $P < 1E-06$ were clumped and ambiguous alleles were excluded, leaving 19 SNPs"

MR studies normally included only SNP with a P lower than 5×10^{-8} . Why did you use this threshold?

We consider the P-value threshold for MR instruments to be a choice based on the bias-variance trade-off (as discussed in the Burgess & Thompson book "Mendelian randomization" first edition, section 5.4.2). We agree that $P < 5 \times 10^{-8}$ is a common threshold and is probably the most stringent one that is routinely used. This leaves some explanatory power "on the table" but it has very low bias, so we also agree that it is a very good default. In the present GWAS, which yielded a modest number of loci, it is appealing to use more loci which explain more of the overall variance (i.e., to risk some bias in order to gain statistical power). The results from the analysis conducted using only $P < 5 \times 10^{-8}$ variants were also included as a sensitivity analysis (Supplementary Table 11). Reassuringly to us, the MR using $P < 1e-6$ instruments and the MR using $P < 5e-8$ instruments both had concordant observations for LAmin. The overall statistical evidence was stronger using the 19 SNPs in the $P < 1e-6$ analysis, while the effect size estimate was greater using the 10 SNPs in the $P < 5e-8$ analysis (but with weaker statistical evidence). As to why the precise P value of $1e-6$ was chosen, this is essentially arbitrary: this author (JPP) routinely clumps variants at that threshold to try to gain insight into traits based on not only the $P < 5e-8$ significant loci, but also at loci that are near-significance, and which often become significant in future GWAS with larger sample sizes. (A follow up question could be why, then, not routinely clump down to $P = 0.05$ or $P = 1e-3$? The rationale is that clumping at such weak P value thresholds becomes expensive, since so many SNPs achieve those modest P values.)

Other phenotypes

Is it possible to calculate "LA passiv EF" from your algorithm?

Thanks for this question, which is about the left atrial passive emptying function (LAPEF). LAPEF is an important parameter that has been hypothesized to mechanistically link left atrial dysfunction to pulmonary vein dilation (a probable trigger for atrial fibrillation). The very short answer is that our current approach would be useful input to make that analysis possible, but it's not a value that is produced directly by our method. Since our method produces a volume estimate for every time point during the cardiac cycle (50 time points in the UK Biobank imaging), the calculations that we report in this manuscript can be obtained by taking the maximum, the minimum, or their combination. To instead calculate LAPEF, one could use these same data and manually mark the timepoints for the maximum volume and the plateau volume (just before active emptying). This would be time consuming to manually annotate for every study, so we would probably propose to automate this rather than to manually annotate tens of

thousands of studies. We could imagine treating this as a 1D semantic segmentation problem, where there are 50 volume measurements per person, labeled as being in the filling, passive emptying+plateau, or active emptying phases for a few hundred individuals to form a training set. Then, a 1D U-Net model could be trained to recognize those phases, allowing us to take the max volume and the last volume from the passive/plateau phase and compute the LAPEF. (Alternatively, perhaps a standard 1D CNN could be used with keypoint regression, or similar approaches.) Having said that, this remains a hypothesis that we have not yet tested, so although we know it can be manually done, and we think it could be done at scale, this is not something that we have actually tested/computed. We believe such an analysis is worth attempting for future work.

Reviewer #2

Concerns raised have been adequately addressed.

Reviewer #3

Thanks the authors for the revised the submission, which has satisfactorily addressed all my comments. I do not have further comments.